The Neotropical land snails (Mollusca, Gastropoda) collected by the ‘Comisión Científica del Pacífico’

Breure Abraham S.H. ashbreure@gmail.com 1 2
Araujo Rafael 3
1 Royal Belgian Institute of Natural Sciences , Brussels , Belgium
2 Department of Zoology, Naturalis Biodiversity Center , Leiden , Netherlands
3 Museo Nacional de Ciencias Naturales-CSIC , Madrid , Spain
Rogers Sean
Electronic publication date: 2017 Mar 14
Publication date: 2017
Volume: 5
Electronic Location ID: e3065
Received 2016 Nov 23; Accepted 2017 Feb 4
Copyright: ©2017 Breure and Araujo
Copyright year: 2017
Copyright holder: Breure and Araujo
License: This is an open access article distributed under the terms of the Creative Commons Attribution License, which permits unrestricted use, distribution, reproduction and adaptation in any medium and for any purpose provided that it is properly attributed. For attribution, the original author(s), title, publication source (PeerJ) and either DOI or URL of the article must be cited.
License URL: https://creativecommons.org/licenses/by/4.0/

Keywords: Mollusca, Gastropoda, Biohistory, Biographical data, 19th century, Expedition, South America, Central America

Funding: European Community Research Infrastructure Action ES-TAF-5322 This work was supported by a grant from the SYNTHESYS Project (http://www.synthesys.info) which was financed by European Community Research Infrastructure Action under the FP7 Integrating Activities Programme; the grant was awarded to ASHB under number ES-TAF-5322. The funders had no role in study design, data collection and analysis, decision to publish, or preparation of the manuscript.

==============================
The land snails collected by the ‘Comisión Científica del Pacifíco’ (CCP), a Spanish expedition to South and Central America from 1862–1866, are restudied and revised. The historical context of the expedition and the study of its collected material are described. Biographical data is given for the main persons involved. The land snails were previously studied by Joaquin Hidalgo between 1867 and 1893. A total of 3,470 specimens belonging to 211 species are treated in this paper. Of 34 species mentioned by Hidalgo is his catalogue, the corresponding material could not be located. Bulimus visendus Hidalgo, 1869 is now placed in the genus Synapterpes Pilsbry, 1896, a new combination.

Introduction

On the 10th August 1862, a group of Spanish scientists sailed away from Cadiz for an expedition that would last until the 18th January 1866. This group of men, known as the ‘Comisión Científica del Pacifíco’ (CCP), would visit many parts of South, and some parts of Central, America and collected many specimens of animals, artefacts and plants and made photographs and illustrations of the remarkable things they observed (Barreiro, 1926; Miller, 1968; Puig-Samper, 1988; Calatayud, 1994; López-Ocón, 2003). The group (Fig. 1) consisted of Patricio Paz y Membiela (zoologist and president of the CCP), Manuel Almagro y Vega (anthropologist and ethnographer), Fernando Amor y Mayor (geologist and entomologist; vice-president), Francisco de Paula Martínez y Sáez (zoologist), Marcos Jiménez de la Espada (zoologist), Rafael Castro y Ordóñez (photographer and draftsman), and Juan Isern (botanist). The taxidermist Bartolomé Puig y Galup was selected shortly before their departure, but left the CCP in autumn 1863 whilst in Chile. The land molluscs, which were mainly collected by Paz y Membiela and Martínez y Sáez (see below), were studied by Joaquin González Hidalgo who published an extensive catalogue with supplements (Hidalgo, 1870; Hidalgo, 1875; Hidalgo, 1893a; Hidalgo, 1893b).

Figure 1 Members of the ‘Comisión Científica del Pacífico’.

Left to right, upper row: Isern, Jiménez de la Espada, Almagro; lower row: Castro, Amor, Paz, Martínez, Puig (MNCN archive).

This study of the molluscan part of the CCP material deserves to be placed in its historical context to understand several details that will be discussed below. In the section ‘context of the collection’ we will therefore briefly elaborate on the creation of the CCP, its itinerary, the way the study of its collected material was undertaken and the results that were published. We will also present biographical data on relevant CCP members and the persons directly involved in the study of the land molluscs. Until now, the full extent of the CCP collection of land molluscs was not precisely known, as Almagro (1866) presumably only gave numbers of the material which was exhibited to the public in 1866, and Hidalgo listed only the species recognised without mentioning any numbers. A partial summary, listing only the new species described by Hidalgo, was given by Calvo (1994). The recent discovery of an undescribed species which appeared to be mixed in with other material (Breure & Araujo, 2015) stimulated this current study, which aims to revise all of the known land mollusc material collected by the CCP.

Material and Methods

The core historical material of the Museo Nacional de Ciencias Naturales mollusc collection is from three Spanish malacologists: Paz y Membiela with 40,000 specimens representing 12,000 species and subspecies, Hidalgo with 8,000 species and Azpeitia with 80,000 specimens of 8,171 species (6,594 gastropods and 1,577 bivalves) (Barreiro, 1992). Other historical material comes from the founder of the Museum, P. Franco Dávila (1711–1786), and from M. Graells (1809–1898; Fig. 2), who was director of the museum between 1851 and 1867. The material which had been previously recognised as originating from the CCP, has been restudied and identified according to modern literature. Material from the collections of Paz y Membiela and Hidalgo is only included if this CCP origin could be ascertained (e.g., by the label type or by the collection locality). Material from the Azpeitia collection is included when a similar lot had been found with an undisputed CCP origin.

Figure 2 Portraits of persons mentioned in this study.

(A) Mariano Graells, 1862 (MNCN-CSIC); (B) Patricio Paz y Membiela, unknown date (CCHS-CSIC).

Besides references to the original publications of the species, only references are given to publications citing the CCP material. The section on systematics follows the classification given by Bouchet & Rocroi (2005), with the exception of the Orthalicoidea. All nomenclatural innovations are explicitly identified in the individual species accounts and any literature citations based on misidentifications are identified as such to distinguish them from intended synonymies.

Type material lists only the primary types of taxa (if known), unless taxa were based on material collected by the CCP. Abbreviations for depositories: IFML, Instituto y Fundación Miguel Lillo, Tucumán, Argentina; MNCN, Museo Nacional de Ciencias Naturales, Madrid, Spain; MNHN, Muséum nationale d’Histoire naturelle, Paris, France; NHMUK, Natural History Museum, London, UK; NMW, National Museum Wales, Cardiff, UK; RBINS, Royal Belgian Institute of Natural Sciences, Brussels, Belgium; ZMB, Zoologisches Museum, Humboldt-Universität für Naturkunde, Berlin, Germany; ZSM, Zoologische Staatssammlung, München, Germany. Other abbreviations used: Coll., collection; H, shell height; leg., legit; /, end of line in quotation of original text.

Results

The context of the collection

History of the CCP

The expedition and the CCP were organised by the Spanish government in the last years of the reign of Isabel II, during a very boisterous political time. The CCP, composed of several naturalists, was included in a military expedition to visit some of the former Spanish colonial regions in South and Central America (Río de la Plata, Valdivia, Valparaíso, Copiapó, Cobija, El Callao, Guayaquil, Nueva Granada, Central America and Mexico) which had recently become independent. The Commission members were shipped in the frigate “Triunfo” under the commandant Croquer; other ships of the squad were the frigate “Resolución” and the schooners “Covadonga” and “Vencedora”, all under general Pinzón as chief of the Expedition.

The CCP was considered a national priority connected to the protection programme promoted by the (French) Société Impériale Zoologique d’Acclimatation and the framework of a pan-hispanist political and cultural movement, whose objectives agreed with those of Mariano de la Paz Graells, who was adviser of Queen Isabel II, Director of the Museo Nacional de Ciencias Naturales and Jardín Botánico, member of the Commission organisation and editor of the scientific instructions for the expedition. Nevertheless, military goals were probably the main objective in the organisation of the expedition (Puig-Samper, 1988). Although the CCP started out together, they split up frequently in to smaller groups once they reached South America and the itinerary of the expedition is thus rather complicated (Calatayud, 1994: 249–282). A brief description was published immediately after the return to Spain by Almagro (1866); further details may be found in Puig-Samper (1988), Calatayud (1994), and López-Ocón (2003).

Before the CCP left, rather detailed instructions had been made about what especially had to be collected (Puig-Samper, 1988), and among the zoologists, tasks were divided which is reflected in their biographies. As may be seen, all CCP members mentioned below had links to Spanish universities or academic centres at the time of their selection.

Patricio Paz y Membiela (1808–1874; hereafter: Paz, Fig. 3) came from a marine and military background and travelled all over the world having visited South America three times in total (Barreiro, 1992: 438), but apart from the visit with the CCP no dates nor itineraries of his travels are known. He formed an excellent shell collection, which probably originated from his relationship with the Cuban naturalists Felipe Poey and Nicolás Gutiérrez in Matanzas and La Habana respectively. He was designated as President of the Commission in 1862, but his continuous confrontations with the commandant of the “Triunfo” lead to his resignation to Queen Isabel II in 1863; he left the CCP in Callao. Once the CCP had returned to Spain, he was entrusted along with Francisco de Paula Martínez in the preparation of the zoological material for a public exhibition in the Botanical Garden in Madrid in 1866 (López-Ocón & Badía, 2003). After his death, his mollusc collection comprising 12,000 species and 40,000 specimens was sold to the MNCN (Barreiro, 1992: 437).

Figure 3 Portraits of persons mentioned in this study.

(A) Francisco de Paula Martínez y Sáez, unknown date (Calatayud, 1994); (B) Manuel Almagro, unknown date (Servicio de Reproducción de Documentos Biblioteca Tomás Navarro Tomás, CCHS-CSIC).

Fernando Amor y Mayor (1822–1863; hereafter: Amor) finished his studies of pharmacy in 1845 in Madrid, and became full professor in the Institutes of Cuenca, Córdoba and Valladolid. He had good contacts with Mariano de la Paz Graells, who was MNCN director from 1851 and one of the scientific advisors of Queen Isabel II. Amor travelled in Morocco during 1859, probably associated to a military expedition to explore the future territory of the Spanish war in Morocco (Barras de Aragón, 1949; Puig-Samper, 1988). In 1862 he was designated as a member and vice-president of the CCP, and entrusted with geology and entomology. He died in San Francisco during the expedition possibly due to a disease contracted in the Atacama desert (Perejón, 2012). He wrote a diary, probably lost in the fire of the “Triunfo”, the ship on which the CCP travelled during the expedition. Part of this diary was saved by Barreiro (1926).

Francisco de Paula Martinez y Sáez (1835–1908; hereafter: Martinez, Fig. 4), finished his studies of natural sciences in 1857 in Madrid, and worked at the MNCN entrusted with the collections of vertebrates. He was professor of mineralogy and botany at the Central University in Madrid during 1861–1862, and full professor of natural history in the institute of Teruel. In 1862 Martinez was designated as member and secretary of the CCP (Gogorza, 1908). He was entrusted with aquatic mammals and reptiles, fishes, crustaceans, annelids, molluscs and zoophytes. He replaced Paz as president after his decomission and the death of Fernando Amor y Mayor in 1863, and planned and executed the last part of the Commission’s itinerary (“El gran viaje”) on the rivers Napo and Amazonas with Manuel Almagro, Marcos Jiménez de la Espada and Juan Isern. He wrote a diary of the expedition (Calatayud, 1994), and the books ‘Moluscos del Viaje al Pacífico, 2. Bivalvos marinos’ (Martínez, 1879?) and ‘Distribución metódica de los vertebrados’ (Martínez, 1879).

Figure 4 Portraits of persons mentioned in this study.

(A) Juan Isern, unknown date (Blanco, Rodríguez & Rodríguez, 2006); (B) Joaquin Hidalgo, 1882 (Crosse archive).

Manuel Almagro y Vega (1834–1895; hereafter: Almagro, Fig. 5) studied medicine in Cuba, Madrid and Paris, where he worked in the hospitals des Enfants, Dieu and la Pitié. In 1862 he was designated as member of the CCP for anthropologic and ethnographic studies. Almagro was one of the first professional anthropologists making field studies in the Americas (Puig-Samper, 1988). He wrote an analysis after the return of the Commission for the exhibition of the material at the Real Jardín Botánico in Madrid in 1866 (Almagro, 1866). This exhibition, an explicit wish of the CCP’s political and scientific sponsors, brought the results of the CCP to the general public and was considered a success (López-Ocón & Badía, 2003).

Figure 5 Portraits of persons mentioned in this study.

(A) Florentino Azpeitia, unknown date (photo R. Araujo); (B) Hippolyte Crosse, unknown date (Tual & Fischer, 1899).

Juan Isern Batlló y Carrera (1825–1866; hereafter: Isern, Fig. 6) studied botany and medicine in Barcelona, Girona and Madrid. He was in contact with foreign botanists like Willkomm and Webb, exchanging with them Catalonian and German plants. He worked at the MNCN and the Real Jardín Botánico in Madrid since 1851 where he was in direct contact with Graells and Miguel Colmeiro, directors of both scientific centres respectively. In 1862 Isern was designated as member of the CCP for botanical studies. He wrote an unpublished diary that is today in the archive of the Real Jardín Botánico (Blanco, Rodríguez & Rodríguez, 2006).

Figure 6 Portraits of persons mentioned in this study.

(A) Louis Pfeiffer, 1856 (Wheeler, 1949); (B) Rudolph Philippi, unknown date (CCHS-CSIC).

Marcos Jiménez de la Espada (1831–1898) studied natural sciences in Madrid and in 1853 worked at the Central University. From 1857 he worked in the collections of the MNCN until his designation as member of the CCP for geological, zoological, anthropological, ethnographical, botanical and geographical studies. Although he was a disciple of Graells, Espada was rather critical about the organisation of the Commission (Puig-Samper, 1988). He published ‘Vertebrados del Viaje al Pacífico, Batracios’ (Jímenez de la Espada, 1875).

We will end this section with biographical data on persons who, although not members of the CCP, are important in the rest of this study. In Madrid two persons were related to the material collected by the CCP, Joaquin Hidalgo and Florentino Azpeitia. Abroad, only a few malacologists were entrusted with descriptions of part of the new species among this material, Hippolyte Crosse, Louis Pfeiffer and Rudolph Philippi.

Joaquin G. Hidalgo (1839–1923; Fig. 7) studied medicine in Madrid at the San Carlos Hospital, and afterwards natural sciences at the Central University. He started with an interest in Mineralogy but decided to finalise his study in medicine on the advice of his professor Rafael Martinez y Molina; he graduated in 1861 and settled in Madrid as a medical doctor. Nevertheless, he began in these years with his collection of shells and his first naturalistic travels within Spain subsidised by Pedro González de Velasco (1815–1882), who worked at the San Carlos Hospital. In 1860 he came into contact with the military Patricio Paz y Membiela in Barcelona and together they worked on his extensive malacological collection. This was probably the reason why in 1862 Paz invited Hidalgo to become a member of the CCP as a naturalist; an offer which Hidalgo declined because of his medical practice. In his place Martínez became member of the Commission. During 1862 and 1875 Hidalgo gave classes at the Central University in zoology, mineralogy and botany. He travelled to Paris in 1865 and 1868, where he was in contact with Deshayes, Crosse and Fischer. Hidalgo had also relationships with Gassies, Souverbie, Guestier, Morelet, Morlet, Jousseaume, Fischer Jr., Dautzenberg, Dollfus, de Folin, Petit de la Saussaye, and Locard. We have found two documents compiled by Hidalgo that lead us to suggest he was sensitive to the opinions of others about his work. The first [Enumeración: MNCN Library F-II-5727] listed the comments of foreign colleagues as published in their own work (if not in French, translated into Spanish); the comments are mainly from Crosse, with additional ones from e.g., P. Fischer, Jeffreys, Dautzenberg, Kobelt, Pfeiffer, and Drouët. The second document (Hidalgo, 1918?) [Relación: MNCN Library F-II-5737] gives an overview of excerpts (translated in Spanish) of 47 correspondents who wrote favourable sentences about his work in letters addressed to Hidalgo. Although, unfortunately, the correspondence of Hidalgo has not been located, this document was used for a partial reconstruction of his network (Breure & Backhuys, 2017). Both documents had been printed and were apparently primarily aimed at Spanish readers. Hidalgo published his malacological manuscripts in the Journal de Conchyliologie, the Real Academia de Ciencias Exactas, Físicas y Naturales de Madrid, and the Sociedad Española de Historia Natural founded by Velasco, Martinez Molina and Zapater. Importantly, Crosse and Fischer helped Hidalgo in the drawing and publication of the plates of his manuscripts. In 1867 he published his first catalogue of Spanish marine molluscs. Afterwards, he published the books on the CCP molluscs with the plates made in Paris (see below). Thanks to Graells, in 1877, Hidalgo was admitted to the Real Academia de Ciencias Exactas, Físicas y Naturales and at the MNCN. He worked again at the University between 1888 and 1897 in botany, mineralogy and zoology, and in 1897 became full professor of mineralogy. He was also involved with the mineral collection at the MNCN. In 1900 he changed the professorship of mineralogy for one in molluscan zoogeography. Hidalgo was director of the MNCN from 3 July 1900 to 2 July 1901 (Barreiro, 1992: 301, 309). He donated his malacological library (c. 2,000 publications) and sold his shell collection to the MNCN in 1913 (Barreiro, 1992: 321, 455–457). In summary, he published 7,600 pages on malacology with 336 plates (made by Arnoul, Delahaye, Laporta and Arroyo) (Hidalgo, 1913?); a bibliography was published by Azpeitia (1923).

Figure 7 Handwritings.

(A) P. Paz (MNCN-CSIC); (B) J. Hidalgo (Crosse archive).

Florentino Azpeitia Moros (1859–1934; Fig. 8) was professor of geology and paleontology in the Escuela Especial de Ingenieros de Minas in Madrid. He was friendly with Hidalgo since 1883, when Azpeitia was treated by Hidalgo as a medical doctor because of gastric fever. From this point, the two men worked together on malacology, Hidalgo being the master and Azpeitia the disciple. He became member of the Sociedad Española de Historia Natural in 1897 for his studies in molluscs and diatoms. Azpeitia was the author of numerous scientific works on geology, botany and zoology; some of the most importance were his ‘Monografía de las Melanopsis vivientes de España’ (1929) and ‘Conchas bivalvas de agua dulce de España y Portugal’ (1933) (Álvarez Halcón, 1997; Álvarez Halcón, 1998). His molluscan collection, with 80,000 specimens of 8,171 species (6,594 gastropods and 1,577 bivalves) was donated to the MNCN in 1934 (Barreiro, 1992: 342).

Figure 8 Labels.

(A–C) Supposedly original (field) labels written by Paz; (D, E) Labels from the Paz collection, written by Hidalgo; (F) Label from the Hidalgo collection; (G, H) Labels from the Azpeitia collection; (I) Old MNCN label; (J) Modern MNCN label.

Hippolyte Crosse (1826–1898; Fig. 9) studied law, but had an interest in natural history from a young age. After a trip to southern France, Corsica and Sicily in 1849, he devoted himself totally to malacology. The Journal de Conchyliologie had been established in 1850 by Petit de la Saussaye, but ceased to appear after a few years. In 1856 it was resurrected by P. Fischer and A.C. Bernardi, and in 1861 Crosse joined them, soon becoming managing director. Together with Fischer he made the journal one of the outstanding malacological journals of the late 19th century (Poyard in Poyard et al., 1898?: 3–6). As managing director he was in contact with all the major malacologists of the era, and received many type specimens of species published in the journal (Fischer-Piette, 1950). From his extensive correspondence with Hidalgo, we know that he also regularly received specimens of CCP material for his own collection, which was auctioned after his death (Breure & Backhuys, 2017; Tual & Fischer, 1899).

Figure 9 Material collected by the CCP.

(A–L) Helicinidae. Bourciera heliciniforme (Pfeiffer, 1853), MNCN 15.05/13857, (A) ventral view, (B) umbilical view, (C) apical view; Helicina angulata Sowerby II, 1842, MNCN 15.05/76223, (D) ventral view, (E) umbilical view, (F) apical view; Helicina brasiliensis Gray, 1824, MNCN 15.05/39940, (G) ventral view, (H) umbilical view, (I) apical view; Helicina variabilis Wagner, 1827, MNCN 15.05/39941, (J) ventral view, (K) umbilical view, (L) apical view. Scale 5 mm.

Louis Pfeiffer (1804–1877; Fig. 10) studied medicine at the Universities of Göttingen and Marburg, after which he did postgraduate work in Paris and Berlin. In autumn 1826 he settled in his city of birth Kassel to practice medicine. After his first marriage in 1833 he gave up his medical practice and devoted himself to botany and malacology, making extensive excursions through Germany and the Low Countries. In 1838–1839 Pfeiffer made a trip to Cuba, together with Johannes Gundlach, which had a significant bearing on his future work. After his return to Germany he received further Cuban shells from Gundlach and Felipe Poey. Pfeiffer made frequent trips to Paris and London to consult literature not accessible in Kassel and to study the collections brought to Europe by the great French voyages, and especially those of Hugh Cuming in London. After the death of his youngest son during the Franco–Prussian war in 1870 his health was much impaired (Wheeler, 1949).

Figure 10 Material collected by the CCP.

(A–F) Neocyclotidae. Buckleyia martinezi (Hidalgo, 1866), MNCN 15.05/3232, (A) ventral view, (B) apical view; Neocyclotus crosseanus (Hidalgo, 1866), MNCN 15.05/3217, (C) ventral view; Neocyclotus giganteus (Sowerby I in Reeve, 1842), MNCN 15.05/17560, (D) ventral view, (E) umbilical view, (F) apical view. Scale 5 mm.

Rudolph Philippi (1808–1904; Fig. 11) was sent at a young age to Switzerland to have private education by the renowned Swiss philosopher Pestalozzi. He soon became interested in the flora and fauna, but graduated as a Doctor of Medicine in Berlin in 1830. During a visit to southern Italy in 1831–1833 he studied the molluscs and the geology of the region. In 1835 he got a position as professor at the Polytechnic Institute of Kassel. Soon afterwards he suffered from health problems and returned to Naples from 1837–1839. Due to the unstable political circumstances he decided to emigrate in 1850 and was appointed as professor of German in Valdivia, Chile. In 1853 he became director of the Museo Nacional de Historia natural in Santiago, as well as professor of botany and zoology at the University. In 1854 he was rejoined by his family, which also brought his library and scientific collections to Santiago. He made important contributions to the knowledge of the flora and fauna of Chile (Emig, 2015). On the 18th May 1863 the CCP members met with Philippi in the Santiago museum and they revised collections in the museum, made an excursion together with Philippi and their visit ended with a banquet on the1st June (Blanco, Rodríguez & Rodríguez, 2006: 112–114). Philippi named several species of plants and molluscs after CCP members (this study; Blanco, Rodríguez & Rodríguez, 2006: 264).

Figure 11 Material collected by the CCP.

(A–I) Neocyclotidae. Neocyclotus cumingii (Sowerby I in Broderip & Sowerby I, 1832), MNCN 15.05/76223, (A) ventral view, (B) umbilical view, (C) apical view; Neocyclotus granulatus (Pfeiffer, 1862), MNCN 15.05/21506, (D) ventral view, (E) umbilical view, (F) apical view; Neocyclotus haematomma (Pfeiffer, 1862), MNCN 15.05/20093, (G) ventral view, (H) umbilical view, (I) apical view. Scale 5 mm.

The ‘Comisión del Estudio de collecciones del Pacifico’

When in 1868 in Spain a new, liberal government came to power that wanted to promote science (López-Ocón, 1997), attention for the CCP material was renewed after it had wained following the exhibition in 1866. A ‘Comisión de Estudio de las collecciones del Pacifico’ (CEcP) was established that aimed to study the materials collected by the CCP and publish the results. At its start on 14 January 1868, the CEcP consisted of Francisco Méndez Álvaro (president), Juan Villanova y Piera (vice-president), Francisco de Paula Martínez y Sáez (secretary), Manuel Almagro, Manuel de Galdo, Joaquin Hidalgo, Marcos Jiménez de la Espada, Sandalio Pereda, José Solano y Eulate, and Lucas de Tornos. In April 1868 the Ministry of Public Instruction asked for the completion of their task as soon as possible (Blanco, Rodríguez & Rodríguez, 2006: 260). The CEcP members complained about the short notice given and argued that they needed several years to produce good scientific results. They even recommended they should travel to several European countries for their studies. The minutes of the Committee, however, show that the study of the material did not progress very smoothly [Archive MNCN CN0042/755/001]. The members of the CEcP soon discovered that they lacked the literature to identify the material, and Hidalgo made a list of desired malacological publications. The list comprised 34 books and two journal series, covering all the important malacological works since the beginning of the 19th century [Archive MNCN CN0041/749/015]. Ten of these books, however, did not or only partially reached the CEcP [Archive MNCN CN0041/749/016]. Besides the new scientific books, these minutes of the Committee showed that the CCP material had generally been split into two collections, of which one was retained for the MNCN, and one was to be sent to other schools, institutes and Museums; we have found no information about the latter. During the Spanish revolution and the abdication of Isabel II in the Autumn of 1868, Méndez Álvaro and José Solano were ousted and the former replaced by the new President M. M. J. de Galdo. In November 1868 the new Committee decided to entrust to Martínez, Jímenez de la Espada and Hidalgo the elaboration of scientific papers on the CCP material to be published in 1869. The outlines for these scientific ‘Memorias’ were accorded by the CEcP members in June 1869. In November 1869 the Commission apparently had received some money from the Ministry, which they decided to spend partly for these publications (see also Breure & Backhuys, 2017). The minutes of the CEcP end with this meeting, suggesting less organisational activities in 1870 [Archive MNCN CN0042/755/001]. During that year, again as a result of political changes, President Galdo was replaced by Lucas Tornos. By order of the Ministry (Ministerio de Instrucción Pública) the CEcP was dissolved on the 1st August 1872 providing that the director of the MNCN would be dealing with all the zoological material from the CCP. Galdo protested against this dissolution and, after another political change, the members of the Committee were re-installed by the new Government in May 1873. However, work on the scientific publications stopped and the CEcP was again dissolved on the 30th June 1875. The zoological CCP material arrived to the MNCN in 1880 (Puig-Samper, 1988: 351–352).

It is likely that Hidalgo already started working on CCP material before this time, resulting in his 1867 paper. This may explain the ‘flux’ of the material: from Paz to Hidalgo to Azpeitia’s collections; we also found some specimens in the ‘Coll. Graells’ (i.e., historical collection of MNCN) which may have been used for exhibitions over time and which may have originated from the CCP material.

The Mollusca collected by the CCP

Following Almagro (1866), who recorded the data for the exhibition of the CCP material in the Jardín Botánico, the collection of molluscs from the CCP comprised 816 different species, and 38,755 specimens, collected mainly by Paz and Martinez, and some by Jiménez de la Espada, Isern and Almagro. There were also 767 specimens belonging to 43 species of molluscs that had been be-gifted by Barreiros, Jameson, Philippi, Richardson, and Zameron. Grouped in another way, 741 specimens of marine bivalves, 300 of freshwater bivalves, 2,117 terrestrial gastropods, 1,277 freshwater gastropods and 2,557 marine gastropods were collected. There were also 975 specimens in 117 jars of alcohol preserved material, as was stipulated in the instructions for the expedition made by Graells (Puig-Samper, 1988). In 1868 and 1869 the collections of duplicate specimens were sent to several Spanish universities and institutes. In 1880 all the material collected by the CCP was moved to the MNCN (Puig-Samper, 1988). More detailed information on the localities and sources of the molluscs collected can be found in Puig-Samper (1988) and Calatayud (1994), and will be given below for the land molluscs.

All the Mollusca specimens of the CCP were studied by Martinez, who was responsible for molluscs during the expedition, and by Hidalgo (1893a, 1893b), with the exception of the freshwater bivalves that were studied by Lea (1866a, 1866b, 1867, 1869a, 1869b) and Haas; Haas, during his forced stay in Spain due to the unfavourable political climate in Germany (Haas, 1915), was invited to the MNCN where he studied the mussels collected by the CCP (Haas, 1916). Hidalgo and Martinez wrote the three volumes of the ‘Moluscos del viaje al Pacífico’, which included terrestrial gastropods (Hidalgo, 1872), marine bivalves (Martínez, 1879?), and marine gastropods (Hidalgo, 1879). There has been some confusion in the literature about the dates of publication, especially about the first part. Both the first and second parts have the date ‘1869’ printed on the title page, and this has generally been accepted by subsequent authors. The first part was published in Madrid by Cárlos Bailly-Baillière. The second and third parts bear the inscription on the title page ‘Imprenta de Miguel Ginesta’; the final volume appeared in 1879, and this date has been undisputed. As we know (Breure & Backhuys, 2017), the plates for the first and second part were executed in Paris and delivered in Madrid in November 1871. The text for the first part, however, still had to be finished by Hidalgo and this volume did not appear before December 1872. Hidalgo himself was aware of the potential problem of the discrepancy between the date on the title page (‘1869’) and the actual publication date. He inserted at the end of the text a ‘Note’ to draw attention to this discrepancy (Hidalgo, 1872: 152):

Nota. No concluida de imprimir la presente parte hasta 1872, esta es la verdadera fecha de publicación de nuestro libro y no la de 1869 que figura en la portada. Si el Gobierno de S. M. facilita medios necesarios para la impresión, ejecución de láminas, etc., del resto de la obra y si se nos indemniza del tiempo invertido en este trabajo, que hemos hecho sin sueldo ni gratificación alguna, daremos á conocer á nuestros lectores las demás especies de Moluscos recogidas por los naturalistas de la Comisión científica española.

[Note. Not having finished the print of the present part until 1872, this is the true date of publication of our book and not 1869 as contained in the cover.

If the Government will provide the necessary means for printing, execution of plates, etc., [for] the rest of the work, and if we are indemnified [for] the time invested in this work, which we have done without any payment, we will disclose to our readers the rest of the species of molluscs collected by the members of the CCP]

This note means that the book of Martinez was not yet published in 1872 and, as Breure & Backhuys (2017) have shown, the actual date of publication was much later; it was published in 1879 or even later.

Finally, it should be remarked that not only in the publications of Hidalgo, but also of others (notably Crosse), the suggestion was given that much of the CCP material was collected by Paz. This was only true in the cases where Paz actually visited the region; in other cases, after his premature return to Spain, Paz did not even visit some of the localities (e.g., Ecuador) but the material became nevertheless part of his collection or was misleadingly referred to as having been collected by him.

Labels and handwriting

The material was found with labels that allowed its provenance to be ascertained, in most cases, unambiguously. The labels from the former Collection of Paz are characterised by a red frame; their locality data is usually more general than the data which has been published for the lot. The handwriting of these labels is in Hidalgo’s hand. One clear exception is a lot where the original label in the handwriting of Paz has very specific locality data, while only a very generalised locality has been published by Hidalgo. Compare Figs. 7A–7B for examples of handwriting of Paz and Hidalgo. In most cases, the labels bear the annotation “Cat. Am. mer. no. XYZ”; this refers to the catalogue published by Hidalgo (1870), which totalled 201 species (Hidalgo, 1870), increasing to 242 species (Hidalgo, 1893a; Hidalgo, 1893b). Labels from lots collected by Martínez bear his name and generally have a more precise locality; they all formed part of the former collection of Hidalgo. However, the handwriting is written in a hand unknown to us. In the former collection of Azpeitia the labels are small and Azpeitia’s handwriting (Figs. 8G–8H) was very fine and clear. Some labels had been glued to the shells, and generally this has been maintained with the addition of a modern label. In a few cases the original labels have been lost, and all the data is from modern labels. Generally, Azpeitia copied the localities from the data published by Hidalgo. Figure 8 gives an overview of all the styles of labels associated with the CCP material.

Systematics

Remarks. The numbers between square brackets following the taxon names refer to Supplementary file 1, column ‘nr.’ available on Figshare: https://doi.org/10.6084/m9.figshare.4231904.v1. For the species described as new from the CCP material the etymology is added in the case of eponyms.

Family Helicinidae Férussac, 1822

Genus Bourciera Pfeiffer, 1852

Bourciera Pfeiffer, 1852a: 178.

Type species. Bourciera heliciniforme Pfeiffer, by monotypy.

Bourciera heliciniforme (Pfeiffer, 1853) [1]

(Figs. 9A–9C)

Cyclostoma heliciniforme Pfeiffer, 1853 [1852–1860]: 243, pl. 32 figs. 8–10; Pfeiffer, 1854b: 151.

Bourciera helicinaeformis; Hidalgo, 1870: 69; Hidalgo, 1893a: 117.

Type locality. “im Thale Yaraqui der Republik Equador”.

Type material. NHMUK 20130062 (3), probable syntypes.

Material examined. “Quito, Ecuador”, Coll. Azpeitia, MNCN 15.05/76226 (1); Coll. Paz, MNCN 15.05/13857 (3).

Remarks. Pfeiffer originally described this species from material collected by Bourcier, but his paper was not published until 1854 (Pfeiffer, 1854b: 151; cf. Duncan, 1937: 81). In his 1853 publication he erroneously referred to “Proceed. Zool. Soc. 1851” [sic, 1852]. The name was spelled in both papers as Cyclostoma heliciniforme, thus later authors have made an unjustified emendation with the spelling helicinaeformis.

Genus Helicina Lamarck, 1799

Helicina Lamarck, 1799: 76.

Type species. Helicina neritella Lamarck, 1799, by subsequent designation (Children, 1823: 239).

Helicina angulata Sowerby II, 1842 [2]

(Figs. 9D–9F)

Helicina angulata Sowerby II, 1842 [1842–1847]: 12, pl. 2 fig. 61, pl. 3 fig. 100; Hidalgo, 1870: 69; Hidalgo, 1872: 152; Hidalgo, 1893a: 118.

Type locality. “Brazil”.

Type material. Not located.

Material examined. “Macahé, Brasil”, Coll. Azpeitia ex “Martínez y Paz”, MNCN 15.05/76224 (1).

Remarks. Simone (2006) has cited this species with the erroneous year of publication ‘1873’, which has been copied by some subsequent authors.

Helicina brasiliensis Gray, 1824 [3]

(Figs. 9G–9I)

Helicina brasiliensis Gray, 1824: 66; Hidalgo, 1870: 69; Hidalgo, 1872: 150; Hidalgo, 1893a: 118; Hidalgo, 1893b: 317.

Type locality. “Brazil”.

Type material. Not located.

Material examined. “Macahé, Brasil”, Coll. Azpeitia ex “Martínez y Paz”, MNCN 15.05/39940 (3).

Remarks. Gray described this species based on material from “Mr. G.B. Sowerby”. Originally the lot contained four specimens; however, one specimen of Helicina angulata Sowerby II, 1842 appeared to have mixed in.

Helicina variabilis Wagner, 1827 [4]

(Figs. 9J–9L)

Helicina variabilis Wagner, 1827: 25; Hidalgo, 1870: 69; Hidalgo, 1893a: 117.

Type locality. [Brazil] “in Provinciae Paraënsi”.

Type material. Not located.

Material examined.“Bahia, Brasil”, Coll. Azpeitia, MNCN 15.05/39941 (2); “Rio Janeiro, Brasil”, Coll. Azpeitia, MNCN 15.05/39942 (3).

Family Neocyclotidae Kobelt & Möllendorff, 1897

The most recent review of the Cyclophorid mainland species is the work of Bartsch & Morrison (1942), who introduced many new genera and subgenera based on (often subtle) shell characteristics; provisionally we follow herein Solem (1956) who made only a distinction between Aperostoma (operculum corneous, without calcareous elements) and Neocyclotus (operculum at least partially calcareous). While this distinction may be gross and not apt for historical collections, where opercula may not have been preserved, it is here used by lack of better. It may be noted that only in a few lots of CCP material the opercula are present; in those cases they seem at least partially calcareous. The majority of the species is thus provisionally placed in Neocyclotus. Clearly, this group urgently needs a revision, preferably including molecular studies.

Genus Buckleyia Higgins, 1872

Aperostoma (Buckleyia) Higgins, 1872: 686.

Type species. “Aperostoma montezumi Hidalgo” [Cyclophorus martinezi Hidalgo, 1866; see remarks], by monotypy.

Remarks. Azpeitia (1923) listed all species described by Hidalgo, who never used the specific epithet montezumi; Higgins (1872: pl. 56 figs. 7–7a) illustrated Cyclophorus martinezi Hidalgo, 1866 when he designated the type species of his new subgenus.

Buckleyia martinezi (Hidalgo, 1866) [5]

(Figs. 10A–10B)

Cyclophorus martinezi Hidalgo, 1866a: 273, pl. 8 fig. 5; Hidalgo, 1870: 68; Hidalgo, 1893a: 34, 116; Azpeitia, 1923: 66; Baratech et al., 1993: 197, pl. 4 figs. 3a–3c.

Type locality. “Baeza Reipublicae Aequatoris”.

Type material. “Cyclophorus/Martinezi Hidalgo/tipo figurado”, Coll. Paz, MNCN 15.05/3232 (1), holotype.

Additional material examined. “Baeza (Ecuador)”, Coll. Hidalgo, MNCN 15.05/3225 (1).

Remarks. Hidalgo (1866) stated he had seen only one specimen on which his description was based. Baratech et al. (1993: 197) correctly considered it to be the holotype, although in the legend of their plate it is considered a syntype. The additional specimen that was found, probably also originates from the material collected by Martinez in March 1865, but it is herein not considered as type material.

Etymology. Named after the collector, Francisco de Paula Martinez y Sáez.

Genus Neocyclotus Crosse & P. Fischer, 1888

Neocyclotus Crosse & P. Fischer in P. Fischer & Crosse, 1888 [1880–1902]: 148.

Remarks. Authorship is herein given as published; the work was published in parts, the date of publication is after Crosnier & Clark, 1998. It may be noted that the last ‘livraison’ of this work was published posthumously in 1902, and may have been edited by H. Fischer.

Type species. Cyclostoma dysoni Pfeiffer, 1851, by subsequent designation (Pilsbry, 1910: 533).

Neocyclotus crosseanus (Hidalgo, 1866) [6]

(Fig. 10C)

Cyclophorus crosseanus Hidalgo, 1866b: 343, pl. 14 fig. 1; Hidalgo, 1870: 68; Hidalgo, 1893a: 36, 117; Azpeitia, 1923: 66; Baratech et al., 1993: 273.

Type locality. “Republica Aequatoria”.

Type material. “Ecuador”, Coll. Hidalgo, MNCN 15.05/3217 (1), MNHN (2), syntypes.

Remarks. The MNCN specimen, which was originally figured, has been affected by Byne’s disease. Baratech et al. (1993) already mentioned that moreover the syntypes in MNHN were in a bad condition.

Etymology. Named after Hippolyte Crosse.

Neocyclotus cumingii (Sowerby I in Broderip & Sowerby I, 1832) [7]

(Figs. 11A–11C)

Cyclostoma cumingii Sowerby I in Broderip & Sowerby I, 1832a: 32.

Cyclophorus cumingi; Hidalgo, 1870: 68; Hidalgo, 1893a: 116.

Type locality.“America Meridionali (Island of Tumaco)”.

Type material. Not located.

Material examined. “Quito, Ecuador”, Coll. Azpeitia, MNCN 15.05/76223 (1).

Neocyclotus giganteus (Sowerby I in Reeve, 1842) [8]

(Figs. 10D–10F)

Cyclostoma giganteum ‘Gray’ Sowerby I in Reeve, 1842: 99, pl. 184 fig. 17.

Cyclotus fischeri Hidalgo, 1867: 305, pl. 8 fig. 3; Hidalgo, 1870: 67; Hidalgo, 1872: 144, pl. 8 figs. 9–11; Hidalgo, 1875: 129; Hidalgo, 1893a: 115; Hidalgo, 1893b: 310; Azpeitia, 1923: 82; Fischer-Piette, 1950: 69; Baratech et al., 1993: 273.

Type locality. Not given.

Type material. Not located.

Additional type material. “Cyclotus Fischeri/Hidalgo/type/J. Conchyl. 1867, Juillet”, Coll. Paz, MNCN 15.05/17560 (1); “Quito”, Coll. Hidalgo ex Paz, MNCN 15.05/3261 (3); MNHN (1), syntypes of Cyclotus fischeri Hidalgo, 1867.

Additional material examined. “Aguarico (Ecuador)”, Coll. Hidalgo ex Martínez leg., MNCN 15.05/3262 (4); “Quito, Ecuador”, Coll. Azpeitia, MNCN 15.05/3305 (1); “Quito y Aguarico”, Coll. Azpeitia ex Isern leg., MNCN 15.05/76215 (1); “Pacifico 186”, Coll. Hidalgo, MNCN 15.05/20009 (1).

Remarks. The species was figured on the basis of ‘Gray MSS in Brit. Mus.’. The material of Martínez was collected between 17–25 July 1865 near the Aguarico river (Calatayud, 1994: 243–244) in Dept. Orellana on the border near Peru. Hidalgo (1872: 152), in his errata, attributed his Cyclotus fischeri to ‘C. giganteus Gray’; this author, however, never made this name available. According to Baratech et al. (1993) the specimen in the MNHN collection could be part of the original series, a statement with which we concur.

Etymology. Named after Paul Fischer (1835–1893), who Hidalgo has met during his first visit to Paris (Breure & Backhuys, 2017).

Neocyclotus granulatus (Pfeiffer, 1862) [9]

(Figs. 11D–11F)

Cyclotis granulatus Pfeiffer, 1862: 275; Hidalgo, 1870: 67; Hidalgo, 1893a: 116.

Type locality.“Ecuador”.

Type material. NHMUK 20160364 (3), syntypes.

Material examined. “Quito”, Coll. Hidalgo ex Paz leg., MNCN 15.05/21506 (2); “Quito, Ecuador”, Coll. Azpeitia, MNCN 15.05/76222 (3).

Neocyclotus haematomma (Pfeiffer, 1862) [10]

(Figs. 11G–11I)

Cyclophorus haematomma Pfeiffer, 1862: 276; Hidalgo, 1870: 68; Hidalgo, 1893a: 117.

Type locality. “Ecuador”.

Type material. NHMUK 2016065 (3), syntypes.

Material examined.“196”, Coll. Hidalgo, MNCN 15.05/20093 (1); “Quito, Ecuador”, Coll. Azpeitia, MNCN 15.05/76231 (1).

Remarks. Hidalgo (1870) listed this species as number 196 in his catalogue, stating it had been collected in “Quito (Paz)”.

Neocyclotus hidalgoi (Crosse, 1866) [11]

Cyclophorus hidalgoi Crosse, 1866: 354, pl. 14, fig. 4; Hidalgo, 1870: 66; Hidalgo, 1893a: 116.

Type locality. “Republica Aequatoris”.

Type material. Not located.

Remarks. This species was described by Crosse based on material from “Coll. Hidalgo”. However, no material could be traced, nor in the MNCN nor in the MNHN collections.

Etymology. Named after Joaquin Hidalgo.

Neocyclotus pazi (Crosse, 1866) [12]

(Figs. 12A–12C)

Figure 12 Material collected by the CCP.

(A–I) Neocyclotidae. Neocyclotus pazi (Crosse, 1866), MNCN 15.05/21591, (A) ventral view, (B) umbilical view, (C) apical view; Neocyclotus perezi (Hidalgo, 1866), MNCN 15.05/3264, (D) ventral view, (E) umbilical view, (F) apical view; Neocyclotus prominulus (d’Orbigny, 1837), MNCN 15.05/39927, (G) ventral view, (H) umbilical view, (I) apical view; Neocyclotus quitensis (Pfeiffer, 1854), MNCN 15.05/76212, (J) ventral view, (K) umbilical view, (L) apical view. Scale 5 mm.

Cyclotus pazi Crosse, 1866: 356, pl. 14, fig. 3; Hidalgo, 1870: 67; Hidalgo, 1872: 148, pl. 8 figs. 14–15; Hidalgo, 1893a: 116; Hidalgo, 1893b: 314.

Type locality. “Ambato, Reipublicae Aequatoris”.

Type material.“Ambato, Ecuador”, Coll. Hidalgo, MNCN 15.05/21591 (25), syntypes.

Remarks. Crosse (1866) stated “Coll. Paz, Hidalgo, et Crosse”, therefore the material in the MNCN is considered as syntypes.

Etymology. Named after Patricio Paz y Membiela.

Neocyclotus perezi (Hidalgo, 1866) [13]

(Figs. 12D–12F)

Cyclotus perezi Hidalgo, 1866b: 344, pl. 14, fig. 2; Hidalgo, 1872: 147, pl. 8, figs. 12–13; Hidalgo, 1893a: 38; Azpeitia, 1923: 82; Calvo, 1994: 283.

Type locality. “Baeza, Reipublicae Aequatoris”.

Type material. “Baeza (Ecuador)”, Coll. Hidalgo ex Martínez leg., MNCN 15.05/3264 (15); “Baeza (Ecuador)”, “Pacifico 188”, Coll. Hidalgo, MNCN 15.05/3263 (15), syntypes.

Additional material examined. “Ecuador”, Coll. Hidalgo, MNCN 15.05/3265 (2); “Baeza”, Coll. Azpeitia, MNCN 15.05/76204 (25); “Ecuador”, Coll. Hidalgo, MNCN 15.05/76204 (576).

Remarks. The material was collected by Martinez in March 1865 (Calatayud, 1994: 229).

Etymology. Named after Laureano Pérez Arcas (1824–1894), director of the MNCN from 1868 to 1870; he was befriended with Hidalgo (Breure & Backhuys, 2017).

Neocyclotus prominulus ( d’Orbigny, 1837) [14]

(Figs. 12G–12I)

Cyclostoma prominula ‘Férussac’ d’Orbigny, 1837 [ 1834–1847]: 362.

Cyclotus prominulus; Hidalgo, 1870: 68; Hidalgo, 1893a: 116; Hidalgo, 1893b: 315.

Type locality. “la province des Mines, au Brésil”.

Type material. MNHN, probable syntypes (Simone, 2006: 42, fig. 39).

Material examined. “Río de Janeiro (Brasil)”, Coll. Azpeitia, MNCN 15.05/39927 (3).

Remarks. This species was described by d’Orbigny on the basis of material presented to him in Rio de Janeiro, using a name from the Coll. Férussac. Simone (2006: 42) cited this species with the wrong year of publication.

Neocyclotus quitensis (Pfeiffer, 1854) [15]

(Figs. 12J–12L)

Cyclostoma (Cyclotus) quitense Pfeiffer, 1854a: 61.

Cyclotus quitensis; Hidalgo, 1870: 67; Hidalgo, 1872: 146; Hidalgo, 1893a: 115; Hidalgo, 1893b: 312.

Type locality. “Quito”.

Type material. NHMUK 20160366 (3), syntypes.

Material examined. “Quito”, Coll. Azpeitia, MNCN 15.05/76212 (1).

Remarks. This species, originally described from the Cuming collection, was mentioned by Hidalgo (1870) from “Quito (Paz), Napo (Martinez)”. The latter material has not been located.

Family Succineidae Beck, 1837

Genus Omalonyx d’Orbigny, 1837

Omalonyx d’Orbigny 1837 [ 1834–1847]: 229.

Type species. Helix (Cochlodina) unguis d’Orbigny, 1835, by monotypy.

Omalonyx cf. unguis (d’Orbigny, 1835) [16]

(Fig. 13A)

Figure 13 Material collected by the CCP.

(A–C) Succineidae. Omalonyx cf. unguis (d’Orbigny, 1835), MNCN 15.05/12096, (A) ventral view; Succinea donneti Pfeiffer, 1853, MNCN 15.05/76203, (B) ventral view; Succinea peruviana (Philippi in Pfeiffer, 1867), MNCN 15.05/76208, (C) ventral view. (D) Pupillidae. Pupoides paredesii (d’Orbigny, 1835), MNCN 15.05/14914, ventral view. (E–F) Vertiginidae. Gastrocopta oblonga (Pfeiffer, 1854), MNCN 15.05/39925, (E) ventral view; Gastrocopta pazi (Hidalgo, 1869), MNCN 15.05/3285, (F) ventral view. (G) Clausiliidae. Peruinia peruana (Troschel, 1847), MNCN 15.05/37075, ventral view. Scale line 0.5 mm (E), 1 mm (D, F), 5 mm (all others).

Helix unguis d’Orbigny, 1835: 2 [nomen nudum].

Succinea (Omalonyx) unguis d’Orbigny 1835 [ 1834–1847]: pl. 22 figs. 1–7; d’Orbigny 1837 [ 1834–1847]: 229.

Omalonyx unguis; Hidalgo, 1870: 30; Hidalgo, 1872: 7; Hidalgo, 1893a: 78; Hidalgo, 1893b: 309.

Type locality. “les bords inondés du Parana, près de Corrientes (…) les marais de la province de Moxos, république de Bolivia”.

Type material. Not located.

Material examined.“Bahia”, Coll. Paz, MNCN 15.05/12096 (3); “Bahia”, Coll. Hidalgo, MNCN 15.05/15770 (3).

Remarks. Helix unguis ‘Fer.’ was mentioned only by d’Orbigny (1835: 2), without description or reference; it is a nomen nudum. The figures in d’Orbigny (1834–1847) were published in the same year (1835), but the text only in 1837, allowing to make reference to Moricand (1836) who had recognized the species in material from Bahia; the CCP material originates from the same region and was probably collected during September 1862 (Calatayud, 1994: 249).

Genus Succinea Draparnaud, 1801

Succinea Draparnaud, 1801: 32.

Type species. Helix putris Linnaeus, 1758, by subsequent designation (Gray, 1847: 171).

Succinea donneti Pfeiffer, 1853 [17]

(Fig. 13B)

Succinea donneti Pfeiffer, 1853: 19; Hidalgo, 1870: 30; Hidalgo, 1872: 6, pl. 2 figs. 16–17; Hidalgo, 1875: 127; Hidalgo, 1893a: 78; Hidalgo, 1893b: 308.

Type locality. [Chile] “prope Coquimbo”.

Type material. NHMUK 20160368 (3), syntypes.

Material examined.“P 4”, [Coll. Hidalgo], MNCN 15.05/76203 (3).

Remarks. This lot was found without label except a species label; however, similar lots had been found which proved to originate from Hidalgo’s collection. Moreover, the indication “P 4” provided a link to Hidalgo (1870), who lists this species from “Coquimbo, Chili (Paz); Chunchuco, Chili (Martínez)”.

Succinea peruviana (Philippi in Pfeiffer, 1867) [18]

(Fig. 13C)

Succinea peruviana Philippi in Pfeiffer, 1867: 78; Hidalgo, 1870: 30; Hidalgo, 1875: 127, pl. 7 fig. 1; Hidalgo, 1893a: 78.

Type locality. “Peruvia”.

Type material. Not located.

Material examined.“P 3”, [Coll. Hidalgo], MNCN 15.05/76208 (9).

Remarks. This lot was found without label except a species label; however, similar lots had been found which proved to originate from Hidalgo’s collection. Moreover, the indication “P 3” provided a link to Hidalgo (1870), who stated the material to be collected by Paz at “Lomas de Pumará, Amancaez et Cerro de las Conchitas, environs de Lima”; the collecting date was mid-July 1863 (Calatayud, 1994: 258).

Family Pupillidae Turton, 1831

Genus Pupoides Pfeiffer, 1854

Bulimus (Pupoides) Pfeiffer, 1854c: 192.

Type species. Bulimus nitidulus Pfeiffer, 1839, by subsequent designation (Kobelt, 1902 [1899–1902]: 917).

Pupoides paredesii (d’Orbigny, 1835) [19]

(Fig. 13D)

Helix paredesii d’Orbigny, 1835: 21.

Pupa paredesii; Hidalgo, 1870: 65; Hidalgo, 1893a: 114.

Type locality. “provincia Pazensi (republica Boliviana); provincia Limacensi (republica Peruviana)”.

Type material. NHMUK 1854.12.4.236–237 (11), syntypes.

Material examined. “Lima”, Coll. Paz, MNCN 15.05/14845 (47), MNCN 15.05/14914 (46).

Remarks. Hidalgo (1870) quoted this species from “Lima, Pérou; Guayaquil, Equateur; Cobija, Bolivia (Paz)”. Material of the last two localities has not been found.

Family Vertiginidae Fitzinger, 1833

Genus Gastrocopta Wollaston, 1878

Gastrocopta Wollaston, 1878: 515.

Type species. Pupa acarus Benson, 1856, by subsequent designation (Pilsbry, 1916 [1916–1918]: 7).

Gastrocopta oblonga (Pfeiffer, 1854) [20]

(Fig. 13E)

Pupa oblonga Pfeiffer, 1854a: 69; Hidalgo, 1870: 65; Hidalgo, 1872: 141; Hidalgo, 1893a: 114.

Type locality. “—?”.

Type material. NHMUK 20160367 (2), syntypes.

Material examined. “Bahia”, Coll. Hidalgo ex Paz leg., MNCN 15.05/39925 (5); “Sta. Lucia Montevo.”, Coll. Hidalgo, MNCN 15.05/76233 (42).

Gastrocopta pazi (Hidalgo, 1869) [21]

(Fig. 13F)

Pupa pazi Hidalgo, 1869c: 412; Hidalgo, 1870: 66; Hidalgo, 1875: 129, pl. 7 fig. 7; Hidalgo, 1893a: 58, 114.

Type locality. “Amancaez, republica Peruvian; Guayaquil, republica Aequatoris; Panama (Paz)”.

Type material. “Amancaez”, Coll. Hidalgo, MNCN 15.05/3284 (13); “Amancaez, cerca de Lima”, Coll. Azpeitia, MNCN 15.05/3285 (7); “Amancaez”, Coll. Azpeitia, MNCN 15.05/3286 (1); “Guayaquil”, Coll. Hidalgo, MNCN 15.05/3281 (46), syntypes.

Remarks. All material has no original labels from Paz. The specimens from Panama could not be located.

Etymology. Named after Patricio Paz y Membiela.

Family Clausiliidae Gray, 1855

Genus Incania Poliński, 1922

Nenia (Incania) Poliński, 1922: 125.

Type species. Clausilia chacaensis Lubomirski, 1880, by subsequent designation (Pilsbry, 1926: 10).

Incania crossei (Hidalgo, 1869) [22]

Clausilia crossei Hidalgo, 1869c: 413; Hidalgo, 1870: 66, pl. 6 fig. 9.

Type locality. “Baeza, Equateur”.

Remarks. This species was based on material collected by Martinez. Baratech et al. (1993: 285) listed this species already as one of which the type material could not be located in the MNCN collection.

Etymology. Named after Hippolyte Crosse.

Genus Peruinia Poliński, 1922

Nenia (Peruinia) Poliński, 1922: 125.

Type species. Clausilia peruana Troschel, 1847, by subsequent designation (Pilsbry, 1926: 10).

Peruinia peruana (Troschel, 1847) [23]

(Fig. 13G)

Clausilia peruana Troschel, 1847: 51; Hidalgo, 1870: 66; Hidalgo, 1893a: 115.

Type locality. “Peru”.

Type material. Not located.

Material examined. “Chanchamayo”, Coll. Hidalgo ex Isern leg., MNCN 15.05/37075 (4), MNCN 15.05/37083 (4); MNCN 15.05/18308 (185, in ethanol).

Family Amphibulimidae P. Fischer, 1874

Genus Plekocheilus Guilding, 1828

Plekocheilus Guilding, 1828: 532.

Type species. Caprella undulata Guilding, 1824, by monotypy.

SubgenusPlekocheilus (Eurytus) Albers, 1850

Eurytus Albers, 1850: 169.

Type species. Helix pentadina d’Orbigny, 1835, by subsequent designation (Albers, 1860: 195).

Plekocheilus (Eurytus) aristaceus (Crosse, 1869) [24]

(Fig. 14A)

Figure 14 Material collected by the CCP.

(A–E) Amphibulimidae. Plekocheilus (Eurytus) aristaceus (Crosse, 1869), MNCN 15.05/13475, (A) ventral view; Plekocheilus (Eurytus) cardinalis (Pfeiffer, 1853), MNCN 15.05/13705, (B) ventral view; Plekocheilus (Eurytus) floccosus (Spix in Wagner, 1827), MNCN 15.05/76205, (C) ventral view; Plekocheilus (Eurytus) jimenezi ( Hidalgo, 1872), MNCN 15.05/3158, (D) ventral view, (E) dorsal view. Scale line 5 mm (A, B), 1 cm (C–E).

Bulimus aristaceus Crosse, 1869: 185; Crosse, 1870: 105, pl. 6 fig. 5; Hidalgo, 1870: 54, pl. 6 fig. 5; Hidalgo, 1893a: 102.

Plekocheilus (Eurytus) aristaceus; Breure & Araujo, 2015: 87, fig. 1; Breure & Mogollón, 2016: 14, figs. 8A–8C, 14.

Type locality. “Quito, reipublicae Aequatoris”.

Type material.“Quito, Ecuador”, MNCN 15.05/7180, lectotype (Breure & Araujo, 2015: 87); “Ecuador”, “(Cat. Am. mer. no. 125)”, Coll. Paz, MNCN 15.05/13475 (1), paralectotype.

Remarks. Crosse (1869) stated “(Paz)”, making the impression this material was collected by Paz. However, since Paz did not visit Ecuador with the CCP, this material must have been collected by one of the other members. Since the publication of Breure & Araujo (2015) designating the lectotype, we have found now an additional specimen among the CCP material. This specimen has a damaged last whorl, which has slightly influenced the shape of the aperture; its is lighter in colour but otherwise matches the lectotype.

Plekocheilus (Eurytus) cardinalis (Pfeiffer, 1853) [25]

(Fig. 14B)

Bulimus cardinalis Pfeiffer, 1853: 316; Hidalgo, 1870: 55; Hidalgo, 1872: 92; Hidalgo, 1893a: 102; Hidalgo, 1893b: 219.

Type locality. “Quito”.

Type material. ZMB 112721 (1), syntype.

Material examined. “Quito”, “(Cat. Am. mer. no. 126)”, Coll. Paz, MNCN 15.05/13705 (2); “Napo (Ecuador)”, “Pacifico 126”, Coll. Hidalgo ex Martínez y Saez leg., MNCN 15.05/36846 (2).

Remarks. Hidalgo (1870) mentioned two localities “Environs de Quito (Paz); Napo, Equateur (Martínez)”; in his 1872 publication only the latter locality was mentioned. Compared to the syntype of this species (Borrero & Breure, 2011: figs. 15E–15F), the specimens from the CCP have a more thickened peristome and parietal callus.

Plekocheilus (Eurytus) floccosus (Spix in Wagner, 1827) [26]

(Fig. 14C)

Achatina floccosa Spix in Wagner, 1827: 10, pl. 9 figs. 3–4.

Bulimus floccosus; Hidalgo, 1870: 61; Hidalgo, 1872: 127, pl. 7 figs. 1–4; Hidalgo, 1893a: 110; Hidalgo, 1893b: 215.

Type locality. “sylvis Provinciarum septemtrionalium Brasiliae”.

Type material. ZSM 20020116 (1), syntype (Breure & Mogollón, 2016: figs. 3C–3D).

Material examined.“Ecuador”, “(Cat. Am. mer. no. 165)”, Coll. Paz, MNCN 15.05/13285 (2); “165 Pacifico”, Coll. Hidalgo, MNCN 15.05/76205 (1).

Remarks. The locality was given as “Napo, Équateur (Martínez)” (Hidalgo, 1870). Hidalgo said he had seen only three specimens, two not full-grown from his own collection and from the collection of Paz, and an adult specimen from the MNCN. However, we found two shells originating from the Coll. Paz. The shell from Hidalgo’s own collection is MNCN 15.05/76205, which corresponds to Hidalgo, 1872: pl. 7 figs. 3–4.

Plekocheilus (Eurytus) jimenezi (Hidalgo, 1872) [27]

(Figs. 14D–14E)

Bulimus gibbonius Hidalgo, 1870: 54; Hidalgo, 1875: 128. Not Bulimus gibbonius Lea, 1838.

Bulimus jimenezi Hidalgo, 1872: 93, 152, pl. 5 figs. 2–3; Hidalgo, 1893a: 68, 102; Hidalgo, 1893b: 217; Azpeitia, 1923: 58; Baratech et al., 1993: 215.

Plekocheilus (Eurytus) jimenezi; Borrero & Breure, 2011: 43, figs. 13B–13D; Breure & Mogollón, 2016: 17, figs. 10C–10F, 14.

Type locality. [Ecuador] “San José”.

Type material.“San José (Ecuador)”, Isern & Jimenez de Espada leg., MNCN 15.05/1066 (2); “Napo, Ecuador”, “(Cat. Am. mer. no. 122)”, Coll. Paz, MNCN 15.05/3158 (2), syntypes.

Additional material examined.“Ecuador”, Coll. Graells, MNCN 15.05/3307 (1).

Remarks. Hidalgo has written on the label of MNCN 15.05/1066 “uno de los exemplars figurado”. Breure & Mogollón (2016: 18) have suggested that “San José” would be San José de Suno. The itinerary of Isern and Jimenez de Espada (Calatayud, 1994: 278) only mentions San José de Monti; this locality cannot be traced with modern gazetteers, but it is likely in the same general region.

Figure 15 Material collected by the CCP.

(A–D) Amphibulimidae. Plekocheilus (Eurytus) lynciculus (Deville & Hupé, 1850), MNCN 15.05/13389, (A) ventral view; Plekocheilus (Eurytus) taylorianus (Reeve, 1849), MNCN 15.05/13706, (B) ventral view; Plekocheilus (Eurytus) tricolor (Pfeiffer, 1853), MNCN 15.05/6943, (C) ventral view; Plekocheilus (Plekocheilus) cecepeus Breure & Araujo, 2015, MNCN 15.05/60013H, (D) ventral view. Scale line 0.5 mm.

Etymology. Named after Marcos Jiménez de la Espada.

Plekocheilus (Eurytus) lynciculus (Deville & Hupé, 1850) [28]

(Fig. 15A)

Bulimus lynciculus Deville & Hupé, 1850: 640, pl. 15 fig. 1; Hidalgo, 1870: 54; Hidalgo, 1872: 94; Hidalgo, 1893a: 102.

Type locality. “Mission de Sarayacu, sur les bords de la rivière de l’Ucuyali, Pérou”.

Type material. Not located.

Material examined. “Napo, Ecuador”, “(Cat. Am. mer. no. 124)”, Coll. Paz, MNCN 15.05/13389 (2); “Pacifico 124”, Coll. Hidalgo, MNCN 15.05/21312 (1).

Remarks. Of the three specimens the one figured herein seems to have been collected rather fresh and, although the peristome is unexpanded, seems to exhibit the features of this species the best. One specimen was found with locality data “Napo (Ecuador)”, Coll. Hidalgo ex Martínez, MNCN 15.05/7214, identified as this species, which appeared to be a specimen of Plekocheilus (Eudolichotis) distorta (Bruguière, 1792). This was likely not material collected by the CCP, as this species occurs in northern Venezuela; this region was not visited by the CCP.

Plekocheilus (Eurytus) taylorianus (Reeve, 1849) [29]

(Fig. 15B)

Bulimus taylorianus Reeve, 1849 [1848–1850]: pl. 81 fig. 602; Hidalgo, 1870: 54; Hidalgo, 1893a: 102.

Type locality. [Ecuador] “Environs of Quito”.

Type material. NHMUK 1874.12.11.271, lectotype (Breure, 1978: 16).

Material examined. “Quito”, “(Cat. Am. mer. no. 123)”, Coll. Paz, MNCN 15.05/13706 (2); “Pacifico 123”, Coll. Paz, MNCN 15.05/36941 (3); “Quito, Ecuador”, Coll. Azpeitia, MNCN 15.05/7351 (2).

Remarks. Hidalgo (1870) wrote “Quito (Paz et Martínez)”; it is possible that the Azpeitia shells were originally collected by Martínez.

Plekocheilus (Eurytus) tricolor (Pfeiffer, 1853) [30]

(Fig. 15C)

Bulimus tricolor Pfeiffer, 1853: 325.

Bulimus semipictus Hidalgo, 1869a: 188; Hidalgo, 1870: 56, pl. 6 fig. 7; Hidalgo, 1872: 95, pl. 6 figs. 8–9; Hidalgo, 1893a: 49, 104; Hidalgo, 1893b: 217; Azpeitia, 1923: 58; Fischer-Piette, 1950: 72; Baratech et al., 1993: 216.

Plekocheilus (Eurytus) tricolor; Breure & Mogollón, 2016: 24, figs. 2K–2M, 13C–13D, 16.

Type locality.“Gualea, Neu Granada”.

Type material. Not located.

Additional type material examined. MHNH-IM-2000-28113, lectotype of Bulimus semipictus Hidalgo (Fischer-Piette, 1950: 72); “Baeza, Ecuador”, “(Cat. Am. mer. no. 138)”, Coll. Paz, MNCN 15.05/76217 (2); “Baeza (Ecuador)”, Coll. Hidalgo ex Martínez y Saez leg., MNCN 15.05/6943 (6), MNCN 15.05/3209 (1); “Baeza, Ecuador”, Coll. Azpeitia, MNCN 15.05/76229 (2), paralectotypes of Bulimus semipictus Hidalgo.

Subgenus Plekocheilus s.str.

Plekocheilus (Plekocheilus) cecepeus Breure & Araujo, 2015 [31]

(Fig. 15D)

Plekocheilus (Plekocheilus) cecepeus Breure & Araujo, 2015: 89, fig. 2; Breure & Mogollón, 2016: 25, figs. 8D–8F.

Type locality. “Ecuador, Quito”.

Type material. “Quito”, MNCN 15.05/60013H, holotype; MNCN 15.05/60013P (5), MNCN 15.05/7477P (3), paratypes.

Etymology. Named after the CCP members collectively.

Family Megaspiridae Pilsbry, 1904

Genus Megaspira Jay, 1836

Megaspira Jay, 1836: 39.

Type species. Megaspira ruschenbergiana Jay, 1836, by monotypy.

Megaspira elatior (Spix in Wagner, 1827) [32]

(Fig. 16A)

Figure 16 Material collected by the CCP.

(A–F) Megaspiridae. Megaspira elatior (Spix in Wagner, 1827), MNCN 15.05/19283, (A) ventral view; Thaumastus (Thaumastiella) cf. koepckei Zilch, 1953, MNCN 15.05/13501, (B) ventral view; Thaumastus (Thaumastus) achilles (Pfeiffer, 1853), MNCN 15.05/13299, (C) ventral view; Thaumastus (Thaumastus) cf. orcesi Weyrauch, 1967, MNCN 15.05/7567, (D) ventral view; Thaumastus (Thaumastus) foveolatus (Reeve, 1849), MNCN 15.05/13497, (E) ventral view; Thaumastus (Thaumastus) hartwegi (Pfeiffer in Philippi, 1846), MNCN 15.05/13507, (F) ventral view. Scale line 1 cm.

Pupa elatior Spix in Wagner, 1827: 20.

Megaspira elatior; Hidalgo, 1870: 66; Hidalgo, 1893a: 114.

Type locality. [Brazil] “cum praecedentibus [in Provinciis mediis orientalibus]”.

Type material. Not located.

Material examined.“Rio Janeiro”, “comprado”, Coll. Hidalgo ex [Paz or Martínez y Saez], MNCN 15.05/19283 (1), MNCN 15.05/19285 (2); “Rio Janeiro, Brasil”, Coll. Azpeitia, MNCN 15.05/39943 (2).

GenusThaumastus Martens in Albers, 1860

Bulimulus (Thaumastus) Martens in Albers, 1860: 215.

Type species. Bulimus hartwegi Pfeiffer in Philippi, 1846, by original designation.

SubgenusThaumastus (Thaumastiella) Weyrauch, 1956

Thaumastus (Thaumastiella) Weyrauch, 1956: 11.

Type species. Bulimulus sarcochrous Pilsbry, 1897, by original designation.

Thaumastus (Thaumastiella) cf. koepckei Zilch, 1953 [33]

(Fig. 16B)

Thaumastus (Scholvienia) koepckei Zilch, 1953: 53, figs. 7–9, pl. 14 fig. 3.

Bulimus porphyreus [sic] Pfeiffer; Hidalgo, 1870: 45; Hidalgo, 1872: 65; Hidalgo, 1893a: 91 [all partim].

Type locality. “Peru Hacienda Monteseco”.

Type material. SMF 111487, holotype.

Material examined. “Peru”, “(Cat. Am. mer. no. 69)”, Coll. Paz, MNCN 15.05/13501 (2).

Remarks. These specimens had been identified as Bulimus porphyrius Pfeiffer, 1847, but they are missing both the characteristic white, peripheral girdle, and the rudely wrinkled sculpture on the last whorls (Breure & Ablett, 2015: fig. 11iv). Instead, the shell shape and colouration reminds us of Thaumastus (Thaumastiella) species and we tentatively identify this material as T. (T.) koepckei Zilch, 1953.

Subgenus Thaumastus s. str.

Thaumastus (Thaumastus) achilles (Pfeiffer, 1853) [34]

(Fig. 16C)

Bulimus achilles Pfeiffer, 1853: 378.

Bulimus thompsoni [sic] Pfeiffer; Hidalgo, 1870: 45; Hidalgo, 1893a: 91 [all partim].

Type locality.[Brazil] “in ripis fluvii Amazonum”.

Type material. NHMUK 1975286, lectotype (Breure, 1978: 32).

Material examined. “Machahé”, “(Cat. Am. mer. no. 68)”, Coll. Paz, MNCN 15.05/13299 (2).

Remarks. These species had been misidentified as “Bulimus thompsoni Pfr”, possibly because the locality was misinterpreted as Ecuadorian, while it is actually in Brazil.

Thaumastus (Thaumastus) cf. orcesi Weyrauch, 1967 [35]

(Fig. 16D)

Thaumastus (Thaumastus) orcesi Weyrauch, 1967: 473, fig. 2.

Type locality. “Ecuador, cuenca del río Esmeraldas, 35 km al noroeste de Quito, region de Nanegal, 1,500 m”.

Type material. IFML-MOLL 3165, holotype (Breure, 2012: pl. 6 figs. 59–61).

Material examined. “Loja, Equateur”, Coll. Hidalgo, MNCN 15.05/7567 (1).

Remarks. This material was found undetermined in the Hidalgo collection, but has an original label in the handwriting of Paz; it is tentatively regarded as CCP material. The specimen is very similar to Weyrauch’s species, but was found at a disjunct locality.

Thaumastus (Thaumastus) foveolatus (Reeve, 1849) [36]

(Fig. 16E)

Bulimus foveolatus Reeve, 1849 [1848–1850]: pl. 73 fig. 526; Hidalgo, 1870: 45; Hidalgo, 1872: 56, pl. 6 figs. 4–5; Hidalgo, 1893a: 92; Hidalgo, 1893b: 203.

Type locality. “Vitoe, near Sarma [sic, Tarma], Alto-Peru”.

Type material. NHMUK 1975275, lectotype (Breure, 1979: 44).

Material examined.“Chanchamayo, Peru”, “(Cat. Am. mer. no. 71)”, Coll. Paz, MNCN 15.05/3496 (2); “Chanchamayo, Peru”, Coll. Paz, MNCN 15.05/13497 (1); “Chanchamayo (Perú)”, Coll. Hidalgo ex Isern leg., MNCN 15.05/36922 (4); “Pacifico 71”, Coll. Hidalgo, MNCN 15.05/36921 (2).

Remarks. The lot with the single specimen corresponds to the one which Hidalgo mentioned to have spiral lines on the last whorl. This is caused by a shell repair at the beginning of the last whorl. All material was collected by Isern, who was the only CCP member to visit the Chanchamayo region in autumn 1863 (Calatayud, 1994: 257).

Thaumastus (Thaumastus) hartwegi (Pfeiffer in Philippi, 1846) [37]

(Fig. 16F)

Bulimus hartwegi Pfeiffer in Philippi, 1846 [1845–1847]: 111, pl. 4 fig. 1; Hidalgo, 1870: 44; Hidalgo, 1872: 64, pl. 4 figs. 4–5; Hidalgo, 1893a: 91; Hidalgo, 1893b: 241.

Type locality.“respublica [sic] Aequatoris, ubi ad ‘El Catamaija’ prope Loxa”.

Type material. NHMUK 1975126 (1), syntype.

Material examined. “Ecuador”, “(Cat. Am. mer. no. 67)”, Coll. Paz, MNCN 15.05/13507 (2); “Pacifico 67”, Coll. Hidalgo, MNCN 15.05/36945 (1); “Cuenca (Ecuador)”, Coll. Hidalgo ex Jameson, MNCN 15.05/36942 (1); “Cuenca (Ecuador)”, Coll. Azpeitia, MNCN 15.05/14296 (1).

Remarks. The material was mentioned as “Hab. Quito et Cuenca, Équateur (Paz)” by Hidalgo (1870); it agrees with the variation observed in this taxon. The specimen from lot MNCN 15.05/36945 was figured in Hidalgo, 1872: pl. 4 figs. 4–5.

Thaumastus (Thaumastus) largillierti (Philippi, 1845) [38]

(Fig. 17A)

Figure 17 Material collected by the CCP.

(A–C) Megaspiridae. Thaumastus (Thaumastus) largillierti (Philippi, 1845), MNCN 15.05/8096, (A) ventral view; Thaumastus (Thaumastus) magnificus (Grateloup, 1839), MNCN 15.05/13704, (B) ventral view; Thaumastus (Thaumastus) taunaisii (Férussac, 1822), MNCN 15.05/36932, (C) ventral view. Scale line 1 cm.

Bulimus largillierti Philippi, 1845 [1845–1847]: 11, pl. 3 fig. 6.

Bulimus taunaisii Férussac; Hidalgo, 1870: 45; Hidalgo, 1872: 66; Hidalgo, 1893a: 91; Hidalgo, 1893b: 204 [all partim].

Type locality. “Brasilien, Santa Catarina”.

Type material. Not located.

Material examined.“Brasil”, Coll. Azpeitia, MNCN 15.05/8096 (1).

Remarks. Hidalgo misidentified this species as Bulimus taunaisii Férussac; he (Hidalgo, 1893b: 207) mentioned that this species was collected at “Santa Catalina”.

Thaumastus (Thaumastus) magnificus (Grateloup, 1839) [39]

(Fig. 17B)

Bulimus magnificus Grateloup, 1839: 165, pl. 4 fig. 1; Hidalgo, 1893a: 124.

Type locality. “Pérou”.

Type material. NHMUK 1907.11.22.24, lectotype (Breure, 1978: 31).

Material examined. “Rio Janeiro”, “(Cat. Am. mer. no. […])”, Coll. Paz, MNCN 15.05/13704 (2); “Pacifico 229”, Coll. Hidalgo, MNCN 15.05/36934 (1); “Brasil”, Coll. Azpeitia, MNCN 15.05/7327 (3).

Remarks. This species, of which the lectotype was recently re-figured by Breure & Mogollón, 2016: figs. 27C–27E, is likely restricted to eastern Brazil.

Thaumastus (Thaumastus) taunaisii (Férussac, 1822) [40]

(Fig. 17C)

Helix (Cochlostyla) taunaisii Férussac, 1822 [ 1821–1822]: 48.

Bulimus taunaisii; Hidalgo, 1870: 45; Hidalgo, 1872: 66; Hidalgo, 1893a: 91; Hidalgo, 1893b: 204 [all partim].

Type locality. [Brazil] “in ripis fluvii Amazonum”.

Type material. Not located.

Material examined.“Rio Janeiro”, “(Cat. Am. mer. no. 70)”, Coll. Paz, MNCN 15.05/13288 (2); “Rio Janeiro (Brasil)”, “Pacifico 70”, Coll. Hidalgo ex “Paz y Martínez” leg., MNCN 15.05/36932 (8); “Macahé (Brasil)”, Coll. Hidalgo ex Martínez leg., MNCN 15.05/21565 (4); “Rio Janeiro, Brazil”, Coll. Azpeitia, MNCN 15.05/7349 (1).

Family Orthalicidae Martens in Albers, 1860

Genus Clathrorthalicus Strebel, 1909

Orthalicus (Clathrorthalicus) Strebel, 1909: 150.

Type species.Orthalicus wallisi Strebel, 1909, by original designation (Strebel, 1909: 102).

Clathrorthalicus corydon (Crosse, 1869) [41]

(Fig. 18A)

Figure 18 Material collected by the CCP.

(A–G) Orthalicidae. Clathrorthalicus corydon (Crosse, 1869), MNCN 15.05/8077, (A) ventral view; MNCN 15.05/21868, (B) ventral view, (C) lateral view, (D) dorsal view; Corona pfeifferi (Hidalgo, 1869), MACN 15.05/3280, (E) ventral view, (F) lateral view, (G) dorsal view. Scale line 5 mm.

Bulimus corydon Crosse, 1869: 185; Crosse, 1870: 104, pl. 6, fig. 6; Hidalgo, 1870: 46, pl. 6 fig. 6; Hidalgo, 1893a: 93.

Clathrorthalicus corydon; Breure & Mogollón, 2016: 46, figs. 39D–39G.

Type locality. “Quito, reipublicae Æquatoris (Paz)”.

Type material. “Ecuador”, Coll. Paz “(Cat. Am. mer. no. 80)”, MNCN 15.05/13683 (1), syntype; “Quito”, Coll. Paz “Bulimus Corydon, Crosse/Quito type/Journ. Conchyl. XVII, p./1869 communic. Paz B. 1868”, MNCN 15.05/21868 (1), syntype.

Additional material examined.“Quito, Ecuador”, Coll. Azpeitia, MNCN 15.05/8077 (1).

Remarks. From the labels and further information from the correspondence between Hidalgo and Crosse (Breure & Backhuys, 2017), it may be inferred Hidalgo had two specimens when he was making the Catalogue of the CCP material (Hidalgo, 1870). One specimen was sent to Crosse for description and returned to Hidalgo; both specimens are considered as belonging to the original series. The specimen from the Azpeitia collection undoubtedly originates from Hidalgo, but is not considered as type material since it cannot be ensured it was already in his possession during 1869.

GenusCorona Albers, 1850

Achatina (Corona) Albers, 1850: 193.

Type species. Helix (Cochlitoma) regina Férussac, 1821, by subsequent designation (Martens in Albers, 1860: 226).

Corona pfeifferi (Hidalgo, 1869) [42]

(Fig. 18B)

Orthalicus pfeifferi Hidalgo, 1869c: 412; Hidalgo, 1870: 65, pl. 6 fig. 8; Hidalgo, 1872: 135, pl. 8 figs. 3–4; Hidalgo, 1893a: 56, 113; Hidalgo, 1893b: 292; Azpeitia, 1923: 80; Baratech et al., 1993: 217.

Corona pfeifferi; Breure & Mogollón, 2016: 50, figs. 41A–41E, 43, 89A.

Type locality.[Ecuador] “Canelos, reipublicae Aequatoris”.

Type material.“Canelos, Ecuador”, Coll. Paz, MACN 15.05/3280 (1), syntype. Coll. Hidalgo, MNCN 15.05/18985 (2).

Remarks. Although the material was said to have been collected by Martinez (Baratech et al., 1993), the actual collector was Almagro in June 1865 (Calatayud, 1994: 240 (note 173), 280).

Etymology. Named after Louis Pfeiffer.

Corona regalis (Hupé, 1857) [43]

(Figs. 19A–19B)

Figure 19 Material collected by the CCP.

(A–F) Orthalicidae. Corona regalis (Hupé, 1857), MNCN 15.05/18964, (A) ventral view; MNCN 15.05/61001, (B) ventral view; Kara thompsonii (Pfeiffer, 1845), MNCN 15.05/13701, (C) ventral view; Orthalicus bifulguratus (Reeve, 1849), MNCN 15.05/15386, (D) ventral view; Orthalicus princeps (Broderip in Sowerby I & II, 1833), MNCN 15.05/1898, (E) ventral view; Porphyrobaphe (Oxyorthalicus) irrorata (Reeve, 1849), MNCN 15.05/13287, (F) ventral view. Scale line 1 cm.

Bulimus regalis Hupé, 1857: 34, pl. 10 fig. 3.

Orthalicus bensoni Reeve; Hidalgo, 1870: 64; Hidalgo, 1872: 133, pl. 7 fig. 13; Hidalgo, 1893a: 113; Hidalgo, 1893b: 289.

Orthalicus regina Férussac; Hidalgo, 1870: 64; Hidalgo, 1872: 134; Hidalgo, 1893a: 113; Hidalgo, 1893b: 293.

Type locality.“le Brésil”.

Type material. Not located.

Material examined. “Napo”, “174”, Coll. Hidalgo ex Martinez leg., MNCN 15.05/18964 (2); “Napo”, “175”, Coll. Hidalgo ex Martínez leg. “Ejemplar figurado”, MNCN 15.05/61001 (1).

Remarks. Lot MNCN 15.05/18964 comprises one adult shell and one juvenile; both are sinistral. The systematic position follows the provisional scheme of Breure & Mogollón (2016: 48), awaiting a thorough revision of the genus. The dextral specimen of lot MNCN 15.05/61001 shows superficial resemblance to Orthalicus bensoni (Reeve, 1849), but they lack the fine spiral striation which is present on the type (Breure & Mogollón, 2016: fig. 48C), are more slender, and have the aperture more elongate-ovate. The specimen is herin tentatively referred to Corona regalis (Hupé, 1857), of which the type material has not been located. The original figure (Breure & Mogollón, 2016: fig. 42A) shows a sinistral specimen, but it is known that eniantomorphy occurs within this species (cf. Breure & Mogollón, 2016: figs. 84A–84B). Compared to these figures, the specimen shows three, small spiral bands.

GenusKara Strebel, 1910

Thaumastus (Kara) Strebel, 1910: 16.

Type species. Bulimus thompsonii Pfeiffer, 1845, by monotypy.

Kara thompsonii (Pfeiffer, 1845) [44]

(Fig. 19C)

Bulimus thompsonii Pfeiffer, 1845b: 74; Hidalgo, 1870: 45; Hidalgo, 1872: 63, pl. 6 figs. 2–3; Hidalgo, 1893a: 91; Hidalgo, 1893b: 243.

Type locality. [Ecuador] “Quito”.

Type material. NHMUK 1975464, lectotype (Breure, 1978: 34).

Material examined. “Cuenca (Ecuador)”, Coll. Paz, MNCN 15.05/36937 (2);“Ecuador”, “(Cat. Am. mer. no. 68)”, Coll. Paz, MNCN 15.05/13701 (2); “Pacifico 68”, Coll. Paz, MNCN 15.05/36956 (2); “Cuenca Ecuador”, Coll. Azpeitia, MNCN 15.05/76214 (2 juv.).

Remarks. Hidalgo (1870) reported this species from “Machache et Cuenca, Equateur (Paz)”. In Hidalgo (1872) only the latter locality was mentioned, as ‘Machache’ was likely an error for the Brazilian locality Macahé. This material was not collected by the CCP members themselves as they did not visit Cuenca (Calatayud, 1994); according to Almagro (1866: 164) these shells were a gift from “Yameson” [Jameson] (cf. Calatayud, 1994: 203, 207).

Genus Orthalicus Beck, 1837

Orthalicus Beck, 1837: 59.

Type species. Buccinum zebra Müller, 1774, by subsequent designation (Herrmannsen, 1847 [ 1847–1849]: 159).

Orthalicus bifulguratus (Reeve, 1849) [45]

(Fig. 19D)

Bulimus bifulguratus Reeve, 1849 [1848–1850]: pl. 82 fig. 606.

Orthalicus bifulguratus; Hidalgo, 1893a: 126.

Type locality. [Colombia] “Andes of Columbia”.

Type material. NHMUK 20140082, lectotype (Breure & Schouten, 1985: 29).

Material examined. “Quito”, Coll. Paz, MNCN 15.05/15386 (1).

Remarks. This species was added to the catalogue in 1893; the label of Hidalgo seems to have been lost. The specimen is somewhat smaller and slenderer than the lectotype, but shows the same sculpture on the dorsal side of last whorl.

Orthalicus princeps (Broderip in Sowerby I & II, 1833) [46]

(Fig. 19E)

Bulinus princeps Broderip in Sowerby I & II, 1833 [1832–1841]: fig. 18.

Orthalicus princeps; Hidalgo, 1870: 64; Hidalgo, 1872: 136; Hidalgo, 1893a: 113; Hidalgo, 1893b: 290.

Type locality. [El Salvador] “Conchagua, Central America”.

Type material. Not located.

Material examined.“Taboga”, Coll. Hidalgo ex Martínez leg., MNCN 15.05/18960 (7); “173”, Coll. Hidalgo, MNCN 15.05/18983 (1).

Remarks. Hidalgo (1870) mentioned this species from “Panama (Martínez)”; in 1893 he added “en Colombia”. The island of Taboga was visited by Martinez both in August and in October 1863 (Calatayud, 1994: 258, 261).

Genus Porphyrobaphe Shuttleworth, 1856

Porphyrobaphe Shuttleworth, 1856: 70.

Type species. Bulimus iostomus Sowerby I, 1824, by subsequent designation (Martens in Albers, 1860: 227).

SubgenusPorphyrobaphe (Oxyorthalicus) Strebel, 1909

Porphyrobaphe (Oxyorthalicus) Strebel, 1909: 117.

Type species. Bulimus irrorata Reeve, 1849, by original designation (Strebel, 1909: 102).

Porphyrobaphe (Oxyorthalicus) irrorata (Reeve, 1849) [47]

(Fig. 19F)

Bulimus irrorata Reeve, 1849 [1848–1850]: pl. 62 fig. 427; Hidalgo, 1870: 44; Hidalgo, 1872: 59, pl. 6 fig. 1; Hidalgo, 1893a: 90; Hidalgo, 1893b: 213.

Type locality.“Brazil? New Granada?”.

Type material. NHMUK 1975248 (3), syntypes.

Material examined. “Ecuador”, “(Cat. Am. mer. no. 63)”, Coll. Paz, MNCN 15.05/13287 (2) [white peristome, as ‘var. grevillei’]; “Ecuador”, “(Cat. Am. mer. no. 63)” Coll. Paz, MNCN 15.05/13286 (2) [one specimen with peristome ‘jaune-orange’]; “Nanegal (Ecuador)”, Coll. Hidalgo ex Martínez y Saez leg., MNCN 15.05/36907 (4), one shell labelled with “P-63” inside the aperture; [without locality; unregistered; ‘P-63’ written inside aperture, probably split of from one of the lots mentioned above] (1).

Remarks. Hidalgo (1870) mentioned both material from the Paz and Martínez collections, each with different and more precise localities (“La Mocha et Guaranda” respectively “île de Puna et Macas”). The material of the former two localities was collected by Jímenez de la Espada and Isern in November 1864 (Calatayud, 1994: 268).

SubgenusPorphyrobaphe s.str.

Porphyrobaphe (Porphyrobaphe) iostoma (Sowerby I, 1824) [48]

(Fig. 20A)

Figure 20 Material collected by the CCP.

(A–F) Orthalicidae. Porphyrobaphe (P.) iostoma (Sowerby I, 1824), MNCN 15.05/36949, (A) ventral view; Scholvienia alutacea (Reeve, 1849), MNCN 15.05/13076, (B) ventral view; Scholvienia bifasciata (Philippi, 1845), MNCN 15.05/13282, (C) ventral view; Scholvienia porphyria (Pfeiffer, 1847), MNCN 15.05/36851, (D) ventral view; Scholvienia iserni (Philippi, 1867), MNCN 15.05/13365, (E) ventral view, (F) dorsal view. Scale line 5 mm (B, C, E, F), 1 cm (A, D).

Bulimus iostoma Sowerby I, 1824: 58, pl. 5 fig. 1; Hidalgo, 1870: 44; Hidalgo, 1872: 60, pl. 5 figs. 7–8; Hidalgo, 1893a: 90; Hidalgo, 1893b: 285.

Type locality. No type locality given.

Type material. Not located.

Material examined. “Guayaquil”, “(Cat. Am. mer. no. 66)” Coll. Paz, MNCN 15.05/3495 (2), MNCN 15.05/13498 (1), MNCN 15.05/13499 (1), MNCN 15.05/13500 (1); “Guayaquil”, Coll. Hidalgo ex “Paz y Martínez” leg. “uno de los ejemplares figurado”, MNCN 15.05/36949 (6).

Remarks. Hidalgo (1870) mentioned this material as “Guayaquil (Paz et Martinez), île de Puna et Macas (Martinez)”. One of the specimens is very small but otherwise seems adult and typical.

Genus Scholvienia Strebel, 1910

Scholvienia Strebel, 1910: 20.

Type species. Bulimus bitaeniatus Nyst, 1845, by subsequent designation (Pilsbry, 1932: 391).

Scholvienia alutacea (Reeve, 1849) [49]

(Fig. 20B)

Bulimus alutaceus Reeve, 1849 [1848–1850]: pl. 72 fig. 522.

Bulimus tarmensis Philippi; Hidalgo, 1870: 61; Hidalgo, 1872: 114, pl. 4 figs. 8–9; Hidalgo, 1893a: 109; Hidalgo, 1893b: 207.

Type locality. [Peru] “Cuzco, Bolivia”.

Type material. NHMUK 1975148, lectotype (Breure, 1978).

Material examined. “Chanchamayo”, “(Cat. Am. mer. no. 121)”, Coll. Paz, MNCN 15.05/13076 (2); “Peru”, “(Cat. Am. mer. no. 163)”, Coll. Paz, MNCN 15.05/13168 (3).

Remarks. Hidalgo (1870) listed this material as “Hab. Chanchamayo, Pérou (Isern)”; the label “(Cat. Am. mer. no. 121)” was apparently misplaced. Possibly these shells were among the material listed by Isern (“28 Bulimus y 4 en alcohol”), collected near Acobamba on the 8th October 1863 (Blanco, Rodríguez & Rodríguez, 2006: 143).

Scholvienia bifasciata (Philippi, 1845) [50]

(Fig. 20C)

Bulimus bifasciatus Philippi, 1845 [1845–1847]: 10, pl. 3 fig. 5; Hidalgo, 1870: 46; Hidalgo, 1872: 68; Hidalgo, 1893a: 92; Hidalgo, 1893b: 209.

Type locality. [Peru] “sylvae peruanae”.

Type material. Not located.

Material examined. “Chanchamayo”, “(Cat. Am. mer. no. 73)”, Coll. Paz, MNCN 15.05/13282 (1); “Chanchamayo (Perú)”, “Pacifico”, Coll. Hidalgo ex Isern, MNCN 15.05/7189 (6); “Chanchamayo”, Coll.Hidalgo ex Isern leg., MNCN 15.05/21243 (10); “Peru”, Coll. Hidalgo ex Isern leg., MNCN 15.05/20339 (2); “Peru”, Coll. Azpeitia, MNCN 15.05/8128 (1).

Remarks. The material consists of specimens ranging in shell height from 44.9 to 59.0 mm, all showing the same characteristics. Awaiting a revision of this and morphologically similar species from the same area (Breure & Mogollón, 2016: 67), all specimens are considered to be conspecific.

Scholvienia iserni (Philippi, 1867) [51]

(Figs. 20E–20F)

Bulimus iserni Philippi, 1867: 75; Hidalgo, 1870: 45; Hidalgo, 1872: 67, pl. 6 figs. 6–7; Hidalgo, 1893a: 92; Hidalgo, 1893b: 208.

Type locality. [Peru] “prope La Oroya”.

Type material. Not located.

Material examined. “Chanchamayo, Peru”, Coll. Paz “(Cat. Am. mer. no. 72)”, MNCN 15.05/13365 (2); “Chanchamayo (Perú)”, Coll. Hidalgo [ex Isern leg.], MNCN 15.05/37156 (4).

Remarks. According to the published data by Hidalgo this material was collected by Isern, possibly between La Oroya and Tarma on the 29th September 1863 (Blanco, Rodríguez & Rodríguez, 2006: 143). Also Philippi (1867) mentioned “legit amicus infelix, Johannes Isern”.

Etymology. Named after the collector, Juan Isern y Battló.

Scholvienia porphyria (Pfeiffer, 1847) [52]

(Fig. 20D)

Bulimus porphyrius Pfeiffer, 1847a: 114; Hidalgo, 1870: 45; Hidalgo, 1872: 65; Hidalgo, 1893a: 91.

Type locality.“Bolivia”.

Type material. NHMUK 1975277, lectotype (Breure, 1978: 46).

Material examined. “Peru”, Coll. Hidalgo ex Almagro, MNCN 15.05/36851 (3).

Remarks. This species is known to occur in Peru, Dept. Apurimac (Breure & Mogollón, 2016: 71); the material was probably collected by Almagro during his trip through this region in August 1863 (Calatayud, 1994: 256).

GenusSultana Shuttleworth, 1856

Orthalicus (Sultana) Shuttleworth, 1856: 58.

Type species. Helix sultana Dillwyn, 1817, by tautonomy.

Subgenus Sultana (Metorthalicus) Pilsbry, 1899

Orthalicus (Metorthalicus) Pilsbry, 1899: 187.

Type species. Bulimus yatesi Pfeiffer, 1855, by original designation.

Sultana (Metorthalicus) deburghiae (Reeve, 1859) [53]

(Fig. 21A)

Figure 21 Material collected by the CCP.

(A–D) Orthalicidae. Sultana (Metorthalicus) deburghiae (Reeve, 1859), MNCN 15.05/36960, (A) ventral view; Sultana (Metorthalicus) fraseri (Pfeiffer, 1858), MNCN 15.05/13505, (B) ventral view; Sultana (Metorthalicus) kellettii (Reeve, 1850), MNCN 15.05/6881, (C) ventral view; Sultana (Metorthalicus) yatesi yatesi (Pfeiffer, 1855), MNCN 15.05/13504, (D) ventral view. Scale line 1 cm.

Bulimus deburghiae Reeve, 1859: 123.

Bulimus gloriosus Pfeiffer; Hidalgo, 1870: 44; Hidalgo, 1872: 62, pl. 4 figs. 2–3; Hidalgo, 1893a: 90; Hidalgo, 1893b: 287.

Type locality. “Peruvian side of the Amazon”.

Type material. NHMUK 19601622, lectotype (Breure & Schouten, 1985: 27).

Material examined. “San José (Ecuador)”, Coll. Hidalgo ex “Isern y Espada” leg., MNCN 15.05/36960 (2); “Ecuador”, “(Cat. Am. mer. no. 64)”, Coll. Paz, MNCN 15.05/13702 (2), MNCN 15.05/76247 (1), MNCN 15.05/76248 (1).

Remarks. The material was probably collected by Isern in June 1863 (Calatayud, 1994: 278). One of the specimens corresponds to Hidalgo, 1872: pl. 4 figs. 2–3.

Sultana (Metorthalicus) fraseri (Pfeiffer, 1858) [54]

(Fig. 21B)

Bulimus fraseri Pfeiffer, 1858: 239; Hidalgo, 1870: 44; Hidalgo, 1893a: 90.

Type locality. “in provincia Cuenca reipublicae Aequatoris”.

Type material. NHMUK 20140083, lectotype (Breure & Schouten, 1985: 28).

Material examined. “Ecuador”, Coll. Paz, MNCN 15.05/13505 (2); “Pacifico 62”, Coll. Paz, MNCN 15.05/36963 (2); “Quito, Ecuador”, Coll. Azpeitia, MNCN 15.05/76216 (1).

Remarks. Hidalgo (1870) wrote “Trouvé sur le chemin de Quito, à 30 ou 40 kilomètres de Chimborazo (Paz)”. Probably collected by Almagro or Isern during their trip from Guayaquil to Quito (Calatayud, 1994: 268).

Sultana (Metorthalicus) kellettii ( Reeve, 1850) [55]

(Fig. 21C)

Bulimus kellettii Reeve, 1850 [1848–1850]: pl. 89 fig. 661.

Bulimus fungairinoi Hidalgo, 1867: 72, pl. 4 fig. 4, 478; Hidalgo, 1870: 44; Hidalgo, 1872: 58, pl. 3 figs. 8–9; Hidalgo, 1893a: 90; Hidalgo, 1893b: 285; Azpeitia, 1923: 58; Fischer-Piette, 1950: 68.

Bulimus jungairignoi [sic] Baratech et al., 1993: 215.

Sultana (Metorthalicus) kellettii; Breure & Mogollón, 2016: 75, figs. 73A, 79A–79B, 80.

Type locality. “Ecuador?”.

Type material. NHMUK 1975241, lectotype (Breure & Schouten, 1985: 28).

Additional type material. “Cuenca (Ecuador)”, Coll. Hidalgo ex Jamieson, MNCN 15.05/3159 (2); “Ecuador”, “(Cat. Am. mer. no. 65)”, Coll. Paz, MNCN 15.05/6881 (1), syntypes of Bulimus fungairinoi Hidalgo.

Material examined.“Quito, Ecuador”, Coll. Azpeitia, MNCN 15.05/3162 (1); “Cuenca, Ecuador”, Coll. Azpeitia, MNCN 15.05/3161 (1); “Cuenca, Ecuador”, Coll. Paz, MNCN 15.05/3160 (1).

Remarks. This species was initially published as Bulimus jungairinoi, but Hidalgo made Crosse correct this in the index (p. 478); see also Breure & Backhuys, 2017. This correction has to be considered as a lapsus calami (Art. 32.5.1.1 ICZN Code). The material was not collected by the CCP members themselves, but was a gift of J. Jameson (cf. Calatayud, 1994: 203).

Etymology. Hidalgo named his taxon after Eduardo Fungairiño, a befriended Madrid-based malacologist (Breure & Backhuys, 2017).

Sultana (Metorthalicus) yatesi yatesi (Pfeiffer, 1855) [56]

(Fig. 21D)

Bulimus yatesi Pfeiffer, 1855: 93, pl. 31 fig. 5; Hidalgo, 1872: 59; Hidalgo, 1893a: 125.

Type locality. [Peru] “Meobamba”.

Type material. NHMUK 1975239, lectotype (Breure & Schouten, 1985: 28).

Material examined. “Ecuador”, “(Cat. Am. mer. no. […])”, Coll. Paz, MNCN 15.05/13504 (1).

Remarks. Hidalgo (1893) published this species with locality data “República del Peru (Almagro)”. The species is known to occur in northern Peru at the eastern side of the Andes, but the subspecies Sultana (Metorthalicus) yatesi galactostoma (Ancey, 1890) has been reported from Ecuador without specific locality (Breure & Mogollón, 2016). These authors also reported a record for the nominate taxon from the Chanchamayo valley. In any case, there is no evidence this material was collected by Almagro, who has not travelled in Peru in areas where this species does occur. If the label “Ecuador” has to be trusted, it is likely this specimen was collected on the eastern slopes of the Cordillera.

Family Odontostomidae Pilsbry & Vanatta, 1898

Genus Anctus Martens in Albers, 1860

Anctus Martens in Albers, 1860: 214.

Type species. Bulimus angiostomus Wagner, 1827, by monotypy.

Anctus angiostomus (Wagner, 1827) [57]

(Fig. 22A)

Figure 22 Material collected by the CCP.

(A–M) Odontostomidae. Anctus angiostomus (Wagner, 1827), MNCN 15.05/13152, (A) ventral view; Bahiensis bahiensis (Moricand, 1834), MNCN 15.05/13097, (B) ventral view; Bahiensis janeirensis (Sowerby I in Sowerby I & II, 1833), MNCN 15.05/13196, (C) ventral view; Spixia charpentieri (Grateloup in Pfeiffer, 1850), MNCN 15.05/20205, (D) ventral view; Spixia striata (Spix in Wagner, 1827), MNCN 15.05/13078, (E) ventral view; Cyclodontina inflata (Wagner, 1827), MNCN 15.05/8456, (F) ventral view; Macrodontes gargantua (Férussac, 1822), MNCN 15.05/13366, (G) ventral view; Moricandia dubiosa (Jay, 1839), MNCN 15.05/12998, (H) ventral view; Burringtonia exesa (Spix in Wagner, 1827), MNCN 15.05/13364, (I) ventral view; Plagiodontes daedaleus (Deshayes in Férussac & Deshayes, 1851), MNCN 15.05/13153, (J) ventral view; Plagiodontes dentata (Wood, 1828), MNCN 15.05/13167, (K) ventral view; Burringtonia leucotrema (Beck, 1837), MNCN 15.05/13470, (L) ventral view; Burringtonia labrosa (Menke, 1828), MNCN 15.05/13472, (M) ventral view. Scale line 5 mm.

Bulimus angiostomus Wagner, 1827: 14.

Bulimus capueira Spix; Hidalgo, 1893a: 125.

Type locality. [Brazil] “Capueira a Brasiliensibus dictis, in Provinces septemtrionalibus”.

Type material.

Material examined. “Brazil”, Coll. Paz, MNCN 15.05/13152 (4); “Brasil”, Coll. Azpeitia, MNCN 15.05/8075 (2).

Remarks. This species was listed in Hidalgo (1893a) as “Bulimus capueira Spix”, which is a synonym.

Genus Bahiensis Jousseaume, 1877

Bahiensis Jousseaume, 1877: 311.

Type species. Helix (Cochlogena) bahiensis Moricand, 1834, by monotypy.

Bahiensis bahiensis (Moricand, 1834) [58]

(Fig. 22B)

Helix (Cochlogena) bahiensis Moricand, 1834: 541, pl. 1 fig. 6.

Bulimus bahiensis; Hidalgo, 1870: 63; Hidalgo, 1893a: 112.

Type locality. [Brazil] “le Brésil dans les bois près de Bahia [Salvador]”

Type material. MHNG-INVE-64638 (31), syntypes.

Material examined. “Rio Janeiro”, “(Cat. Am. mer. no. 172)”, Coll. Paz, MNCN 15.05/13097 (5); “Rio Janeiro”, “(comprado)”, Coll. Hidalgo, MNCN 15.05/20324 (1).

Remarks. The shell from the Hidalgo collection originated without doubt from Paz, who bought the material while in Brazil.

Bahiensis janeirensis (Sowerby I in Sowerby I & II, 1833) [59]

(Fig. 22C)

Bulinus janeirensis Sowerby I in Sowerby I & II, 1833 [1832–1841]: 8, fig. 97.

Bulimus janeirensis; Hidalgo, 1870: 52; Hidalgo, 1893a: 99.

Type locality. [Brazil] “Rio de Janeiro”.

Type material. Not located.

Material examined. “Rio Janeiro”, “(Cat. Am. mer. no. 109)”, Coll. Paz, MNCN 15.05/13196 (2).

Genus Burringtonia Parodiz, 1944

Burringtonia Parodiz, 1944: 4.

Type species. Helix (Cochlodina) pantagruelina Moricand, 1834, by original designation.

Burringtonia exesa (Spix in Wagner, 1827) [60]

(Fig. 22I)

Clausilia exesa Spix in Wagner, 1827: pl. 14 fig. 1.

Bulimus exesus; Hidalgo, 1870: 51; Hidalgo, 1893a: 98.

Type locality. Not given.

Type material. ZSM.

Material examined. “Brasil”, “(Cat. Am. mer. no. 105)”, Coll. Paz, MNCN 15.05/13364 (4).

Burringtonia labrosa (Menke, 1828) [61]

(Fig. 22M)

Scarabus labrosus Menke, 1828: 78.

Bulimus pantagruelinus Moricand; Hidalgo, 1870: 51; Hidalgo, 1893a: 98.

Type locality. “inter Rio et Campos, in Brasilia”.

Type material. Not located.

Material examined.“Brasil”, “(Cat. Am. mer. no. 103)”, Coll. Paz, MNCN 15.05/13472 (1); “Rio Janeiro”, “(comprado)”, Coll. Hidalgo ex Paz, MNCN 15.05/36849 (4).

Remarks. Lot MNCN 15.05/13472 corresponds with the material identified by Hidalgo as “Bulimus pantagruellinus Moricand”.

Burringtonia leucotrema (Beck, 1837) [62]

(Fig. 22L)

Odontostomus leucotremus Beck, 1837: 54.

Bulimus leucotrema; Hidalgo, 1893a: 122

Type locality. “Brasil. Bah[ia]”.

Type material. Not located.

Material examined. “Brazil”, Coll. Paz, MNCN 15.05/13470 (1).

Remarks. Hidalgo (1893a) recorded as locality “Bahia, en el Brasil (Paz)”.

Genus Cyclodontina Beck, 1837

Pupa (Cyclodontina) Beck, 1837: 88.

Type species. Clausilia pupoides Spix in Wagner, 1827, by subsequent designation (Pilsbry, 1898: 57).

Cyclodontina inflata (Wagner, 1827) [63]

(Fig. 22F)

Pupa inflata Wagner, 1827: 20.

Type locality. [Brazil] “in Provinciis mediis orientalibus”.

Type material. ZSM.

Material examined. “Brasil”, Coll. Azpeitia ex Paz leg., MNCN 15.05/8456 (1).

Remarks. This species was not mentioned in Hidalgo’s catalogue, but the material is likely originating from the CCP.

Genus Macrodontes Swainson, 1840

Clausilia (Macrodontes) Swainson, 1840: 334.

Type species. Macrodontes sowerbyii Swainson, 1840, by monotypy.

Macrodontes gargantua (Férussac, 1822) [64]

(Fig. 22G)

Helix (Cochlodina) gargantua Férussac 1822 [ 1821–1822]: 62.

Bulimus odontostomus Sowerby; Hidalgo, 1870: 51; Hidalgo, 1893a: 98.

Type locality. Not given.

Type material. Not located.

Material examined. “Corcovado, Rio Jan.[eiro]”, “(Cat. Am. mer. no. 104)”, Coll. Paz, MNCN 15.05/13366 (4); “Corcovado, Rio Janeiro, Brasil”, Coll. Azpeitia ex Paz leg., MNCN 15.05/7333 (1).

Genus Moricandia Pilsbry & Vanatta in Pilsbry, 1898

Odontostomus (Moricandia) Pilsbry & Vanatta in Pilsbry, 1898: 57.

Type species. Helix fusiformis Rang, 1831, by original designation.

Moricandia dubiosa (Jay, 1839) [65]

(Fig. 22H)

Bulimus dubiosus Jay, 1839: 122, pl. 7 fig. 6.

Bulimus fusiformis Rang; Hidalgo, 1870: 50.

Type locality.“Brazil ?”.

Type material. Not located.

Material examined.“Rio Janeiro”, “(Cat. Am. mer. no. 101)”, Coll. Paz, MNCN 15.05/12998 (3); “Rio Janeiro (comprado)”, Coll. Hidalgo, MNCN 15.05/37050 (1).

Remarks. One of the specimens was bought by Paz in Rio de Janeiro, where it does occur in the vicinity (Simone, 2006). The systematic position is following the same author.

Genus Plagiodontes Doering, 1877

Plagiodontes Doering, 1877: 318.

Remarks. The year of publication is according to Breure & Miquel (2012: 19).

Type species. Pupa dentata Wood, 1828, by subsequent designation (Pilsbry, 1898: 57).

Plagiodontes daedaleus (Deshayes in Férussac & Deshayes, 1851) [66]

(Fig. 22J)

Pupa dealdalea Deshayes in Férussac & Deshayes, 1851 [1819–1851]: [2 (2)] 217, pl. 162 figs. 23–24.

Bulimus daedaleus; Hidalgo, 1870: 51; Hidalgo, 1893a: 98.

Type locality. “Brésil”.

Type material. Not located.

Material examined.“Republ. Argentina”, “(Cat. Am. mer. no. 107)”, Coll. Paz, MNCN 15.05/13153 (4).

Remarks. According to Hidalgo (1870) the material was collected “Salto Oriental”. See Calatayud, 1994: 252 for the itinerary of Paz, and part of the CCP, through Argentina.

Plagiodontes dentata (Wood, 1828) [67]

(Fig. 22K)

Pupa dentata Wood, 1828: 50, pl. 8 fig. 71.

Bulimus dentatus; Hidalgo, 1870: 51; Hidalgo, 1872: 80; Hidalgo, 1893a: 98; Hidalgo, 1893b: 187.

Type locality. Not given.

Type material. NHMUK 1840.9.12.50 (2), syntypes.

Material examined. “La Concordia”, Coll. Hidalgo ex Paz leg., MNCN 15.05/36385 (21); “Montevideo”, Coll. Hidalgo ex Martínez leg., MNCN 15.05/36382 (7); “Las Mercedes”, Coll. Hidalgo ex Paz leg., MNCN 15.05/36293 (3); “Republ. Argentina”, “(Cat. Am. mer. no. 106)”, Coll. Paz, MNCN 15.05/13167 (5).

Remarks. Hidalgo (1872) mentioned the specimens from Paz as collected at “La Concordia y Las Mercedes”. See Calatayud, 1994: 251–252 for the places visited around Montevideo.

Genus Spixia Pilsbry & Vanatta in Pilsbry, 1898

Odontostomus (Spixia) Pilsbry & Vanatta in Pilsbry, 1898: 57.

Type species. Clausilia striata Spix in Wagner, 1827, by subsequent designation (Pilsbry, 1901 [1901–1902]: 67).

Remarks. The designation by Pilsbry (1901 [1901–1902]) was “O. spixii Orb.”, which was afterwards shown to comprise two species (Breure & Ablett, 2012: 25–26).

Spixia charpentieri (Grateloup in Pfeiffer, 1850) [68]

(Fig. 22D)

Bulimus charpentieri Grateloup in Pfeiffer, 1850: 14; Hidalgo, 1870: 52; Hidalgo, 1872: 81; Hidalgo, 1893a: 99; Hidalgo, 1893b: 185.

Type locality. [Argentina] “Cardova [sic, Cordoba] reipubl. Argentinae”.

Type material. Not located.

Material examined. “Republ. Argentina”, “(Cat. Am. mer. no. 108)”, Coll. Paz, MNCN 15.05/13091 (7); “Republ. Argentina”, “(Cat. Am. mer. no. 108)”, Coll. Paz, MNCN 15.05/13096 (4); “Cordoba de Tucuman”, Coll. Hidalgo ex Paz, MNCN 15.05/20205 (41); “Cordoba, Rep. Argentina”, Coll. Azpeitia, MNCN 15.05/7192 (19); Coll. Hidalgo, MNCN 15.05/19972 (28); “Pupa Porriana Grateloup. Cordoba, Rep. Argentina—sp.nov. | Bulimus Charpentieri”, Coll. Paz, MNCN 15.05/76225 (2).

Remarks. The material varies in size and colouration, some with a brownish apex and fine, axial lines, others totally whitish.

Spixia striata (Spix in Wagner, 1827) [69]

(Fig. 22E)

Clausilia striata Spix in Wagner, 1827: pl. 14 fig. 1.

Bulimus exesus Spix; Hidalgo, 1870: 51; Hidalgo, 1893a: 98.

Type locality. [Brazil] “in Provinciis S. Pauli et Sebastianopolitana”.

Type material. ZSM.

Material examined. “Rio Janeiro”, “(Cat. Am. mer. no. 105)”, Coll. Paz, MNCN 15.05/13078 (3).

Remarks. These specimens were found identified by Hidalgo as “Bulimus exesus Spix”, which refers to Pupa exesa Wagner, 1827.

Family Bothriembryontidae Iredale, 1937

Genus Plectostylus Beck, 1837

Plectostylus Beck, 1837: 58.

Type species. Bulimus peruvianus Bruguière, 1789, by subsequent designation (Gray, 1847: 176).

Plectostylus broderipii (Sowerby I in Broderip & Sowerby I, 1832) [70]

(Fig. 23A)

Figure 23 Material collected by the CCP.

(A–G) Bothriembryontidae. Plectostylus broderipii (Sowerby I in Broderip & Sowerby I, 1832), MNCN 15.05/13467, (A) ventral view; Plectostylus coquimbensis (Broderip in Broderip & Sowerby I, 1832), MNCN 15.05/13670, (B) ventral view; Plectostylus cf. reflexus (Pfeiffer, 1842), MNCN 15.05/13886, (C) ventral view; Plectostylus chilensis (Lesson in Lesson, Garnot & Guérin-Méneville, 1830), MNCN 15.05/13384, (D) ventral view; Plectostylus peruvianus (Bruguière, 1792), MNCN 15.05/13466, (E) ventral view; Plectostylus punctulifer (Sowerby I in Sowerby I & II, 1833), MNCN 15.05/76237, (F) ventral view; Plectostylus coturnix (Sowerby I in Broderip & Sowerby I, 1832), MNCN 15.05/13678, (G) ventral view. Scale line 5 mm.

Bulinus broderipii Sowerby I in Broderip & Sowerby I, 1832a: 30.

Bulimus broderipi [sic]; Hidalgo, 1870: 58; Hidalgo, 1872: 117. [partim].

Type locality. [Chile] “prope Copiapo Chilensium”

Type material. NHMUK 20100655, lectotype (Breure & Ablett, 2012: 8).

Material examined. “Bolivia”, “(Cat. Am. mer. no. 151)”, Coll. Paz, MNCN 15.05/13467 (6); “Paposo”, Coll. Hidalgo, MNCN 15.05/20193 (2), MNCN 15.05/37162 (10); Coll. Azpeitia, MNCN 15.05/8073 (3); “Huasco, Chile”, Coll. Azpeitia, MNCN 15.05/8074 (2).

Remarks. The material was listed in Hidalgo (1870) as “Huasco, Chili (Martínez); Paposo, Bolivia (Paz)”; both localities are in present-day Chile. Therefore, lot MNCN 15.05/13467 is likely also from Paposo; lot MNCN 15.05/8074 may have originated from Martínez, but reference to his name has been lost. See Calatayud, 1994: 258 for the itinerary in northern Chile.

Plectostylus chilensis (Lesson in Lesson, Garnot & Guérin-Méneville, 1830) [71]

(Fig. 23D)

Bulimus chilensis Lesson in Lesson et al., 1830 [1826–1831]: pl. 7 fig. 3; Hidalgo, 1870: 55; Hidalgo, 1872: 103; Hidalgo, 1893a: 103; Hidalgo, 1893b: 227.

Type locality. [Chile] “l’ancienne ville de Penco, dans la province de la Concepcion” (Lesson, 1831 [1830–1831]: 317).

Type material. Not located.

Material examined. “Santo. de Chile”, “(Cat. Am. mer. no. 129)”, Coll. Paz, MNCN 15.05/13384 (4); “Valparaiso”, Coll. Hidalgo ex “Martínez y Paz” leg., MNCN 15.05/36386 (8); “Pacifico 129”, Coll. Hidalgo, MNCN 15.05/20192 (4).

Remarks. The dates of publication of Lesson are according to Cretella (2010).

Plectostylus coquimbensis (Broderip in Broderip & Sowerby I, 1832) [72]

(Fig. 23B)

Bulinus coquimbensis Broderip in Broderip & Sowerby I, 1832a: 30.

Bulimus coquimbensis; Hidalgo, 1870: 59; Hidalgo, 1872: 116; Hidalgo, 1893a: 107; Hidalgo, 1893b: 223.

Type locality.“Chili, Coquimbo”.

Type material. Not located.

Material examined.“Chile”, “(Cat. Am. mer. no. 152)”, Coll. Paz, MNCN 15.05/13670 (3); “Coquimbo”, Coll. Hidalgo ex Martínez, MNCN 15.05/21226 (8).

Remarks. Hidalgo (1870) wrote “Coquimbo, República de Chile (Paz y Martínez)”, so we must assume that both lots were collected in the same region.

Plectostylus coturnix (Sowerby I in Broderip & Sowerby I, 1832) [73]

(Fig. 23G)

Bulinus coturnix Sowerby I in Broderip & Sowerby I, 1832a: 30.

Bulimus coturnix; Hidalgo, 1870: 58; Hidalgo, 1872: 115; Hidalgo, 1893a: 106; Hidalgo, 1893b: 224.

Type locality. [Chile] “Huasco”.

Type material. NHMUK 20100620 (5), possible syntypes.

Material examined. “Chile”, “(Cat. Am. mer. no. 150)”, Coll. Paz, MNCN 15.05/13678 (6); “Huasco”, Coll. Hidalgo ex Martínez, MNCN 15.05/20322 (3); “150”, Coll. Hidalgo, MNCN 15.05/20245 (1).

Remarks. The material was collected at “Huasco” by both Paz and Martínez according to Hidalgo (1870, 1872). The largest specimen exceeds the measurement given by Hidalgo (1872).

Plectostylus peruvianus (Bruguière, 1792) [74]

(Fig. 23E)

Bulimus peruvianus Bruguière, 1792: 320; Hidalgo, 1870: 55; Hidalgo, 1872: 102; Hidalgo, 1893a: 103; Hidalgo, 1893b: 225.

Type locality. “Pérou”.

Type material. MNHN 24188, lectotype (Breure, 1975: 1143).

Material examined. “Chile”, “(Cat. Am. mer. no. 128)”, Coll. Paz, MNCN 15.05/13359 (4); MNCN 15.05/13466 (2); “Pacifico 128”, MNCN 15.05/76213 (2); “Valparaiso”, Coll. Hidalgo ex Martínez leg., MNCN 15.05/7338 (2); “Valparaiso, Chile”, Coll. Azpeitia, MNCN 15.05/13888 (2).

Plectostylus punctulifer (Sowerby I in Sowerby I & II, 1833) [75]

(Fig. 23F)

Bulinus punctulifer Sowerby I in Sowerby I & II, 1833 [1833–1838]: 36

Bulimus broderipi [sic]; Hidalgo, 1870: 58; Hidalgo, 1872: 117; Hidalgo, 1893a: 106; Hidalgo, 1893b: 221 [partim].

Type locality. [Chile] “Questa Prado”.

Type material. NHMUK 1975171 (8), syntypes.

Material examined. “Paposo”, Coll. Hidalgo, MNCN 15.05/76237 (1).

Remarks. This specimen was among lot MNCN 15.05/20193, identified as Bulimus broderipii, but may be regarded as a somewhat odd specimen of Plectostylus punctulifer which occurs sympatrically at this locality (JF Araya, pers. comm., 2016).

Plectostylus cf. reflexus (Pfeiffer, 1842) [76]

(Fig. 23C)

Succinea reflexa Pfeiffer, 1842: 56.

Type locality. “Pichidanque probe Coquimbo, Chile”.

Type material. NHMUK 1975358, lectotype (Breure, 1978: 202).

Material examined. “Chile”, Coll. Hidalgo, MNCN 15.05/13886 (1).

Remarks. The (subadult) specimen is only tentatively referred to this species.

Family Bulimulidae Tryon, 1867

Genus Auris Spix in Wagner, 1827

Auris Spix in Wagner, 1827: 13.

Type species. Bulimus melastomus Swainson, 1820, by subsequent designation (Gray, 1847: 175).

Auris chrysostoma (Moricand, 1836) [77]

(Fig. 24A)

Figure 24 Material collected by the CCP.

(A–D) Bulimulidae. Auris chrysostoma (Moricand, 1836), MNCN 15.05/8123, (A) ventral view; Auris egregia (Jay, 1836), MNCN 15.05/13387, (B) ventral view; Auris melastoma (Swainson, 1820), MNCN 15.05/13477, (C) ventral view; Auris illheocola (Moricand, 1836), MNCN 15.05/13277, (D) ventral view. Scale line 5 mm (B–C), 1 cm (A, D).

Helix (Cochlogena) rhodospira var. β chrysostoma Moricand, 1836: 428.

Bulimus swainsoni; Hidalgo, 1893a: 123.

Type locality. [Brazil] “environs de Bahia [Salvador]”.

Type material. MHNG-INVE-60161 (5), syntypes.

Material examined.“Brasil”, Coll. Paz, MNCN 15.05/13502 (1); “Brasil”, Coll. Azpeitia, MNCN 15.05/8123 (1); “Rio Janeiro”, “(comprado)”, Coll. Hidalgo ex Martínez, MNCN 15.05/36691 (9).

Remarks. The specimen from the Azpeitia collection is much smaller than the type specimen (Breure, 2016: fig. 88), but otherwise seems to be adult.

Auris egregia (Jay, 1836) [78]

(Fig. 24B)

Pupa egregia Jay, 1836: 81, pl. 1 fig. 4.

Bulimus bilabiatus Broderip; Hidalgo, 1893a: 123.

Type locality. “Brazil”.

Type material. Not located.

Material examined. “Brasil”, Coll. Paz, MNCN 15.05/13387 (1).

Remarks. The shape of the aperture and the colouration of the peristome makes us identify this specimen as Jay’s species (Simone, 2006: fig. 425). The sculpture on the ventral side of the last whorl is stronger than in his figure.

Auris illheocola (Moricand, 1836) [79]

(Fig. 24D)

Helix (Cochlogena) rhodospira var. illheocola Moricand, 1836: 428.

Bulimus illheocola; Hidalgo, 1893a: 123.

Type locality. [Brazil] “Illheos”.

Type material. MHNG-INVE-60171 (6), syntypes; MHNG-INVE-60169 (2), probable syntypes.

Material examined. “Brasil”, Coll. Paz, MNCN 15.05/13277 (1); “Brasil”, “224”, Coll. Paz, MNCN 15.05/36926 (3).

Auris melastoma (Swainson, 1820) [80]

(Fig. 24C)

Bulimus melastomus Swainson, 1820 [1820–1821]: pl. 4; Hidalgo, 1870: 46; Hidalgo, 1893a: 93.

Type locality. “Brazil, in the province of Bahia”.

Type material. Not located.

Material examined.“Brasil”, “(Cat. Am. mer. no. 79)”, Coll. Paz, MNCN 15.05/13477 (2); “Rio Janeiro, Brasil”, Coll. Azpeitia, MNCN 15.05/8108 (2).

Remarks. Hidalgo (1893a) reported the species from “Rio Janeiro, en el Brasil (Martinez)”.

Genus Bostryx Troschel, 1847

Bulimus (Bostryx) Troschel, 1847: 49.

Type species. Bulimus (Bostryx) solutus Troschel, 1847, by monotypy.

Bostryx aequicostatus (Rehder, 1945) [81]

(Fig. 25A)

Figure 25 Material collected by the CCP.

(A–H) Bulimulidae. Bostryx aequicostatus (Rehder, 1945), MNCN 15.05/14546, (A) ventral view; Bostryx bilineatus (Sowerby I, 1833), MNCN 15.05/13216, (B) ventral view; Bostryx conspersus (Sowerby I, 1833), MNCN 15.05/13179, (C) ventral view; Bostryx tricinctus (Reeve, 1848), MNCN 15.05/13321, (D) ventral view; Bostryx modestus (Broderip in Broderip & Sowerby I, 1832), MNCN 15.05/13160, (E) ventral view; Bostryx scalariformis (Broderip in Broderip & Sowerby I, 1832), MNCN 15.05/21318, (F) ventral view; Bostryx laurentii (Sowerby I, 1833), MNCN 15.05/13404, (G) ventral view; Bostryx veruculum (Morelet, 1860), MNCN 15.05/14526, (H) ventral view. Scale line 5 mm.

Bulimus scalarioides Philippi in Pfeiffer, 1867: 77; Hidalgo, 1870: 53; Hidalgo, 1893a: 101; Hidalgo, 1875: 128, pl. 7 fig. 4; Hidalgo, 1893a: 101. Not Bulimus scalarioides Reeve, 1849.

Peronaeus aequicostata Rehder, 1945: 106.

Type locality. [Peru] “provincia Conchucos”.

Type material. Not located.

Material examined. “Peru”, “(Cat. Am. mer. no. 117)”, Coll. Paz, MNCN 15.05/14546 (3).

Remarks. Hidalgo (1870) gave as locality “Pataz, Pérou (Paz)”, which is in northern Peru, Dept. La Libertad. He compared the shells with Bostryx scalaricosta (Morelet, 1863), which is a species from southern Peru and clearly distinct (cf. Breure, 2016: fig. 98). Rehder (1945: 106) noticed the name Bulimus scalarioides Philippi in Pfeiffer, 1867 was preoccupied by Bulimus scalarioides Reeve, 1849, and introduced Peronaeus aequicostata Rehder, 1945 as a replacement name. Philippi described his taxon from “provincia Conchucos”, which is in Dept. Ancash ca. 60 km west of the locality mentioned by Hidalgo (1870). It may be noted that the itinerary of the CCP members does not mention this region (Calatayud, 1994), hence it is unclear who collected this material. The shells, however, correspond to Philippi’s description. Weyrauch (1964: fig. 13) has figured one of the shells from the original series collected by Raimondi, now IFML-MOLL 1223a.

Bostryx affinis (Broderip in Broderip & Sowerby I, 1832) [82]

(Fig. 26A)

Figure 26 Material collected by the CCP.

(A–I) Bulimulidae. Bostryx affinis (Broderip in Broderip & Sowerby I, 1832), MNCN 15.05/13170, (A) ventral view; Bostryx albicans (Broderip in Broderip and Sowerby I, 1832), MNCN 15.05/13162, (B) ventral view; Bostryx anachoreta (Pfeiffer, 1856), MNCN 15.05/13173, (C) ventral view; Bostryx atacamensis (Pfeiffer, 1856), MNCN 15.05/13093, (D) ventral view; Bostryx hamiltoni (Reeve, 1849), MNCN 15.05/9029, (E) ventral view; Bostryx derelictus (Broderip in Broderip & Sowerby I, 1832), MNCN 15.05/13083, (F) ventral view; Bostryx hennahi (Gray, 1828), MNCN 15.05/12993, (H) ventral view; Bostryx holostoma (Pfeiffer, 1846), MNCN 15.05/14604, (I) ventral view. Scale line 5 mm.

Bulinus affinis Broderip in Broderip & Sowerby I, 1832b: 106.

Bulimus affinis; Hidalgo, 1870: 60.

Type locality. “in Peruviâ (Mexillones, desert of Atacama)”.

Type material. NHMUK 20100610 (5), possible syntypes.

Material examined. “Bolivia”, “(Cat. Am. mer. no. 162)”, Coll. Paz, MNCN 15.05/13170 (5), MNCN 15.05/13171 (5).

Remarks. According to Hidalgo (1870) the material was collected at “Paposo, Bolivia (Paz)”; this is in present-day Chile. The specimens appear to be partly subadult and juvenile.

Bostryx albicans (Broderip in Broderip and Sowerby I, 1832) [83]

(Fig. 26B)

Bulinus albicans Broderip in Broderip & Sowerby I, 1832b: 105.

Bulimus albicans; Hidalgo, 1870: 52; Hidalgo, 1872: 84; Hidalgo, 1875: 128; Hidalgo, 1893a: 100.

Type locality. “Copiapo, Chili”.

Type material. NHMUK 20100611 (5), possible syntypes.

Material examined.“Chile”, “(Cat. Am. mer. no. 111)”, Coll. Paz, MNCN 15.05/13162 (6); “Huasco (Chile)”, Coll. Hidalgo ex Martínez, MNCN 15.05/19967 (21); “Huasco, Chile”, Coll. Azpeitia, MNCN 15.05/8068 (5).

Remarks. The locality mentioned in Hidalgo (1870) is “Huasco, Chili (Paz et Martínez)”. In his 1872 publication Hidalgo treated this taxon as a variety (“Testa minor”) of Bulimus albus (=Bostryx erythrostomus; see below).

Bostryx anachoreta (Pfeiffer, 1856) [84]

(Fig. 26C)

Bulimus anachoreta Pfeiffer, 1856: 208; Hidalgo, 1870: 56.

Type locality. [Chile] “Paposo in desert Atacamensi reipublicae Chilensis”.

Type material. ZMB 112729 (2), syntypes.

Material examined. “Bolivia”, “(Cat. Am. mer. no. 141)”, Coll. Paz, MNCN 15.05/13173 (7); “Paposo”, Coll. Hidalgo, MNCN 15.05/19981 (2); Coll. Azpeitia, MNCN 15.05/8067 (5); “Paposo, Bolivia”, Coll. Azpeitia, MNCN 15.05/8445 (2).

Remarks. The material from the Hidalgo and Azpeitia collections are supposed to have been originated from the CCP-material. Lot MNCN 15.05/8445 is only tentatively referred to this species.

Bostryx atacamensis (Pfeiffer, 1856) [85]

(Fig. 26D)

Bulimus atacamensis Pfeiffer, 1856: 207; Hidalgo, 1870: 57; Hidalgo, 1872: 100; Hidalgo, 1875: 128, pl. 7 fig. 5; Hidalgo, 1893a: 105.

Type locality. [Chile] “Paposo in deserto Atacamensi reipublicae Chilensis”.

Type material. NHMUK 1975312, lectotype (Breure, 1978: 53).

Material examined. “Bolivia”, “(Cat. Am. mer. no. 142)”, Coll. Paz, MNCN 15.05/13093 (6).

Remarks. According to Hidalgo (1870) the material originated from “Paposo, Bolivie (Paz)”, which is in present-day Chile.

Bostryx bilineatus (Sowerby I, 1833) [86]

(Fig. 25B)

Bulinus bilineatus Sowerby I, 1833: 37.

Bulimus fontainei Orbigny; Hidalgo, 1872: 126; Hidalgo, 1875: 130; Hidalgo, 1893a: 119; Hidalgo, 1893b: 229.

Type locality. [Ecuador] “ad Sanctam Elena et in Columbiâ”.

Type material. ZMB 10261 (4), syntypes.

Material examined. “Guayaquil”, “(Cat. Am. mer. no. […])”, Coll. Paz, MNCN 15.05/13216 (8).

Remarks. This material is unicoloured and corresponds in this respect with Naesiotus fontainii (d’Orbigny, 1838) (Breure & Ablett, 2014: 78, fig. 16H), but have a protoconch sculpture of excessive fine, spiral lines, which classifies them as Bostryx. In one specimen a very faint light peripheral girdle may be discerned, which corresponds to Sowerby’s taxon (Köhler, 2007: fig. 21). Both Sowerby’s and d’Orbigny’s taxa have about the same shell height and may thus be easily misinterpreted.

Bostryx conspersus (Sowerby I, 1833) [87]

(Fig. 25C)

Bulinus conspersus Sowerby I, 1833: 67.

Bulimus conspersus; Hidalgo, 1870: 60; Hidalgo, 1872: 125

Type locality. [Peru] “collinis prope Lima”.

Type material. NHMUK 20100619 (5), probable syntypes.

Material examined.“Cerro de las Conchitas”, “(Cat. Am. mer. no. 160)”, Coll. Paz, MNCN 15.05/13178 (4), MNCN 15.05/13179 (4); “Lima”, “(Cat. Am. mer. no. 160)”, Coll. Paz, MNCN 15.05/13176 (5); “Lima”, Coll. Hidalgo “Paz”, MNCN 15.05/20329 (15).

Bostryx derelictus (Broderip in Broderip & Sowerby I, 1832) [88]

(Fig. 26F)

Bulinus derelictus Broderip in Broderip & Sowerby I, 1832b: 107.

Bulimus derelictus; Hidalgo, 1870: 53; Hidalgo, 1872: 88; Hidalgo, 1893a: 100; Hidalgo, 1893b: 262.

Type locality. “Cobijam Bolivia [now Chile] (Puerto del Mar)”.

Type material. NHMUK 20100609 (4), probable syntypes.

Material examined. “Cobija”, “(Cat. Am. mer. no. 114)”, Coll. Paz, MNCN 15.05/13083 (3); “Cobija”, Coll. Hidalgo ex Paz leg., MNCN 15.05/37159 (14); “Cobija, Bolivia”, Coll. Azpeitia, MNCN 15.05/9016 (7); “Pacifico 114”, Coll. Hidalgo ex Coll. Paz, MNCN 15.05/21314 (3).

Bostryx erythrostomus (Sowerby I, 1833) [89]

(Fig. 26G)

Bulinus erythrostoma Sowerby I, 1833: 37.

Bulimus albus Sowerby; Hidalgo, 1870: 54; Hidalgo, 1872: 83; Hidalgo, 1893a: 101.

Bulimus erythrostomus; Hidalgo, 1870: 54; Hidalgo, 1872: 85; Hidalgo, 1893a: 102.

Type locality. [Chile] “apud Huasco, Chilae”.

Type material. ZMB 10273 (2), ZMB 41572 (2), ZMB 114329 (1), probable syntypes.

Material examined.“Chile”, “(Cat. Am. mer. no. 119)”, Coll. Paz, MNCN 15.05/12996 (4); “Coquimbo”, “(Cat. Am. mer. no. 120)”, Coll. Paz, MNCN 15.05/13202 (5); “Coquimbo”, Coll. Hidalgo ex Martínez, MNCN 15.05/19964 (6); “Huasco”, Coll. Hidalgo ex Martínez, MNCN 15.05/19963 (11).

Remarks. Hidalgo (1870) mentioned as localities “Chamarcillo (Paz), Huasco et Coquimbo (Paz et Martínez), Chili”. In his 1872 publication, he only mentioned the two latter localities. The specimens which had been identified as Bulimus albus by Hidalgo, are entirely white both inside and outside (cf. Araya, 2015: fig. 5).

Bostryx hamiltoni ( Reeve, 1849) [90]

(Fig. 26E)

Bulimus hamiltoni Reeve, 1849 [1848–1850]: pl. 83 fig. 610.

Type locality. “Near the Lake of Titicaca, Bolivia”.

Type material. NHMUK 1849.5.14.53, lectotype (Breure, 1978: 80).

Material examined.“Puno en la Laguna de Chucuito o lago de Titicaca, Bolivia”, Coll. Azpeitia, MNCN 15.05/9029 (7).

Remarks. Although this material was not recognised by Hidalgo, and not mentioned in his papers, this material was supposedly collected by Almagro or Isern, who visited the area in July 1863 (Calatayud, 1994: 255–256).

Bostryx hennahi (Gray, 1828) [91]

(Fig. 26H)

Bulimus hennahi Gray, 1828: 5, pl. 5 fig. 5; Hidalgo, 1870: 52; Hidalgo, 1872: 87; Hidalgo, 1893a: 100; Hidalgo, 1893b: 270.

Type locality. [Chile] “Plains near Arica”.

Type material. Not located.

Material examined. “Peru”, “(Cat. Am. mer. no. 113)”, Coll. Paz, MNCN 15.05/12993 (4), MNCN 15.05/12992 (4); “Tacna (Perú)”, Coll. Hidalgo, MNCN 15.05/21236 (13); “Tacna (Perú)”, Coll. Azpeitia, MNCN 15.05/7207 (9).

Remarks. Hidalgo (1870) published as locality “Tacna, Pérou”. See also Calatayud, 1994: 258.

Bostryx holostoma (Pfeiffer, 1846) [92]

(Fig. 26I)

Bulimus holostoma Pfeiffer, 1846: 28; Hidalgo, 1870: 56; Hidalgo, 1893a: 104.

Type locality. [Chile] “Cobija, Bolivia”.

Type material. NHMUK 1975345, lectotype (Breure, 1979: 54).

Material examined.“Cobija”, “(Cat. Am. mer. no. 135)”, Coll. Paz, MNCN 15.05/14604 (3).

Figure 27 Material collected by the CCP.

(A–H) Bulimulidae. Bostryx lactifluus (Pfeiffer, 1857), MNCN 15.05/13089, (A) ventral view; Bostryx leucostictus (Philippi, 1856), MNCN 15.05/14540, (B) ventral view; Bostryx pupiformis (Broderip in Broderip & Sowerby I, 1832), MNCN 15.05/13190, (C) ventral view; Bostryx umbilicaris (Souleyet, 1842), MNCN 15.05/14515, (D) ventral view; Bostryx pustulosus (Broderip in Broderip & Sowerby, 1832), MNCN 15.05/14618, (E) ventral view; Bostryx rhodolarynx (Reeve, 1849), MNCN 15.05/3112, (F) ventral view; Bostryx mejillonensis (Pfeiffer in Pfeiffer & Dunker, 1857), MNCN 15.05/13141, (G) ventral view; Bostryx rouaulti (Hupé in Gay, 1854), MNCN 15.05/13309, (H) ventral view. Scale line 5 mm.

Remarks. Hidalgo (1870) mentioned this species from “Cobija, Bolivie (Paz)”; the locality is in present-day Chile.

Bostryx lactifluus (Pfeiffer, 1857) [93]

(Fig. 27A)

Bulimus lactifluus Pfeiffer, 1857: 330; Hidalgo, 1870: 56; Hidalgo, 1893a: 104.

Type locality. “Chili”.

Type material. NHMUK 20100642 (4), possible syntypes.

Material examined. “Cobija”, “(Cat. Am. mer. no. 137)”, Coll. Paz, MNCN 15.05/13089 (6); “Cobja, Bolivia”, Coll. Azpeitia, MNCN 15.05/8097 (4).

Remarks. The locality lies in present-day Chile.

Bostryx laurentii (Sowerby I, 1833) [94]

(Fig. 25G)

Bulinus laurentii Sowerby I, 1833: 37.

Bulimus laurentii; Hidalgo, 1870: 60; Hidalgo, 1893a: 109.

Type locality. [Peru] “Peruvia”.

Type material. Not located.

Material examined. “Lima”, “(Cat. Am. mer. no. 161)”, Coll. Paz, MNCN 15.05/13184 (6); “Cerro de las Conchitas”, “(Cat. Am. mer. no. 161)”, Coll. Paz, MNCN 15.05/13242 (6), MNCN 15.05/13243 (6); “I[sla]. San Lorenzo”, “(Cat. Am. mer. no. 161)”, Coll. Paz, MNCN 15.05/13404 (3).

Bostryx leucostictus (Philippi, 1856) [95]

(Fig. 27B)

Bulimus leucostictus Philippi, 1856: 53; Hidalgo, 1870: 56; Hidalgo, 1893a: 104.

Type locality. [Chile] “Paposo reipublicae Chilensis”.

Type material. Not located.

Material examined. “Atacama”, “(Cat. Am. mer. no. 139)”, Coll. Paz, MNCN 15.05/14540 (6).

Bostryx mejillonensis (Pfeiffer in Pfeiffer & Dunker, 1857) [96]

(Fig. 27G)

Bulimus mejillonensis Pfeiffer in Pfeiffer & Dunker, 1857: 230; Hidalgo, 1870: 52; Hidalgo, 1872: 83; Hidalgo, 1893a: 99; Hidalgo, 1893b: 232.

Type locality. [Chile] “Mejillones in desert Atacamensi”.

Type material. NHMUK 1975322, lectotype (Breure, 1978: 102).

Material examined. “Bolivia”, “(Cat. Am. mer. no. 110)”, Coll. Paz MNCN 15.05/13141 (4); “Paposo (Chile)”, Coll. Hidalgo ex Paz leg., MNCN 15.05/36319 (3).

Remarks. Hidalgo (1870) specified the locality as “Mejillones et Paposo, Bolivia (Paz)”; both places are in present-day Chile.

Bostryx modestus (Broderip in Broderip & Sowerby I, 1832) [97]

(Fig. 25E)

Bulimus modestus Broderip in Broderip & Sowerby I, 1832b: 106.

Bulimus modestus; Hidalgo, 1870: 53; Hidalgo, 1872: 90; Hidalgo, 1893a: 100; Hidalgo, 1893b: 280.

Bulimus limensis Reeve; Hidalgo, 1875: 130; Hidalgo, 1893a: 120.

Bulimus philippii Pfeiffer; Hidalgo, 1870: 53; Hidalgo, 1872: 89.

Bulimus scalariformis; Hidalgo, 1870: 54; Hidalgo, 1872: 91; Hidalgo, 1893a: 101; Hidalgo, 1893b: 281 [all in partim].

Type locality. “Peruviae montibus, Huacho”.

Type material. NHMUK 20120232 (4), possible syntypes.

Material examined.“Lima”, “(Cat. Am. mer. no. 115)”, Coll. Paz, MNCN 15.05/12997 (5); MNCN 15.05/13160 (4); MNCN 15.05/13163 (3); “Lima”, “(Cat. Am. mer. no. 118)”, Coll. Paz, MNCN 15.05/14616 (6); “Lima”, Coll. Hidalgo ex Martínez y Paz, MNCN 15.05/21228 (9); “Lima”, Coll. Hidalgo ex Paz leg., MNCN 15.05/20313 (5); “Lomas de Pumara, Lima”, Coll. Azpeitia, MNCN 15.05/8118 (9); “Lima”, Coll. Azpeitia, MNCN 15.05/76200 (17); Coll. Hidalgo, MNCN 15.05/20318 (5).

Remarks. Hidalgo (1870) distinguished Bulimus modestus, and B. philippii as a variety (under the same catalogue number), but synonymized the two taxa in his 1872 publication. These records were based on material from Paz and Martínez. The shells identified by him as B. scalariformis proved in part to exceed the size of the type material (see below), and resemble B. limensis Reeve, 1849. The systematic position follows Breure & Ablett (2014).

Bostryx nigropileatus ( Reeve, 1849) [98]

Bulimus nigropileatus Reeve, 1849 [1848–1850]: pl. 73 fig. 724 (text no. 725).

Bulimus stenacme Pfeiffer; Hidalgo, 1872: 131; Hidalgo, 1875: 130; Hidalgo, 1893a: 120; Hidalgo, 1893b: 279.

Type locality.“Chachapoyas, Alto-Peru”.

Type material. NHMUK 1975335, lectotype (Breure, 1978: 104).

Material examined. “Perú”, Coll. Paz, MNCN 15.05/14531 (1, subadult).

Remarks. Hidalgo (1872) gave as locality “Tarma”; the material was collected by Isern (see Calatayud, 1994: 257). One specimen was found in the RBINS (Dautzenberg coll., ex Crosse ex Hidalgo).

Bostryx pupiformis (Broderip in Broderip & Sowerby I, 1832) [99]

(Fig. 27C)

Bulinus pupiformis Broderip in Broderip & Sowerby I, 1832b: 105.

Bulimus pupiformis; Hidalgo, 1870: 56; Hidalgo, 1872: 99; Hidalgo, 1893a: 104; Hidalgo, 1893b: 284.

Type locality. “Chili (Huasco)”.

Type material. NHMUK 20100613 (4), probable syntypes.

Material examined. “Bolivia”, “(Cat. Am. mer. no. 142)”, Coll. Paz, MNCN 15.05/13090 (7); “Coquimbo”, “(Cat. Am. mer. no. 136)”, Coll. Paz, MNCN 15.05/13090 (5); “Huasco”, Coll. Hidalgo ex Paz and Martínez leg., MNCN 15.05/20221 (10); “Huasco, Chile”, Coll. Azpeitia, MNCN 15.05/8447 (7).

Remarks. The specimens of lot MNCN 15.05/13090 were found mixed with those of lot MNCN 15.05/13093 (B. atacamensis). Hidalgo mentioned them as a variety of this latter species, but in his 1872 publication he re-classified them as B. pupiformis.

Bostryx pustulosus (Broderip in Broderip & Sowerby, 1832) [100]

(Fig. 27E)

Bulinus pustulosus Broderip in Broderip & Sowerby I, 1832b: 105.

Bulimus pustulosus; Hidalgo, 1870: 53; Hidalgo, 1872: 90; Hidalgo, 1893a: 101; Hidalgo, 1893b: 293.

Type locality. “Chili (Huasco)”.

Type material. NHMUK 1975589 (5), probable syntypes.

Material examined. “Chile”, “(Cat. Am. mer. no. 116)”, Coll. Paz, MNCN 15.05/14618 (4); “Huasco”, Coll. Hidalgo, MNCN 15.05/19966 (50); “Huasco, Chile”, Coll. Azpeitia, MNCN 15.05/8448 (1).

Bostryx rhodolarynx ( Reeve, 1849) [101]

(Fig. 27F)

Bulimus rhodolarynx Reeve, 1849 [1848–1850]: pl. 72 fig. 518; Hidalgo, 1870: 47; Hidalgo, 1872: 73; Hidalgo, 1893a: 95.

Bulimulus (Scutalus) rhodolarynx; Hidalgo, 1893b: 257.

Type locality.[Peru] “Banks of the Aparimao [sic, Apurimac], Alto-Peru”.

Type material. NHMUK 1975434, lectotype; 1975435, paralectotype (Breure, 1978: 116).

Material examined. “Perú”, “(Cat. Am. mer. no. 89)”, Coll. Paz, MNCN 15.05/13669 (3); “Peru”, Coll. Hidalgo ex Almagro leg., MNCN 15.05/7342 (11); “Peru”, Coll. Azpeitia, MNCN 15.05/3112 (1).

Remarks. Hidalgo (1870) only gave the locality “Pérou”, but it is known that Almagro travelled through the region where this species occurs (Calatayud, 1994: 256).

Bostryx rouaulti (Hupé in Gay, 1854) [102]

(Fig. 27H)

Bulimus rouaulti Hupé in Gay, 1854: 110, pl. 3 fig. 8; Hidalgo, 1870: 54; Hidalgo, 1872: 86; Hidalgo, 1893a: 102; Hidalgo, 1893b: 269.

Type locality. [Chile] “Copiapó”.

Type material. MNHN-28119, lectotype (Breure, 1975: 1142).

Material examined. “Chile”, “(Cat. Am. mer. no. 121)”, Coll. Paz, MNCN 15.05/13309 (6); “Coquimbo”, Coll. Hidalgo ex Richardson “(regalado) [a gift]”; “Coquimbo, Chile”, Coll. Azpeitia, MNCN 15.05/8104 (5).

Remarks. According to Hidalgo (1870), the material of Paz was also collected at Coquimbo.

Bostryx scalariformis (Broderip in Broderip & Sowerby I, 1832) [103]

(Fig. 25F)

Bulinus scalariformis Broderip in Broderip & Sowerby I, 1832a: 31.

Bulimus scalariformis; Hidalgo, 1870: 54; Hidalgo, 1872: 91; Hidalgo, 1893a: 101; Hidalgo, 1893b: 281 [all in partim].

Type locality. [Peru] “in Peruviâ. (Ancon)”.

Type material. NHMUK 20100635 (5), NHMUK 20100636 (5), probable syntypes.

Material examined.“Lima”, “(Cat. Am. mer. no. 118)”, Coll. Paz, MNCN 15.05/13094 (8); “Peru”, Coll. Hidalgo ex Paz leg., MNCN 15.05/21318 (5).

Remarks. Only the smaller specimens from the series identified by Hidalgo seems to correspond with this species. However, the transition to Bostryx modestus (Broderip in Broderip & Sowerby, 1832) seem to be gradual and future research may prove these two taxa to be synonyms.

Bostryx tricinctus (Reeve, 1848) [104]

(Fig. 25D)

Bulimus tricinctus Reeve, 1848 [1848–1850]: pl. 57 fig. 380; Hidalgo, 1870: 62; Hidalgo, 1893a: 110.

Type locality. “.—?”.

Type material. NHMUK 1975182, lectotype (Breure, 1978: 132).

Material examined. “Huamachuco”, “(Cat. Am. mer. no. 166)”, Coll. Paz, MNCN 15.05/13321 (5).

Remarks. This species, described by Reeve from a shell without locality data, shows quite some variation in the colour pattern, which may have induced Hidalgo to synonymize this species from northern Peru with shells from Chile identified by him as Bulimus ferrugineus Reeve, 1849. This lot (MNCN 15.05/13312) probably has a wrong locality ([Chile] “Huasco”) and represent rather bleached specimens which are tentatively referred to Bostryx tricinctus.

Bostryx umbilicaris (Souleyet, 1842) [105]

(Fig. 27D)

Bulimus umbilicaris Souleyet, 1842: 102; Hidalgo, 1893a: 125.

Type locality. [Chile] “Bolivie, environs de Cobija”.

Type material. MNHN, lectotype (Breure, 1975: 1140).

Material examined. “Cobija”, Coll. Paz, MNCN 15.05/14515 (1).

Bostryx veruculum (Morelet, 1860) [106]

(Fig. 25H)

Bulimus veruculum Morelet, 1860: 376; Hidalgo, 1870: 56; Hidalgo, 1893a: 104.

Type locality. “Pérou, Ayacucho”.

Type material. MHNG-INVE-60384 (5), MHNG-INVE-60383 (5), syntypes.

Material examined. “Perú”, “(Cat. Am. mer. no. 140)”, Coll. Paz, MNCN 15.05/14526 (1).

Remarks. Hidalgo (1870) wrote “L’étiquette qui portrait la localité exacte de cette coquille a été égarée [the label that gave the exact locality of this shell was lost]”, implying that other material collected by the CCP did have those labels. As Paz did not travel in the region where this species occurs, but Almagro did (Calatayud, 1994: 256), it is supposed that he collected this specimen.

Genus Bulimulus Leach, 1814

Bulimulus Leach, 1814: 42.

Type species. Helix exilis Gmelin, 1791, by original designation.

Bulimulus apodemetes (d’Orbigny, 1835) [107]

(Fig. 28A)

Helix apodemeta d’Orbigny, 1835: 10.

Figure 28 Material collected by the CCP.

(A–K) Bulimulidae. Bulimulus apodemetes (d’Orbigny, 1835), MNCN 15.05/12990, (A) ventral view; Bulimulus bonariensis (Rafinesque, 1833), MNCN 15.05/21562, (B) ventral view; Bulimulus tenuissimus (Férussac in Férussac & Deshayes, 1832), MNCN 15.05/13204, (C) ventral view; Cochlorina aurismuris (Moricand, 1838), MNCN 15.05/13360, (D) ventral view; Cochlorina aurisleporis (Bruguière, 1792), MNCN 15.05/7181, (E) ventral view; Cochlorina navicula (Wagner, 1827), MNCN 15.05/13668, (F) ventral view; Drymaeus (Drymaeus) ambustus (Reeve, 1849), MNCN 15.05/21234, (G) ventral view; Drymaeus (Drymaeus) chenui (Philippi, 1867), MNCN 15.05/20236, (H) ventral view; Drymaeus (Drymaeus) baezensis (Hidalgo, 1869), MNCN 15.05/7354, (I) ventral view, (J) lateral view (lip), (K) dorsal view. Scale line 5 mm.

Bulimus apodemetes; Hidalgo, 1870: 52; Hidalgo, 1872: 85; Hidalgo, 1893a: 100; Hidalgo, 1893b: 252.

Type locality. “republica Argentina; republica Boliviana”; see Breure, 1973: 114.

Type material. NHMUK 1854.12.4.178–182 (28), syntypes.

Material examined. “Cordoba de Tucuman (Rep. Argentina)”, Coll. Hidalgo ex Paz, MNCN 15.05/20305 (17); “Pacifico”, Coll. Hidalgo, MNCN 15.05/36311 (7); “Cordoba [de Tucuman], Argentina”, Coll. Azpeitia, MNCN 15.05/8070 (2); “Republ. Argentina”, “(Cat. Am. mer. no. 112)”, Coll. Paz, MNCN 15.05/12990 (4).

Remarks. Breure & Ablett (2014) have placed this taxon in the genus Bostryx on account of the smooth protoconch of the type material. However, as it cannot be excluded that this material was worn, we have examined the large series of this species in the CCP-material, and additional non-CCP-material (MNCN 15.05/20306, 20308, Coll. Hidalgo). The protoconch sculpture shows some faint axial wrinkles, irregularly spaced and mostly on the lower part of the protoconch, only becoming more densely and prominent towards the transition to the teleoconch. This sculpture is unlike those observed in Caribbean Bulimulus species (Breure, 1974) nor in other Argentinan Bulimulus species, and is somewhat similar to those observed in some Peruvian Bostryx species (e.g., Breure, 1978). Further (molecular) studies should provide more evidence for the systematic position of this species. Awaiting this, and also for the stability of nomenclature, we tentatively concur with the recent review of Cuezzo, Miranda & Constanza Ovando (2013).

Bulimulus bonariensis (Rafinesque, 1833) [108]

(Fig. 28B)

Siphalomphix bonariensis Rafinesque, 1833: 165.

Bulimus montevidensis Pfeiffer; Hidalgo, 1870: 60; Hidalgo, 1875: 128; Hidalgo, 1893a: 108.

Bulimus sporadicus Orbigny; Hidalgo, 1872: 120; Hidalgo, 1893b: 273.

Type locality. “Buenos Ayres in South America”.

Type material. Not located.

Material examined.“Republ. Argentina”, “(Cat. Am. mer. no. 158 Montevidensis)”, Coll. Paz, MNCN 15.05/13156 (4), MNCN 15.05/13158 (4); “La Concordia Republica Argentina”, Coll. Hidalgo ex Paz, MNCN 15.05/20341 (1); “Rosario”, Coll. Hidalgo ex Martínez, MNCN 15.05/21562 (18); “Rosario, Argentina”, Coll. Azpeitia, MNCN 15.05/8098 (2); “Montevideo, Uruguay”, Coll. Azpeitia, MNCN 15.05/8125 (1); “Paysandu, Uruguay”, Coll. Azpeitia, MNCN 15.05/8126 (1).; “Pacifico 158”, MNCN 15.05/76232 (6).

Remarks. This species was mentioned from the following localities in Hidalgo (1870): “La Concordia et Montevideo (Paz), El Rosario, Rép. Argentine (Paz)”. Although the locality “Paysandu, Uruguay” was not mentioned, the specimen from the Azpeitia collection is tentatively also assigned to the CCP material. The systematic position follows Cuezzo, Miranda & Constanza Ovando (2013).

Bulimulus tenuissimus (Férussac in Férussac & Deshayes, 1832) [109]

(Fig. 28C)

Helix tenuissimus Férussac in Férussac & Deshayes, 1832 [1820–1851]: pl. 142B fig. 8.

Bulimus tenuissimus; Hidalgo, 1870: 60; Hidalgo, 1872: 119; Hidalgo, 1893a: 108; Hidalgo, 1893b: 228.

Type locality.“le Brésil et Cayenne”.

Type material. Not located.

Material examined. “Rio Janeiro”, “(Cat. Am. mer. no. 159)”, Coll. Paz, MNCN 15.05/13204 (4).

Genus Cochlorina Jan, 1830

Cochlorina Jan, 1830: 5.

Type species. Bulimus aurisleporis Bruguière, 1792, by subsequent designation (Bequaert, 1948: 190).

Cochlorina aurisleporis (Bruguière, 1792) [110]

(Fig. 28E)

Bulimus aurisleporis Bruguière, 1792: 346; Hidalgo, 1870: 46; Hidalgo, 1872: 70; Hidalgo, 1893a: 93; Hidalgo, 1893b: 190.

Type locality. “l’île de Madagascar [sic]”.

Type material. Not located.

Material examined. “Rio Janeiro”, “(Cat. Am. mer. no. 77)”, Coll. Paz, MNCN 15.05/13373 (4), 13383 (2); “Macahé (Brasil)”, Coll. Hidalgo ex “Martínez y Paz”, MNCN 15.05/37158 (4); “Macahé, Brasil”, Coll. Azpeitia, MNCN 15.05/7182 (2); “Pacifico”, Coll. Azpeitia, MNCN 15.05/7181 (1).

Remarks. This species was listed in the two versions of the catalogue of CCP material (Hidalgo, 1870; Hidalgo, 1893a), with locality data “Macahé, en el Brasil (Paz y Martínez)”.

Cochlorina aurismuris (Moricand, 1838) [111]

(Fig. 28D)

Helix (Cochlogena) aurismuris Moricand, 1838: 140, pl. 3 figs. 1–3.

Bulimus auris muris; Hidalgo, 1893a: 123.

Type locality. [Brazil] “la fazenda de Palmeirinha, entre Caxoeira et Jacobina, province de Bahia”.

Type material. MHNG-INVE-60683 (44), MHNG-INVE-60686 (48), syntypes

Material examined.“Brasil”, Coll. Paz, MNCN 15.05/13360 (1); Coll. Azpeitia, MNCN 15.05/8069 (1).

Remarks. Hidalgo (1870) did not mention this species in his initial catalogue, but in his final overview of the CCP material (Hidalgo, 1893a) the species is listed with locality “Bahia, en el Brasil (Paz)”.

Cochlorina navicula (Wagner, 1827) [112]

(Fig. 28F)

Helix navicula Wagner, 1827: 22.

Bulimus navicula; Hidalgo, 1893a: 123.

Type locality. [Brazil] “sylvis aboriginibus Provinciae Bahiensis”.

Type material. Not located.

Material examined. “Brasil”, Coll. Paz, MNCN 15.05/13668 (1); “Bahia, Brasil”, Coll. Azpeitia, MNCN 15.05/8107 (1).

Remarks. The species is only mentioned in the final version of the catalogue (Hidalgo, 1893a), with locality “Bahia, en el Brasil (Paz)”.

Genus Drymaeus Albers, 1850

Drymaeus Albers, 1850: 155.

Type species. Helix hygrohylaea d’Orbigny, 1835, by subsequent designation (Pilsbry 1898 [1897–1898]: 182).

Subgenus Drymaeus s.str.

Drymaeus (Drymaeus) ambustus ( Reeve, 1849) [113]

(Fig. 28G)

Bulimus ambustus Reeve, 1849 [1848–1850]: pl. 74 fig. 535; Hidalgo, 1870: 57; Hidalgo, 1872: 106; Hidalgo, 1893a: 105; Hidalgo, 1893b: 244.

Bulimus chamaeleon Pfeiffer; Hidalgo, 1870: 57; Hidalgo, 1872: 107; Hidalgo, 1893a: 106.

Type locality. “—?”.

Type material. NHMUK 1975441/1, lectotype (Breure & Eskens, 1981: 5).

Material examined. “La Mocha (Ecuador)”, Coll. Hidalgo ex Paz, MNCN 15.05/21234 (17); “Altipichi (Ecuador)”, Coll. Hidalgo ex Martínez, MNCN 15.05/21230 (9); “Baeza (Ecuador)”, Coll. Hidalgo ex Martínez, MNCN 15.05/20334 (1); “Nanegal”, Coll. Azpeitia, MNCN 15.05/8092 (2); “Ecuador”, “(Cat. Am. mer. no. 144)”, Coll. Paz, MNCN 15.05/13154 (4); “Ecuador”, Coll. Paz “(Cat. Am. mer. no. 145)”, MNCN 15.05/13201 (4); “Peru”, Coll. Hidalgo ex Almagro, MNCN 15.05/20340 (4) [probably a wrong locality label].

Remarks. Hidalgo (1870) mentioned material of Bulimus chamaeleon from “la Mocha (Paz)”, which may correspond to MNCN 15.05/13201. The material identified as this taxon is smaller than Bulimus ambustus. Baeza and La Mocha were visited by Almagro, Espada, Isern and Martinez, “Altipichi” was Alchipichi (visited by Martinez according to Calatayud, 1994: 265), Nanegal was not mentioned in their itinerary (Calatayud, 1994). “Peru” is likely a wrong locality as this species is not otherwise known from that country.

Drymaeus (Drymaeus) baezensis (Hidalgo, 1869) [114]

(Figs. 28I–28K)

Bulimus baezensis Hidalgo, 1869b: 189; Hidalgo, 1870: 48, pl. 1, fig. 3; Hidalgo, 1872: 75, pl. 7, figs. 11–12; Hidalgo, 1893a: 51; Hidalgo, 1893b: 96; Azpeitia, 1923: 72; Fischer-Piette, 1950: 74; Breure, 1975: 1149, pl. 1 fig. 2; Calvo, 1994: 284.

Type locality. “Baeza, reipublitae Aequatorius”.

Type material. MNHN, lectotype (Fischer-Piette, 1950: 74); “Baeza Ecuador”, “(Cat. Am. mer. no. 94)”, Coll. Paz ex Martínez leg., MNCN 15.05/3154 (3), MNCN 15.05/3155 (2); “Baeza (Ecuador)”, Coll. Hidalgo ex Martínez leg., MNCN 15.05/3205 (5), MNCN 15.05/3206 (7), paralectotypes.

Additional material examined. “Baeza, Ecuador”, “(Cat. Am. mer. no. 94)”, Coll. Paz, MNCN 15.05/7354 (2); Coll. Hidalgo “Bulimus Baezensis”, MNCN 15.05/8427 (1); “Baeza, Ecuador”, Coll. Azpeitia, MNCN 15.05/3156 (1).

Remarks. Lot MNCN 15.05/7354 was found identified as “Bulimus membielinus Crosse”. Lots MNCN 15.05/3156 and 8427 probably also originate from the CCP material, but are herein not considered as type material.

Drymaeus (Drymaeus) chanchamayensis (Hidalgo, 1870) [115]

(Figs. 29A–29C)

Figure 29 Material collected by the CCP.

(A–K) Bulimulidae. Drymaeus (Drymaeus) chanchamayensis (Hidalgo, 1870), MNCN 15.05/3157, (A) ventral view, (B) lateral view (lip), (C) dorsal view; Drymaeus (Drymaeus) chimborasensis (Reeve, 1848), MNCN 15.05/13426, (D) ventral view; Drymaeus (Drymaeus) chrysomelas (Martens, 1867), MNCN 15.05/13461, (E) ventral view; Drymaeus (Drymaeus) trujillensis (Philippi, 1867), MNCN 15.05/13679, (F) ventral view; Drymaeus (Drymaeus) membielinus (Crosse, 1867), MNCN 15.05/7355, (G) ventral view, (H) lateral view (lip), (I) lateral view (umbilicus), (J) dorsal view; Drymaeus (Drymaeus) fallax (Pfeiffer, 1853), MNCN 15.05/13148, (K) ventral view. Scale line 5 mm.

Bulimus chanchamayensis Hidalgo, 1870: 49; Hidalgo, 1893a: 72, 96; Azpeitia, 1923: 72; Calvo, 1994: 284.

Type locality. “Chanchamayo, Pérou”.

Type material. “Amazonas”, “(Cat. Am. mer. no. 98)”, Coll. Paz ex Isern leg., MNCN 15.05/3157 (1), holotype.

Remarks. Hidalgo (1870) introduced his species after having compared Pfeiffer, 1867 [1866–1869]: 348, pl. 82 figs. 6–7 (“Bulimus canaliculatus var.”) with Drymaeus (D.) canaliculatus (Pfeiffer, 1845) as figured by Reeve, 1848 [1848–1850]: pl. 41 fig. 256; this was the lectotype as re-figured by Breure & Ablett (2014: 37, figs. 38G–38I) Pfeiffer’s material was collected by Thamm “in regione Amazonien superiore” and was considered as holotype by Köhler (2007: 144, fig. 84), who considered Hidalgo’s taxon as a nomen novum. However, since Hidalgo gave as locality “Chanchamayo, Pérou”, and this material was collected by Isern (Calatayud, 1994: 257), we are certain that Hidalgo had material collected by the CCP at hand when introducing his taxon. Therefore the actual type material for Bulimus chanchamayensis Hidalgo, 1870 is not ZMB 11833 but MNCN 15.05/3157. Since Hidalgo wrote “dans l’exemplaire que j’ai sous les yeux”, we interpret this as referring to a singular specimen at hand; therefore the specimen MNCN 15.05/3157 is the holotype.

Drymaeus (Drymaeus) chenui (Philippi, 1867) [116]

(Fig. 28H)

Bulimus chenui Philippi, 1867: 72; Hidalgo, 1870: 58; Hidalgo, 1872: 113; Hidalgo, 1893a: 106; Hidalgo, 1893b: 249.

Type locality. [Peru] “Pachicamac probe Lima”.

Type material. Not located.

Material examined. “Pachacamac”, Coll. Hidalgo ex Isern, MNCN 15.05/20236 (3).

Drymaeus (Drymaeus) chimborasensis (Reeve, 1848) [117]

(Fig. 29D)

Bulimus chimborasensis Reeve, 1848 [1848–1850]: pl. 44 fig. 275.

Bulimus decoratus Lea; Hidalgo, 1870: 50.

Type locality. “Chimborazo, Columbia [sic, Ecuador], New Granada”.

Type material. NHMUK 1975460 (3), syntypes.

Material examined.“Ecuador”, “(Cat. Am. mer. no. 99)”, Coll. Paz, MNCN 15.05/13426 (1).

Remarks. This shell corresponds to the description of Bulimus chimborasensis Reeve, 1848, but shows a different colour pattern that reminds of B. decoratus Lea, 1838. However, this species was described from “near Carthagena” in northern Colombia. The two species seem nonetheless related.

Drymaeus (Drymaeus) chrysomelas (Martens, 1867) [118]

(Fig. 29E)

Bulimulus (Thaumastus) chrysomelas Martens, 1867: 145.

Bulimus chrysomelas; Hidalgo, 1870: 48; Hidalgo, 1893a: 95.

Type locality. [“oberes Amazonenstromgebiets”].

Type material. ZMB 11835a, lectotype (Köhler, 2007: 144).

Material examined.“Napo, Ecuador”, Coll. Paz “(Cat. Am. mer. no. 92)”, MNCN 15.05/13461 (1).

Remarks. Hidalgo (1870) indicated Martínez as collector. Compared to the lectotype (Köhler, 2007: fig. 85), the specimen has the inside of the aperture and the columella whitish. Martens (1867) did not mention a specific type locality other than in the title of his publication; according to Köhler the material was labelled “Peru, Chanchamayo”.

Drymaeus (Drymaeus) expansus (Pfeiffer, 1848) [119]

(Figs. 30A–30B)

Figure 30 Material collected by the CCP.

(A–H) Bulimulidae. Drymaeus (Drymaeus) expansus (Pfeiffer, 1848), MNCN 15.05/13480, (A) ventral view, (B) lateral view (lip); Drymaeus (Drymaeus) nystianus (Pfeiffer, 1853), MNCN 15.05/7331, (C) ventral view; Drymaeus (Drymaeus) papyraceus (Mawe, 1823), MNCN 15.05/13672, (D) ventral view; Drymaeus (Drymaeus) inaequalis (Pfeiffer, 1857), MNCN 15.05/7210, (E) ventral view; Kuschelenia (Bocourtia) aequatorius (Pfeiffer, 1853), MNCN 15.05/76211, (F) ventral view; Kuschelenia (Bocourtia) caliginosus (Reeve, 1849, MNCN 15.05/21231, (G) ventral view; Kuschelenia (Bocourtia) cf. culminea (d’Orbigny, 1835), MNCN 15.05/20238, (H) ventral view. Scale line 5 mm (all except E), 1 cm (E).

Bulimus expansus Pfeiffer, 1848b: 60; Hidalgo, 1870: 47; Hidalgo, 1872: 71; Hidalgo, 1893a: 91.

Type locality. [Peru] “Huallaga”.

Type material. Not located.

Material examined. “Peru”, “(Cat. Am. mer. no. 81)”, Coll. Paz, MNCN 15.05/13480 (1); “Canelos (Ecuador)”, Coll. Hidalgo ex Almagro leg., MNCN 15.05/37161 (1).

Remarks. Hidalgo (1870) gave only as locality “Canelos, Equator (Almagro)”. This specimen appears not to be full-grown. The locality “Peru” is somewhat doubtful, although Almagro has travelled through this country (Calatayud, 1994: 256).

Drymaeus (Drymaeus) fallax (Pfeiffer, 1853) [120]

(Fig. 29K)

Bulimus fallax Pfeiffer, 1853: 375; Hidalgo, 1870: 50;

Type locality. [Ecuador] “Tunguragua reipublicae Aequatoris”.

Type material. NHMUK 1969142, lectotype (Breure & Ablett, 2014: 72, figs. 26D–26F).

Material examined. “Quito”, “(Cat. Am. mer. no. 100)”, Coll. Paz, MNCN 15.05/13148 (4);

“Quito”, Coll. Hidalgo ex “Paz y Martínez”, MNCN 15.05/37054 (8);

“Quito, Ecuador”, Coll. Azpeitia, MNCN 15.05/8089 (1).

Drymaeus (Drymaeus) inaequalis (Pfeiffer, 1857) [121]

(Fig. 30E)

Bulimus inaequalis Pfeiffer, 1857: 330; Hidalgo, 1870: 48; Hidalgo, 1872: 74, pl 5 figs. 4–5; Hidalgo, 1893a: 96; Hidalgo, 1893b: 231.

Drymaeus inaequalis; Pilsbry 1897 [1897–1898]: 199, pl. 38 figs. 11–15.

Type locality. [Peru] “Banks of the Maranhon”.

Type material. Not located.

Material examined. “Ecuador”, “(Cat. Am. mer. no. 93)”, Coll. Paz, MNCN 15.05/3356 (2).

“Napo (Ecuador)”, “Bulimus hybridus/97./Pacifico”, ex Martínez, MNCN 15.05/7210 (3).

Remarks. The label referring to Bulimus hybridus probably was misplaced. Pilsbry translated the description which Hidalgo (1872) gave and copied his figures [shell actual height 43.5 mm].

Drymaeus (Drymaeus) membielinus (Crosse, 1867) [122]

(Figs. 29G–29J)

Bulimus membielinus Crosse, 1867: 445; Crosse, 1868: 99, pl. 1 fig. 2; Hidalgo, 1870: 47; Hidalgo, 1872: 72; Hidalgo, 1893a: 94; Hidalgo, 1893b: 232.

Type locality.“in Republica Aequatoris”.

Type material. “Ecuador”, Coll. Paz “(Cat. Am. mer. no. 82)”, MNCN 15.05/7355 (1), syntype; “Napo”, Coll. Hidalgo ex Martínez, MNCN 15.05/20344 (1), syntype.

Remarks. Crosse (1867) mentioned this species from “coll. Paz et Hidalgo”, but he did not mention on how many specimens his description was based. Hidalgo (1870) was the first to specify the locality to “Napo, Équateur (Martínez)”; this material (MNCN 15.05/20344) is more faded but still shows traces of a similar colour pattern.

Etymology. Named after Patricio Paz y Membiela.

Drymaeus (Drymaeus) nystianus (Pfeiffer, 1853) [123]

(Fig. 30C)

Bulimus nystianus Pfeiffer, 1853: 374; Hidalgo, 1870: 50; Hidalgo, 1872: 78; Hidalgo, 1893a: 97; Hidalgo, 1893b: 237.

Type locality. [Ecuador] “in valle Pomasqui reipublicae Aequatoris”.

Type material. NHMUK 1975573, lectotype (Breure, 1979: 112).

Material examined.“Machache, Ecuador”, “(Cat. Am. mer. no. 102)”, Coll. Paz, MNCN 15.05/13674 (3); MNCN 15.05/13675 (3); MNCN 15.05/13676 (3); “Machache (Ecuador)”, Coll. Hidalgo ex Paz, MNCN 15.05/21220 (18); “Quito”, Coll. Hidalgo ex Martínez leg., MNCN 15.05/37157 (5); “Bulimus nystianus”, Coll. Hidalgo, MNCN 15.05/21313 (1); Coll. Azpeitia, MNCN 15.05/7331 (5).

Remarks. Hidalgo (1870) gave as localities “Quito (Martínez), Machache, Équateur (Paz)”. The latter locality is a lapsus for Machachi (Calatayud, 1994: 268). The shells from the Azpeitia collection are the only ones with label “Quito”, this material may thus have originated from Martínez, although Azpeitia is known to have copied the published localities on his labels. This is a polymorphic species, which is not unusual in Drymaeus.

Drymaeus (Drymaeus) papyraceus (Mawe, 1823) [124]

(Fig. 30D)

Helix papyracea Mawe, 1823: 168, fig. 7.

Bulimus papyraceus; Hidalgo, 1870: 57; Hidalgo, 1872: 108; Hidalgo, 1893a: 105; Hidalgo, 1893b: 259.

Type locality.“Bahia, Brazil”.

Type material. Not located.

Material examined. “Brasil”, Coll. Paz “(Cat. Am. mer. no. 143)”, MNCN 15.05/13672 (3).; Coll. Hidalgo [ex “Martinex y Paz”], MNCN 15.05/39951 (4); “Bahia, Brasil”, Coll. Azpeitia, MNCN 15.05/13895 (1).

Remarks. Hidalgo (1870) gave as locality “Bahia (Paz et Martinez)”.

Drymaeus (Drymaeus) trujillensis (Philippi, 1867) [125]

(Fig. 29F)

Bulimus trujillensis Philippi, 1867: 73; Hidalgo, 1870: 48; Hidalgo, 1893a: 96.

Type locality. [Peru] “prope Trujillo”.

Type material. Not located.

Material examined. “Perú”, Coll. Paz “(Cat. Am. mer. no. 95)”, MNCN 15.05/13679 (1).

Remarks. According to Hidalgo (1870) this material was collected at “Huamachuco, Pérou (Paz)”. This locality is not listed in Calatayud (1994).

Drymaeus (Drymaeus) sp.

Material examined.“Guayaquil”, Coll. Hidalgo ex Martínez leg., MNCN 15.05/20333 (2).

“Chanchamayo”, Coll. Hidalgo ex Isern leg., MNCN 15.05/20346 (3).

Remarks. Both lots contain material that is too juvenile to be identified with certainty.

Genus Kuschelenia Hylton Scott, 1951

Kuschelenia Hylton Scott, 1951: 539.

Type species. Kuschelenia simulans Hylton Scott, 1951, by monotypy.

Subgenus Bocourtia Rochebrune, 1882

Bocourtia Rochebrune, 1882: 117.

Type species. Bocourtia lymnaeformis Rochebrune, 1882, by subsequent designation (Hubendick, 1951: 114).

Kuschelenia (Bocourtia) aequatorius (Pfeiffer, 1853) [126]

(Fig. 30F)

Bulimus aequatorius Pfeiffer, 1853: 420; Hidalgo, 1870: 59; Hidalgo, 1872: 104; Hidalgo, 1893a: 107; Hidalgo, 1893b: 273.

Type locality. [Ecuador] “reipublicae Aequatoris, monte Schinchulagua”.

Type material. NHMUK 1975377, lectotype (Breure, 1979: 85).

Material examined. “Quito”, “(Cat. Am. mer. no. 154)”, Coll. Paz, MNCN 15.05/76211 (3); “La Mocha (Ecuador)”, Coll. Hidalgo ex Paz, MNCN 15.05/20336 (6); “Quito (Ecuador)”, Coll. Hidalgo ex Martínez, MNCN 15.05/37155 (6); “Pacifico 154”, Coll. Hidalgo, MNCN 15.05/21271 (1); “Quito”, Coll. Azpeitia, MNCN 15.05/7178 (4); “Quito, Ecuador”, Coll. Azpeitia, MNCN 15.05/9014 (1).

Remarks. In both papers by Hidalgo (1870, 1872) this material is said to be from “Quito (Paz et Martínez); la Mocha, Équateur (Paz)”.

Kuschelenia (Bocourtia) caliginosus ( Reeve, 1849) [127]

(Fig. 30G)

Bulimus caliginosus Reeve, 1849 [1848–1850]: pl. 82 fig. 609; Hidalgo, 1870: 59; Hidalgo, 1893a: 108.

Type locality. “—?”.

Type material. NHMUK 20100518/1, lectotype (Breure & Ablett, 2014: 37)

Material examined.“Ecuador”, “(Cat. Am. mer. no. 156)”, Coll. Paz, MNCN 15.05/13468 (2); “156”, Coll. Hidalgo, MNCN 15.05/21231 (1).

Remarks. Hidalgo (1870) specified the locality as “Chimborazo, Équateur (Paz)”; this is the first confirmed locality as the type material was without locality data (Breure & Ablett, 2014).

Kuschelenia (Bocourtia) cotopaxiensis (Pfeiffer, 1853) [128]

(Fig. 31A)

Figure 31 Material collected by the CCP.

(A–G) Bulimulidae. Kuschelenia (Bocourtia) cotopaxiensis (Pfeiffer, 1853), MNCN 15.05/13142, (A) ventral view; Kuschelenia (Bocourtia) petiti (Pfeiffer, 1846), MNCN 15.05/13401, (B) ventral view; Kuschelenia (Kuschelenia) revinctus (Hupé, 1857), MNCN 15.05/76198, (C) ventral view; Kuschelenia (Kuschelenia) tupacii (d’Orbigny, 1835), MNCN 15.05/21241, (D) ventral view; Naesiotus quitensis (Pfeiffer, 1848), MNCN 15.05/13143, (E) ventral view; Neopetraeus lobbii (Reeve, 1849), MNCN 15.05/13464, (F) ventral view; Neopetraeus tessellatus (Shuttleworth, 1852), MNCN 15.05/13370, (G) ventral view. Scale 5 mm.

Bulimus cotopaxiensis Pfeiffer, 1853: 419; Hidalgo, 1870: 59; Hidalgo, 1872: 105; Hidalgo, 1893a: 107.

Type locality. “reipublicae Aequatoris, montem Cotopaxi”.

Type material. NHMUK 1975370, lectotype (Breure, 1978: 175, pl. 9 fig. 9).

Material examined.“Chimborazo”, “(Cat. Am. mer. no. 155)”, Coll. Paz, MNCN 15.05/13142 (3); “La Mocha”, “(Cat. Am. mer. no. 155)”, Coll. Paz, MNCN 15.05/13409 (2); “Antisana (Ecuador)”, Coll. Hidalgo ex Martínez, MNCN 15.05/37105 (11); “Pichincha”, Coll. Hidalgo ex Martínez, MNCN 15.05/20331 (1); “Ecuador”, Coll. Hidalgo ex Paz, MNCN 15.05/37103 (1); [Ecuador], Coll. Hidalgo, MNCN 15.05/21311 (3); “Ecuador”, coll. Azpeitia, MNCN 15.05/9015 (6).

Remarks. Hidalgo (1870) reported the material from “Quito (Paz et Martínez), La Mocha, Équateur (Paz)”. In Hidalgo (1872), he mentioned “Antisana y Pichincha (Martínez), La Mocha (Paz), en la Republic del Ecuador”. One of the specimens from lot MNCN 15.05/13409 is decidedly smaller and somewhat differently shaped, and only tentatively referred to this species. The largest specimen found (MNCN 15.05/20331), is somewhat bleached and worn.

Kuschelenia (Bocourtia) cf. culminea (d’Orbigny, 1835) [129]

(Fig. 30H)

Helix culminea d’Orbigny, 1835: 13.

Type locality. “culminibus Andesensibus, republica Boliviana” (see remarks).

Type material. MNHN, lectotype (Breure, 1975: 1143, pl. 1 fig. 3).

Material examined.“Peru”, Coll. Hidalgo ex Almagro leg., MNCN 15.05/20238 (2).

Remarks. This material had not been identified by Hidalgo and consequently not listed in his catalogues.

Kuschelenia (Bocourtia) petiti (Pfeiffer, 1846) [130]

(Fig. 31B)

Bulimus petiti Pfeiffer, 1846: 31; Hidalgo, 1870: 46; Hidalgo, 1893a: 92.

Type locality. “Peru”.

Type material. NHMUK 1975374, lectotype (Breure, 1978: 181).

Material examined.“Pataz, Peru”, “(Cat. Am. mer. no. 75)”, Coll. Paz MNCN 15.05/13401 (2).

Remarks. The locality of this species has been the topic of some confusion (see Breure & Ablett, 2014). This locality is not mentioned in the itinerary of the CCP (Calatayud, 1994), hence it is unclear who might have collected it.

Subgenus Kuschelenia s.str.

Kuschelenia (Kuschelenia) revinctus (Hupé, 1857) [131]

(Fig. 31C)

Bulimus revinctus Hupé, 1857: 39, pl. 7 fig. 2; Hidalgo, 1870: 58; Hidalgo, 1872: 112, pl. 5 fig. 6.

Type locality.“Pérou, Cuzco”.

Type material. MNHN 23256 (7), syntypes.

Material examined. “Peru”, Coll. Hidalgo ex Almagro leg., MNCN 15.05/76198 (1).

Kuschelenia (Kuschelenia) tupacii (d’Orbigny, 1835) [132]

(Fig. 31D)

Helix tupacii d’Orbigny, 1835: 16.

Bulimus tupacii; Hidalgo, 1893a: 125.

Type locality. “provincia Yungasensi (republica Boliviana)”; restricted to Dept. La Paz, Yanacachi (Breure, 1975).

Type material. MNHN 24710, lectotype (Breure, 1975: 1144, pl. 2 fig. 3).

Material examined.“Bul. Tupacii d’Orb. Chulumani Bolivie 2,500 m.”, “236 Pacifico”, Coll. Hidalgo, MNCN 15.05/21241 (2).

Remarks. Hidalgo (1893) gave as locality “República de Bolivia (Paz)”. The original label in Paz’s handwriting is an exceptional finding among the CCP material.

Genus Naesiotus Albers, 1850

Naesiotus Albers, 1850: 162.

Type species. Bulimus nux Broderip, 1832, by subsequent designation (Dall, 1896: 426).

Naesiotus quitensis (Pfeiffer, 1848) [133]

(Fig. 31E)

Bulimus quitensis Pfeiffer 1848: 230; Hidalgo, 1870: 63; Hidalgo, 1872: 130, pl. 7 figs. 5–8; Hidalgo, 1893a: 111; Hidalgo, 1893b: 263.

Bulimus irregularis Pfeiffer; Hidalgo, 1870: 63; Hidalgo, 1872: 129; Hidalgo, 1875: 128; Hidalgo, 1893a: 111.

Bulimus catloviae [sic, catlowiae] Pfeiffer var.; Hidalgo, 1872: 128, pl. 7 figs. 9–10; Hidalgo, 1893a: 112; Hidalgo, 1893b: 276.

Type locality. [Ecuador] “Quito”.

Type material. NHMUK 1893.2.4.198, lectotype (Breure, 1979: 71).

Material examined. “Pillaro”, “(Cat. Am. mer. no. 168 irregularis)”, Coll. Paz, MNCN 15.05/12999 (5); “Pillaro”, Coll. Hidalgo ex Martínez leg., MNCN 15.05/37051 (9); “Otavalo”, “(Cat. Am. mer. no. 168 irregularis)”, Coll. Paz, MNCN 15.05/13143 (4); “Ibarra”, “(Cat. Am. mer. no. 167)”, Coll. Paz, MNCN 15.05/13145 (4); “Ibarra”, Coll. Hidalgo ex Martínez leg., MNCN 15.05/20307 (11); “Pacifico 167”, Coll. Hidalgo, MNCN 15.05/20005 (8); “Pacifico 168”, Coll. Hidalgo, MNCN 15.05/20195 (7); “Pillaro, Ecuador”, Coll. Azpeitia, MNCN 15.05/8094 (2); “Quito, Ecuador”, Coll. Azpeitia, MNCN 15.05/13891 (1), MNCN 15.05/76210 (9).

Remarks. In Hidalgo (1870) material of Bulimus irregularis originated from “Ibarra, Otalvo et Pillaro, Équateur (Martínez)”; in Hidalgo (1872) only the lot from Pillaro was mentioned.

Genus Neopetraeus Martens, 1885

Neopetraeus Martens, 1885: 194.

Type species. Otostomus millegranus Martens, 1883, by subsequent designation (Pilsbry 1898 [1897–1898]: 163).

Neopetraeus lobbii (Reeve, 1849) [134]

(Fig. 31F)

Bulimus lobbii Reeve, 1849 [1848–1850]: pl. 72 fig. 516; Hidalgo, 1870: 48; Hidalgo, 1893a: 95.

Type locality. “Banks of the Maranon near Balsas, Peru”.

Type material. NHMUK 1975431, lectotype (Breure, 1978: 215, fig. 365).

Material examined.“Perú”, “(Cat. Am. mer. no. 91)”, Coll. Paz, MNCN 15.05/13464 (2);

“Pacifico 91”, Coll. Hidalgo, MNCN 15.05/21264 (1).

Remarks. Hidalgo (1870) gave as specific locality “Cajamarquilla, Pérou (Paz)”; this locality is not listed in the itinerary of the CCP and it thus unclear who might have collected this material.

Neopetraeus tessellatus (Shuttleworth, 1852) [135]

(Fig. 31G)

Bulimus tessellatus Shuttleworth, 1852: 200; Hidalgo, 1870: 61; Hidalgo, 1872: 123; Hidalgo, 1893a: 109; Hidalgo, 1893b: 235.

Bulimus cora d’Orbigny; Hidalgo, 1870: 48; Hidalgo, 1893a: 95.

Type locality. Not given.

Type material. NHMUK 1854.124.124, lectotype (Bulimus cora d’Orbigny; Breure & Ablett, 2014: 50, figs. 58A–58B).

Material examined. “Pataz, Perú”, “(Cat. Am. mer. no. 164)”, Coll. Paz, MNCN 15.05/13372 (2);

“Pataz, Peru”, Coll. Azpeitia, MNCN 15.05/8130 (2); “Haumalies, Perú”, “(Cat. Am. mer. no. 164)”, Coll. Paz, MNCN 15.05/13374 (2); “Sn. Mateo de Huaras”, “Pacifico 164”, Coll. Hidalgo, MNCN 15.05/7352 (7); “Pacifico 164”, Coll. Hidalgo, MNCN 15.05/21266 (3); “Peru”, “(Cat. Am. mer. no. 90)”, Coll. Paz, MNCN 15.05/13370 (2); “Huanuco”, “Pacifico 90”, Coll. Hidalgo ex Coll. Paz, MNCN 15.05/21221 (1).

Remarks. Hidalgo (1870) referred Bulimus cora d’Orbigny, 1835 to the locality “Huanuco, Peru (Paz)” and mentioned to have seen three specimens; for B. tessellatus he mentioned “San Mateo de Huaras (Almagro), Haumalies, Pataz, Pérou (Paz)”. In his 1872 publication he referred only to the material collected by Almagro at San Mateo de Huarás and did not mention the same Paz material explicitly. The shells from Almagro appear not to be full-grown and were referred to “var. Atahualpa, Dohrn”. The locality “Haumalies” is a province in the Huánuco Department. Both localities are not mentioned in the itinerary of the CCP (Calatayud, 1994), neither San Mateo de Huaras nor Pataz. Hence the provenance of this material remains unclear.

Genus Otostomus Beck, 1837

Otostomus Beck, 1837: 55.

Type species. Auris signata Spix in Wagner, 1827, by subsequent designation (Gray, 1847: 174).

Otostomus signatus (Spix in Wagner, 1827) [136]

(Fig. 32A)

Figure 32 Material collected by the CCP.

(A–F) Bulimulidae. Otostomus signatus (Spix in Wagner, 1827), MNCN 15.05/13371, (A) ventral view; Oxychona bifasciata (Burrow, 1815), MNCN 15.05/13128, (B) ventral view; Scutalus mutabilis (Broderip in Broderip & Sowerby I, 1832), MNCN 15.05/13382, (C) ventral view; Scutalus proteus (Broderip in Broderip & Sowerby I, 1832), MNCN 15.05/13390, (D) ventral view; Scutalus versicolor (Broderip in Broderip & Sowerby I, 1832), MNCN 15.05/13146, (E) ventral view; Stenostylus colmeiroi (Hidalgo, 1872), MNCN 15.05/3301, (F) ventral view. Scale line 5 mm.

Auris signata Spix in Wagner, 1827: 17, pl. 12 fig. 3.

Bulimus signatus; Hidalgo, 1870: 46; Hidalgo, 1893a: 93.

Type locality. [Brazil] “sylvis Provinciae Bahiensis”.

Type material. Not located.

Material examined. “Brasil”, “(Cat. Am. mer. no. 78)”, Coll. Paz, MNCN 15.05/13371 (2); “Brasil”, “(comprado)”, Coll. Hidalgo ex Paz, MNCN 15.05/7346 (1); “Brasil”, Coll. Azpeitia, MNCN 15.05/8115 (1).

Genus Oxychona Mörch, 1852

Oxychona Mörch, 1852: 14.

Type species. Trochus bifasciatus Burrow, 1815, by monotypy.

Oxychona bifasciata (Burrow, 1815) [137]

(Fig. 32B)

Trochus bifasciatus Burrow, 1815: 188, pl. 27 fig. 2.

Helix bifasciata; Hidalgo, 1870: 36; Hidalgo, 1872: 29, pl. 1 figs. 10–11; Hidalgo, 1893a: 84; Hidalgo, 1893b: 169.

Type locality. [Brazil] “Pernambuco”.

Type material. Not located.

Material examined. “Brasil”, “(Cat. Am. mer. no. 31)”, Coll. Paz, MNCN 15.05/13128 (4); “P-31”, Coll. Paz, MNCN 15.05/39929 (7); Coll. Hidalgo ex Paz “comprado”, MNCN 15.05/39931 (3); “Rio Janeiro”, Coll. Azpeitia, MNCN 15.05/39930 (3).

Genus Scutalus Albers, 1850

Scutalus Albers, 1850: 160.

Type species. Bulinus proteus Broderip, 1832, by subsequent designation (Martens in Albers, 1860: 217).

Scutalus mutabilis (Broderip in Broderip & Sowerby I, 1832) [138]

(Fig. 32C)

Bulinus mutabilis Broderip in Broderip & Sowerby I, 1832b: 108.

Bulimus mutabilis; Hidalgo, 1870: 47; Hidalgo, 1872: 110; Hidalgo, 1893a: 94.

Bulimus versicolor Broderip; Hidalgo, 1872: 110.

Type locality. [Peru] “in montibus Pervious (Santos)”.

Type material. Not located.

Material examined. “Lima”, “(Cat. Am. mer. no. 86)”, Coll. Paz, MNCN 15.05/13382 (4); “Lima”, Coll. Hidalgo ex Paz, MNCN 15.05/21270 (1).

Remarks. Hidalgo (1872) united this species with Scutalus versicolor (Broderip, 1832), likely on account of material collected by Martínez (see below). We regard S. mutabilis a distinct species, having the last whorl granose as seen with the naked eye, and in the material examined it is decidedly larger than the other species.

Scutalus proteus (Broderip in Broderip & Sowerby I, 1832) [139]

(Fig. 32D)

Bulinus proteus Broderip in Broderip & Sowerby I, 1832b: 107.

Bulimus proteus; Hidalgo, 1870: 55; Hidalgo, 1872: 109; Hidalgo, 1893a: 103; Hidalgo, 1893b: 258. [partim].

Type locality. [Peru] “Peruviae montibus (St. Jacinta, near Samanco)”.

Type material. NHMUK 20100638, lectotype (Breure & Ablett, 2014: 157, figs. 66A–66B).

Material examined. “Lima”, “(Cat. Am. mer. no. 127)”, Coll. Paz, MNCN 15.05/13375 (1); MNCN 15.05/13380 (2); MNCN 15.05/13381 (1); MNCN 15.05/13390 (6); Coll. Hidalgo, MNCN 15.05/36966 (13); “Lima”, Coll. Azpeitia, MNCN 15.05/8444 (2); MNCN 15.05/8449 (2 juv.); MNCN 15.05/8450 (3).

Remarks. Hidalgo (1870) mentioned this species from “Lima (Paz), Pachacamac (Isern)”.

Scutalus versicolor (Broderip in Broderip & Sowerby I, 1832) [140]

(Fig. 32E)

Bulinus versicolor Broderip in Broderip & Sowerby I, 1832b: 108.

Bulimus versicolor; Hidalgo, 1870: 47; Hidalgo, 1872: 110; Hidalgo, 1893a: 94; Hidalgo, 1893b: 260.

Type locality. “in montibus Peruviae (Mongon, near Casma)”.

Type material. NHMUK 1842.5.10.180–182 (4), NHMUK 20100637 (4), possible syntypes.

Material examined. “Lima”, “(Cat. Am. mer. no. 85)”, Coll. Paz, MNCN 15.05/13144 (3); MNCN 15.05/13146 (3); MNCN 15.05/13147 93); MNCN 15.05/13150 (3); MNCN 15.05/13151 (4); “Lima”, Coll. Hidalgo “Paz y Martínez”, MNCN 15.05/20335 (8); “85 var. Pacifico”, Coll. Hidalgo, MNCN 15.05/7357 (10); “Lima”, Coll. Azpeitia, MNCN 15.05/7329 (7); MNCN 15.05/13884 (18).

Remarks. Hidalgo ( 1872: 111) listed this species, which he considered synonymous with Scutalus mutabilis, from “Lima, Republic del Perú (Paz y Martínez)”; all material of S. mutabilis originated from Paz. This material, which may have reached the Azpeitia collection via Hidalgo, may thus have originated from Martínez. This species is smaller, with the last whorl seemingly smooth, but under the lens seen to be decussated and weakly granose (Pilsbry, 1897 [1897–1898]: 16).

Genus Stenostylus Pilsbry, 1898

Drymaeus (Stenostylus) Pilsbry, 1898 [1897–1898]: 184.

Type species. Bulimus nigrolimbatus Pfeiffer, 1854, by subsequent designation (Pilsbry 1898 [1897–1898]: 313).

Stenostylus colmeiroi ( Hidalgo, 1872) [141]

(Fig. 32F)

Bulimus colmeiroi Hidalgo, 1872: 122; Hidalgo, 1875: 129, pl. 7 fig. 3; Hidalgo, 1893a: 70, 119; Hidalgo, 1893b: 224; Azpeitia, 1923: 73; Fischer-Piette, 1950: 82; Breure, 1975: 1153, pl. 10 fig. 6; Calvo, 1994: 284.

Type locality. “Baeza, República del Ecuador”.

Type material. “Baeza (Ecuador)”, Coll. Hidalgo ex Martínez leg., MNCN 15.05/3301 (1), paralectotype; MNHN 20822 (1), lectotype (Fischer-Piette, 1950: 82).

Remarks. Hidalgo did not state on how many specimens his description was based. The measurements were given as “Long. 19, diam. 10 millim.”. The specimen in the MNCN measures H 19.6, D 10.8; it has 4.9 whorls. This corresponds nearly exactly with the measurements given by Hidalgo, while the specimen in the MNHN, considered as “holotype” by Fischer-Piette (1950: 82), has a shell height of 17 mm. From correspondence between Hidalgo and Crosse it is known that Hidalgo often donated material to Crosse (Breure & Backhuys, 2017).

Etymology. Named after Miguel Colmeiro y Penido (1816–1901), director of the Jardín Botánico in Madrid from 1868 to 1901, and co-founder and first President of the Sociedad española de Historia Natural.

Family Simpulopsidae Schileyko, 1999

Genus Leiostracus Albers, 1850

Leiostracus Albers, 1850: 156.

Type species. Bulimus vittatus Spix in Wagner, 1827, by subsequent designation (Martens in Albers, 1860: 213).

Leiostracus onager (Beck, 1837) [142]

(Fig. 33A)

Figure 33 Material collected by the CCP.

(A–H) Simpulopsidae. Leiostracus onager (Beck, 1837), MNCN 15.05/8135, (A) ventral view; Leiostracus perlucidus (Spix in Wagner, 1827), MNCN 15.05/13341, (B) ventral view; Leiostracus vimineus (Moricand, 1834), MNCN 15.05/12995, (C) ventral view; Leiostracus vittatus (Spix in Wagner, 1827), MNCN 15.05/20332, (D) ventral view; Rhinus heterotrichus (Moricand, 1836), MNCN 15.05/13485, (E) ventral view; Rhinus scobinatus (Wood, 1828), MNCN 15.05/8116, (F) ventral view; Simpulopsis rufovirens (Moricand, 1846), MNCN 15.05/20127, (G) ventral view; Simpulopsis sulculosa (Férussac, 1822), MNCN 15.05/20126, (H) ventral view. Scale line 5 mm.

Bulimulus onager Beck, 1837: 64.

Bulimus onager; Hidalgo, 1893a: 125.

Type locality. Not given.

Type material. Not located.

Material examined. “Bahia, Brasil”, Coll. Azpeitia, MNCN 15.05/8132 (1).

Remarks. Hidalgo (1893a) recorded as locality “Bahia, en el Brasil (Paz)”.

Leiostracus perlucidus (Spix in Wagner, 1827) [143]

(Fig. 33B)

Bulimus perlucidus Spix in Wagner, 1827: pl. 7 fig. 2; Hidalgo, 1870: 47; Hidalgo, 1893a: 95.

Type locality. “Brasilia”.

Type material. Not located.

Material examined. “Rio Janeiro”, “(Cat. Am. mer. no. 88)”, Coll. Paz, MNCN 15.05/13341 (1).

Leiostracus vimineus (Moricand, 1834) [144]

(Fig. 33C)

Helix (Cochlogena) viminea Moricand, 1834: 540, pl. 1 fig. 5.

Bulimus vimineus; Hidalgo, 1870: 59; Hidalgo, 1893a: 108.

Type locality. [Brazil] “le Brésil, dans la province de Bahia”.

Type material. MHNG-INVE-64563 (9), syntypes.

Material examined. “Brasil”, “(Cat. Am. mer. no. 157)”, Coll. Paz, MNCN 15.05/12995 (4); “Rio Janeiro”, Coll. Moricand, MNCN 15.05/20069 (1); “Rio Janeiro”, Coll. Azpeitia, MNCN 15.05/8131 (2); “Bahia, Brasil”, Coll. Azpeitia, MNCN 15.05/8455 (1).

Remarks. Hidalgo (1870) reported this species from “Rio Janeiro (Paz)”.

Leiostracus vittatus (Spix in Wagner, 1827) [145]

(Fig. 33D)

Bulimus vittatus Spix in Wagner, 1827: pl. 7 fig. 4; Hidalgo, 1870: 47; Hidalgo, 1893a: 95.

Type locality. [Brazil] “Provinciarum Bahiensis et Pernambucanae”.

Type material. Not located.

Material examined. “Rio Janeiro”, “(Cat. Am. mer. no. 87)”, Coll. Paz MNCN 15.05/13155 (3), MNCN 15.05/13157 (3); “Rio Janeiro”, “(comprado)”, Coll. Hidalgo, MNCN 15.05/20332 (1).

Remarks. Hidalgo (1893a) mentioned “Rio Janeiro et Bahia, en el Brasil (Paz)”.

Genus Rhinus Martens in Albers, 1860

Rhinus Martens in Albers, 1860: 223.

Type species. Bulimus heterotrichus Moricand, 1836, by original designation.

Rhinus heterotrichus (Moricand, 1836) [146]

(Fig. 33E)

Helix (Cochlogena) heterotricha Moricand, 1836: 430, pl. 2 figs. 5–6.

Bulimus heterotrichus; Hidalgo, 1870: 59; Hidalgo, 1893a: 107.

Type locality. Not given [Brazil, Bahia].

Type material. MHNG-INVE-64602 (6), syntypes.

Material examined.“Brazil, Corcovado”, (“Cat. Am. mer. no. 153”), Coll. Paz, 15.05/13485 (4); “Pacifico 113”, Coll. Hidalgo, MNCN 15.05/7566 (1).

Remarks. Hidalgo (1870) gave as locality “Corcobado, à Rio Janeiro (Paz)”.

Rhinus scobinatus (Wood, 1828) [147]

(Fig. 33F)

Bulimus scobinatus Wood, 1828: pl. 8 fig. 77; Hidalgo, 1875: 131; Hidalgo, 1893a: 120.

Type locality. “—”.

Type material. Not located.

Material examined. “Bahia, Brasil”, Coll. Azpeitia ex Paz leg., MNCN 15.05/8116 (1).

Remarks. Hidalgo (1893a) reported this species from “Bahia, en el Brasil (Paz)”.

Genus Simpulopsis Beck, 1837

Simpulopsis Beck, 1837: 100.

Type species. Helix sulculosa Férussac, 1821, by subsequent designation (Martens in Albers, 1860: 223).

Simpulopsis rufovirens (Moricand, 1846) [148]

(Fig. 33G)

Helix (Succinea) rufovirens Moricand, 1846: 147, pl. 5 fig. 4.

Simpulopsis rufovirens; Hidalgo, 1870: 30; Hidalgo, 1893a: 78.

Type locality. [Brazil] “le Brésil, dans la province de Bahia”.

Type material. MHNG-INVE-64632 (50+), MHNG-INVE-78493 (13), syntypes

Material examined. “Rio Janeiro”, Coll. Paz, MNCN 15.05/20127 (1).

Simpulopsis sulculosa (Férussac, 1822) [149]

(Fig. 33H)

Helix (Cochlohydra) sulculosa Férussac, in Férussac & Deshayes 1821 [1819–1841]: pl. 11A fig. 6; Férussac, 1822 [1821–1822]: 27.

Simpulopsis sulculosa; Hidalgo, 1870: 30; Hidalgo, 1872: 5; Hidalgo, 1893a: 78; Hidalgo, 1893b: 220.

Type locality. “Le Brésil”.

Type material. MNHN (2), syntypes.

Material examined. “Rio Janeiro”, Coll. Hidalgo ex “Martínez y Paz” leg., MNCN 15.05/39949 (2), MNCN 15.05/20126 (1), MNCN 15.05/11935 (5).

Remarks. Hidalgo (1872) gave “Botafogo, circa de Rio Janeiro” as a more precise locality; however, the original label stating this locality seems to have been lost.

Family Subulinidae Fischer & Crosse, 1877

Genus Leptinaria Beck, 1837

Achatina (Leptinaria) Beck, 1837: 79.

Type species. Helix unilamellata d’Orbigny, 1835, by subsequent designation (Hermannsen, 1847 [1846–1847]: 583).

Leptinaria anomala (Pfeiffer, 1846) [150]

(Fig. 34A)

Figure 34 Material collected by the CCP.

(A–H) Subulinidae. Leptinaria anomala (Pfeiffer, 1846), MNCN 15.05/20183, (A) ventral view; Leptinaria unilamellata (d’Orbigny, 1835), MNCN 15.05/20147, (B) ventral view; Stenogyra regularis (Pfeiffer, 1852), MNCN 15.05/39953, (C) ventral view; Subulina octona (Bruguière, 1792), MNCN 15.05/39954, (D) ventral view; Obeliscus haplostylus (Pfeiffer, 1846), MNCN 15.05/37048, (E) ventral view; Obeliscus cuneus riparius (Pfeiffer, 1854), MNCN 15.05/15511 (F) ventral view; Obeliscus obeliscus (Moricand, 1834), MNCN 15.05/15513, (G) ventral view; Neobeliscus calcareus (Born, 1778), MNCN 15.05/15512, (H) ventral view; Synapterpes auratus (Pfeiffer, 1846), MNCN 15.05/20330, (I) ventral view; Synapterpes visendus (Hidalgo, 1869), MNCN 15.05/3208, (J) ventral view. Scale line 1 mm (C), 1 cm (F–H), 5 mm (all others).

Achatina anomala Pfeiffer, 1846: 89.

Spiraxis anomala; Hidalgo, 1893a: 126.

Type locality. “Peru”.

Type material. Not located.

Material examined. “Pacifico”, Coll. Hidalgo, MNCN 15.05/20183 (1).

Leptinaria unilamellata (d’Orbigny, 1835) [151]

(Fig. 34B)

Helix (Cochlitomae) unilamellata d’Orbigny, 1835: 9.

Type locality. “provincia Santa Cruz de la Sierra (republica Boliviana)”.

Type material. NHMUK 1854.12.4.84 (6), syntypes.

Material examined. “Guyaquil”, ex Martínez, MNCN 15.05/20147 (4); “Guayaquil, Ecuador”, Coll. Azpeitia, MNCN 15.05/58992 (1).

Remarks. This was material not being identified by Hidalgo and therefore not listed in his catalogue.

Genus Neobeliscus Pilsbry, 1896

Neobeliscus Pilsbry, 1896: 46.

Type species. Helix calcareus Born, 1780, by original designation.

Neobeliscus calcareus (Born, 1778) [152]

(Fig. 34H)

Turbo calcareus Born, 1778: 351.

Bulimus calcareus; Hidalgo, 1870: 55; Hidalgo, 1893a: 105.

Type locality. Not given.

Type material. Not located.

Material examined. “Corcobado, Rio Jan.”, “(Cat. Am. mer. no. 131)”, Coll. Paz, MNCN 15.05/15512 (3); “Brasil (comprado)”, Coll. Hidalgo, MNCN 15.05/7190 (3); “Corcovado, Rio Janeiro, Brasil”, Coll. Azpeitia, MNCN 15.05/76199 (1).

Genus Obeliscus Beck, 1837

Obeliscus Beck, 1837: 61.

Type species. Helix (Cochlicella) obeliscus Moricand, 1834, by tautonymy.

Obeliscus cuneus riparius (Pfeiffer, 1854) [153]

(Fig. 34F)

Bulimus riparius Pfeiffer, 1854b: 155; Hidalgo, 1870: 55; Hidalgo, 1872: 98; Hidalgo, 1893a:104, Hidalgo, 1893b: 297.

Type locality. [Ecuador] “in ripis fluvii Mira, reipublicae Aequatoris”.

Type material. NHMUK 1987018 (3), syntypes.

Material examined.“Sn. José Ecuador”, “(Cat. Am. mer. no. 133)”, Coll. Paz, MNCN 15.05/15511 (4); “Baeza (Ecuador)”, Coll. Hidalgo ex Martínez leg., MNCN 15.05/37160 (12).

Remarks. Hidalgo (1870) recorded as localities “Baeza et San José, Équateur (Martínez)”.

Obeliscus haplostylus (Pfeiffer, 1846) [154]

(Fig. 34E)

Bulimus haplostylus Pfeiffer, 1846: 84; Hidalgo, 1872: 132; Hidalgo, 1875: 130; Hidalgo, 1893a: 119; Hidalgo, 1893b: 298.

Type locality. [Ecuador] “Loxa reipublicae Aequatoris”.

Type material. NHMUK 1987021 (1), probable syntype.

Material examined. “Cuenca (Ecuador)”, Coll. Hidalgo ex Martinex ex Jameson “(regalado)”, MNCN 15.05/37048 (3); “Ecuador”, Coll. Azpeitia, 15.05/76207 (1); Coll. Hidalgo, MNCN 15.05/76206 (1).

Remarks. This species was collected by James Jameson, who gave the material to Martinez (Calatayud, 1994: 207).

Obeliscus obeliscus (Moricand, 1834) [155]

(Fig. 34G)

Helix (Cochlicella) obeliscus Moricand, 1834: 540, pl. 1 fig. 4.

Bulimus obeliscus; Hidalgo, 1870: 55; Hidalgo, 1893a: 103.

Type locality. [Brazil] “Brésil, près de Caravelhas”.

Type material. MHNG-INVE-66256, holotype.

Material examined. “Bahia”, “(Cat. Am. mer. no. 132)”, Coll. Paz, MNCN 15.05/15513 (3); “Pacifico 132”, Coll. Hidalgo, MNCN 15.05/36384 (1); “Bahia, Brasil”, Coll. Azpeitia, MNCN 15.05/39947 (2).

Genus Stenogyra Shuttleworth, 1854

Stenogyra Shuttleworth, 1854: 45.

Type species. Bulimus terebraster Lamarck, 1822, by subsequent designation (Pilsbry in Pilsbry & Vanatta, 1899: 370).

Stenogyra regularis (Pfeiffer, 1852) [156]

(Fig. 34C)

Bulimus regularis Pfeiffer, 1852b: 94; Hidalgo, 1872: 123; Hidalgo, 1875: 130; Hidalgo, 1893a: 119; Hidalgo, 1893b: 299.

Type locality. [Brazil] “prope Rio Janeiro”.

Type material. Not known.

Material examined. “Sta. Catalina, Pacifo.”, Coll. Hidalgo ex Martínez leg., MNCN 15.05/39953 (8); “Rio Janeiro, Pacifo.”, Coll. Hidalgo ex Paz leg., MNCN 15.05/39952 (24).

Genus Subulina Beck, 1837

Subulina Beck, 1837: 76.

Type species. Bulimus octonus Bruguière, 1792, by subsequent designation (Gray, 1847: 178).

Subulina octona (Bruguière, 1792) [157]

(Fig. 34D)

Bulimus octonus Bruguière, 1792: 325.

Achatina octona Chemnitz; Hidalgo, 1875: 131; Hidalgo, 1893a: 121; Hidalgo, 1893b: 300.

Type locality. “l’île de Guadeloupe, & (…) l’île de Saint-Domingue”.

Type material. Not located.

Material examined. “Rio Janeiro”, Coll. Hidalgo ex “Martínez y Paz” leg., MNCN 15.05/39954 (5).

Genus Synapterpes Pilsbry, 1896

Synapterpes Pilsbry, 1896: 46.

Type species. Bulimus hanleyi Pfeiffer, 1846, by original designation.

Synapterpes auratus (Pfeiffer, 1846) [158]

(Fig. 34I)

Bulimus auratus Pfeiffer, 1846: 32; Hidalgo, 1870: 58.; Hidalgo, 1872: 100; Hidalgo, 1893a: 106; Hidalgo, 1893b: 246.

Type locality. “Locality unknown”.

Type material. NHMUK 1987019 (3), syntypes.

Material examined. “Ecuador”, “(Cat. Am. mer. no. 147)”, Coll. Paz, MNCN 15.05/13077 (2); “Baeza (Ecuador”, Coll. Hidalgo ex Martínez leg., MNCN 15.05/20330 (7).

Synapterpes visendus (Hidalgo, 1869) (comb. n.) [159]

(Fig. 34J)

Bulimus visendus Hidalgo, 1869a: 50, pl. 5, fig. 8; Hidalgo, 1870: 58; Hidalgo, 1872: 101, pl. 8 figs. 1–2; Hidalgo, 1893a: 47, 106; Hidalgo, 1893b: 247; Azpeitia, 1923: 74; Breure, 1975: 1153, pl. 1 fig. 5; Calvo, 1994: 284.

Type locality. [Ecuador] “Baeza, Reipublicae Aequatoris”.

Type material. “Baeza, Ecuador”, ex Hidalgo, MNHN-IM-2000-28157, lectotype (Breure, 1975: 1153). “Baeza, Ecuador”, “(Cat. Am. mer. no. 148)”, Coll. Paz, MNCN 15.05/3163 (2); “Baeza (Ecuador)”, Coll. Hidalgo ex Martínez leg., MNCN 15.05/3208 (4); “Baeza, Ecuador”, Coll. Hidalgo, MNCN 15.05/3207 (1); “Baeza, Ecuador”, Coll. Azpeitia, MNCN 15.05/76230 (1), paralectotypes.

Remarks. This taxon has long been considered a Drymaeus (Mesembrinus) species due to misinterpretation of the general shape. Re-studying of the type material including the MNCN-specimens has convinced us that this species belongs to the genus Synapterpes (comb. n.). The protoconch is smooth in all specimens (contrary to the pitted protoconch in Drymaeus), and the combination of the shell shape, shell size, and the colour pattern corresponds with other species of the genus Synapterpes.

Family Spiraxidae H.B. Baker, 1939

Genus Euglandina Crosse & P. Fischer in Fischer & Crosse, 1870

Euglandina Crosse & P. Fischer in P. Fischer & Crosse, 1870 [1870–1878]: 97.

Type species. Achatina aurata var. lignaria Reeve, 1849, by subsequent designation (Pilsbry, 1907 [1906–1907]: 175).

Subgenus Euglandina (Cosmomenus) Baker, 1941

Euglandina (Cosmomenus) Baker, 1941: 54.

Type species. Glandina cumingi Beck, 1837, by original designation.

Euglandina (Cosmomenus) cumingi (Beck, 1837) [160]

(Fig. 35D)

Figure 35 Material collected by the CCP.

(A) Spriraxidae. Euglandina (Cosmomenus) cumingi (Beck, 1837), MNCN 15.05/76219, (A) ventral view. (B–J) Streptaxidae. Hypselartemon deshayesianus (Crosse, 1863), MNCN 15.05/19843, (B) ventral view, (C) umbilical view, (D) apical view; Hypselartemon paivanus (Pfeiffer, 1867), MNCN 15.05/20124, (E) ventral view, (F) umbilical view, (G) apical view; Rectartemon candidus (Spix in Wagner, 1827), MNCN 15.05/20123, (H) ventral view, (I) umbilical view, (J) apical view. Scale line 5 mm.

Glandina cumingi Beck, 1837: 78.

Glandina rosea Férussac; Hidalgo, 1893a: 126.

Type locality. Not stated.

Type material. Not located.

Material examined.“Panamá”, Coll. Azpeitia, MNCN 15.05/76219 (5).

Remarks. Hidalgo (1893a) identified this species as “Glandina rosea Férussac”, reporting it from “Panamá, en Colombia (Paz)”. This country was visited by Amor, Espada and Martinez; this material was likely collected by the latter (see Calatayud, 1994: 259).

Family Streptaxidae Gray, 1860

GenusHypselartemon Wenz, 1947

Hypselartemon Wenz, 1947: 36.

Type species. Streptaxis alveus Dunker, 1845, by original designation.

Hypselartemon deshayesianus (Crosse, 1863) [161]

(Figs. 35B–35D)

Streptaxis deshayesianus Crosse, 1863: 388; Hidalgo, 1870: 39; Hidalgo, 1872: 45, pl. 3 figs. 5–6; Hidalgo, 1893a: 87; Hidalgo, 1893b: 142.

Type locality. “?”.

Type material. Not located.

Material examined. “Rio Janeiro”, Coll. Hidalgo ex “Martínez y Paz”, MNCN 15.05/20106 (62); “Pacifico 51”, Coll. Hidalgo, MNCN 15.05/19843 (7); “Rio Janeiro, Brasil”, Coll. Azpeitia, MNCN 15.05/39945 (4).

Hypselartemon paivanus (Pfeiffer, 1867) [162]

(Figs. 35E–35F)

Streptaxis paivana Pfeiffer, 1867 [1866–1869]: 43, pl. 1 fig. 2; Hidalgo, 1870: 39; Hidalgo, 1872: 44, pl. 3 figs. 3–4; Hidalgo, 1893a: 87; Hidalgo, 1893b: 135.

Type locality. [Brasil] “in Brasilia loco “Macahe” dicto”.

Type material. Not located.

Material examined. “Macahé (Brazil)”, Coll. Hidalgo ex “Paz y Martínez”, MNCN 15.05/20103 (26); Coll. Hidalgo, MNCN 15.05/20124 (3); “Macahe, Brasil”, Coll. Azpeitia, MNCN 15.05/39948 (3).

Remarks. Crosse stated (footnote in Pfeiffer, 1867: 43) that this material originated from Paz. We are therefore confident that the material listed above may be considered as from the original series.

Genus Rectartemon Baker, 1925

Rectartemon Baker, 1925: 36.

Type species. Rectartemon jessei Baker, 1925, by original designation.

Rectartemon candidus (Spix in Wagner, 1827) [163]

(Figs. 35H–35J)

Solarium candidum Spix in Wagner, 1827: pl. 17 figs. 3–4.

Streptaxis candidus Spix; Hidalgo, 1870: 40; Hidalgo, 1872: 42; Hidalgo, 1893a: 88; Hidalgo, 1893b: 134.

Type locality. [Brazil] “Provinciis autralioribis Brasiliae”.

Type material. Not located.

Material examined. “P-52”, Coll. Hidalgo ex “Martínez y Paz”, MNCN 15.05/20123 (1); “P-46”, Coll. Hidalgo, MNCN 15.05/76202 (1) [ex-MNCN 15.05/20117].

Remarks. Hidalgo (1870) mentioned this species from “Desterro, île de Sainte-Catharine, Brésil (Paz et Martinez); Rio Grande, Brésil (Paz)”. See Calatayud, 1994: 250–251.

GenusStreptaxis Gray, 1837

Streptaxis Gray, 1837: 484.

Type species. Helix (Helicogena) contusa Férussac, 1821, by subsequent designation (Herrmannsen, 1849 [1847–1849]: 507).

Streptaxis contusus (Férussac, 1821) [164]

(Figs. 36A–36C)

Figure 36 Material collected by the CCP.

(A–L) Streptaxidae. Streptaxis contusus (Férussac, 1821), MNCN 15.05/20178, (A) ventral view, (B) umbilical view, (C) apical view; Streptaxis crossei (Pfeiffer, 1867), MNCN 15.05/20177, (D) ventral view, (E) umbilical view, (F) apical view; Streptaxis dunkeri (Pfeiffer in Philippi, 1845), MNCN 15.05/20117, (G) ventral view, (H) umbilical view, (I) apical view; Streptaxis uberiformis (Pfeiffer, 1848), MNCN 15.05/20125, (J) ventral view, (K) umbilical view, (L) apical view. Scale line 5 mm.

Helix (Helicogena) contusa Férussac, 1821 [ 1821–1822]: 30; Férussac in Férussac & Deshayes 1821 [1819–1851]: pl. 31 fig. 1, pl. 36A figs. 2–3.

Streptaxis contusus; Hidalgo, 1870: 39; Hidalgo, 1872: 41; Hidalgo, 1893a: 87.

Type locality. “Le Brésil”.

Type material. Not located.

Material examined.“Pacifico 47”, Coll. Hidalgo ex “Martínez y Paz”, MNCN 15.05/20178 (1); “Rio Janeiro”, Coll. Hidalgo ex Martínez leg., MNCN 15.05/20102 (4); “Botofogo, Corcovado, Rio Janeiro, Brasil”, Coll. Azpeitia, MNCN 15.05/36262 (3).

Remarks. Hidalgo (1870) mentioned this species from “Corcobado, à Rio Janeiro (Paz et Martinez)”.

Streptaxis crossei Pfeiffer, 1867 [165]

(Figs. 36D–36F)

Streptaxis crossei Pfeiffer, 1867: 43, pl. 1 fig.1; Hidalgo, 1870: 39; Hidalgo, 1872: 43, pl. 3 figs. 1–2; Hidalgo, 1893a: 87; Hidalgo, 1893b: 139.

Type locality. [Brazil] “Corcobado, props Rio Janeiro Brasiliae”.

Type material. Not located.

Material examined.“Rio Janeiro”, Coll. Hidalgo ex “Martinez y Paz”, MNCN 15.05/20104 (15); Coll. Hidalgo ex “Martinex y Paz”, MNCN 15.05/20177 (5); “Botofogo, Corcovado, Rio Janeiro”, Coll. Azpeitia, MNCN 15.05/39944 (3), MNCN 15.05/76228 (2).

Remarks. Crosse stated (footnote in Pfeiffer, 1867: 43) that this material originated from Paz. We are therefore confident that the material listed above may be considered as from the original series. Hidalgo (1870) reported this species from “Macahé, près de Rio Janeiro (Paz et Martinez)”; Hidalgo (1893a) from “Botafou, en el Cordovado, cerca de Rio Janeiro (Paz y Martinez)”.

Etymology. Named after Hippolyte Crosse.

Streptaxis dunkeri Pfeiffer in Philippi, 1845 [166]

(Figs. 36G–36I)

Streptaxis dunkeri Pfeiffer in Philippi 1845 [1845–1847]: 7, pl. 6 fig. 15; Hidalgo, 1870: 39; Hidalgo, 1893a: 89.

Type locality. “Brasilia, prope Neu-Freiburg”.

Type material. Not located.

Material examined. “P-46”, Coll. Hidalgo MNCN 15.05/20117 (1); “Corcovado, Rio Janeiro, Brasil”, Coll. Azpeitia, MNCN 15.05/39946 (1).

Streptaxis uberiformis Pfeiffer, 1848 [167]

(Figs. 36J–36L)

Streptaxis uberiformis Pfeiffer, 1848a: 89; Hidalgo, 1870: 39; Hidalgo, 1872: 42; Hidalgo, 1875: 128, pl. 7 fig. 8; Hidalgo, 1893a: 87.

Type locality. “Brasilia”.

Type material. NHMUK 20160371 (1), syntype.

Material examined. MNCN 15.05/20125 (1).

Remarks. Although this material has no label stating its locality nor provenance, there is a label in Crosse’s handwriting “No. 7 / Streptaxis uberiformis Pfeiffer / type figure dans le Journal de / Conchyliologie, vol. XXIII”; the specimen corresponds to Hidalgo, 1875: pl. 7 fig. 8.

Family Macrocyclidae Thiele, 1926

Genus Macrocyclus Beck, 1837

Helix (Macrocyclus) Beck, 1837: 24.

Type species. Helix laxata Férussac, 1821 (=Helix peruviana Lamarck, 1822), by subsequent designation (Albers, 1850: 128).

Macrocyclus peruvianus (Lamarck, 1822) [168]

Helix (Helicella) laxata Férussac, 1821 [1821–1822]: 39 (nomen nudum).

Helix peruviana Lamarck, 1822: 76.

Helix laxata; Hidalgo, 1870: 33; Hidalgo, 1893a: 82.

Type locality. “le Pérou”.

Type material. Not located.

Material examined. “Pacifico 22”, Coll. Hidalgo, MNCN 15.05/76221 (1).

Remarks. The shell has been broken due to its fragility and is therefore not photographed.

Family Strophocheilidae Pilsbry, 1902

Genus Anthinus Albers, 1850

Bulimus (Anthinus) Albers, 1850: 148.

Type species. Helix (Cochlogena) multicolor Rang, 1831, by subsequent designation (Martens in Albers, 1860: 189).

Figure 37 Material collected by the CCP.

(A–G) Strophocheilidae. Anthinus multicolor (Rang, 1831), MNCN 15.05/13268, (A) ventral view; Austroborus lutescens (King & Broderip, 1831), MNCN 15.05/13671, (B) ventral view; Chiliborus chilensis (Sowerby I, 1833), MNCN 15.05/13479, (C) ventral view; Chiliborus rosaceus (King & Broderip, 1831), MNCN 15.05/13269, (D) ventral view; Gonyostomus egregius (Pfeiffer, 1845), MNCN 15.05/13368, (E) ventral view; Gonyostomus goniostomus (Férussac, 1821), MNCN 15.05/13369, (F) ventral view; Speironepion milleri (Sowerby I in Sowerby I & II, 1838), MNCN 15.05/13298, (G) ventral view. Scale line 5 mm (A–F), 1 cm (G).

Anthinus multicolor (Rang, 1831) [169]

(Fig. 37A)

Helix (Cochlogena) multicolor Rang, 1831: 55, pl. 3 fig. 1.

Bulimus multicolor; Hidalgo, 1870: 47; Hidalgo, 1893a: 94.

Bulimus miersi Sowerby; Hidalgo, 1870: 47; Hidalgo, 1893a: 94.

Type locality. “Brésil, non loin du Corcovado”.

Type material. Not located.

Material examined. “Brasil”, “(Cat. Am. mer. no. 83)”, Coll. Paz, MNCN 15.05/13458 (3); “Brasil”, “(Cat. Am. mer. no. 84)”, Coll. Paz MNCN 15.05/13268 (2) [as Bulimus miersi Sow.]; “(comprado)”, Coll. Hidalgo, MNCN 15.05/7326 (4) [as Bulimus miersi Sow.]; “Rio Janeiro”, “(comprado)”, Coll. Hidalgo, MNCN 15.05/37104 (1); “83”, Coll. Hidalgo, MNCN 15.05/21261 (1); “Macahé, Brasil”, Coll. Azpeitia, MNCN 15.05/8100 (1); “Rio Janeiro, Brasil”, Coll. Azpeitia, MNCN 15.05/8102 (1).

Remarks. This species was listed in Hidalgo’s catalogue (1870) as number 83 [multicolor] from “Macahé, Brésil”, respectively number 84 [miersi] from “Rio Janeiro”; both localities were credited to Paz. The identification by Hidalgo of part of the material as “Bulimus miersi” was erroneous.

Genus Austroborus Parodiz, 1949

Strophocheilus (Austroborus) Parodiz, 1949: 189. Nom. nov. for Microborus Pilsbry, 1926 not Blanford, 1897.

Type species. Bulimus lutescens King & Broderip, 1831, by original designation.

Austroborus lutescens (King & Broderip, 1831) [170]

(Fig. 37B)

Bulinus lutescens King & Broderip, 1831: 340.

Bulimus lutescens; Hidalgo, 1870: 43; Hidalgo, 1872: 55; Hidalgo, 1893a: 89.

Type locality. “Maldonado”.

Type material. NHMUK 20160373 (5), syntypes.

Material examined. “Montevideo”, “(Cat. Am. mer. no. 61)”, Coll. Paz, MNCN 15.05/13671 (3); “Pacifico 61”, Coll. Hidalgo, MNCN 15.05/21222 (1).

Remarks. Hidalgo (1870) mentioned “Montevideo (Paz et Martínez)”; it is therefore possible that the single shell from the Hidalgo collection originated from Martínez.

Genus Chiliborus Pilsbry, 1926

Borus (Chiliborus) Pilsbry, 1926: 6.

Type species. Bulinus chilensis Sowerby I, 1833, by original designation.

Chiliborus chilensis (Sowerby I, 1833) [171]

(Fig. 37C)

Bulinus chilensis Sowerby I, 1833: 36.

Bulimus crenulatus Pfeiffer; Hidalgo, 1870: 43; Hidalgo, 1872: 54; Hidalgo, 1893a: 89; Hidalgo, 1893b: 200.

Type locality.[Chile] “Coquimbo”.

Type material. Not located.

Material examined.“Talcahuano”, “(Cat. Am. mer. no. 60)”, Coll. Paz, MNCN 15.05/13479 (1); “Huasco”, “(Cat. Am. mer. no. 60)”, Coll. Paz, MNCN 15.05/13478 (2); “Huasco Martínez ”, “Huasco (Chile)”, Coll. Hidalgo ex Martínez, MNCN 15.05/20206 (17).

Remarks. Hidalgo mentioned as localities “Talcahuano et Coquimbo (Paz), Huasco (Paz et Martínez), Chili”. The locality ‘Talcahuano’ was no longer mentioned in Hidalgo (1872). The largest specimen in the material is from this locality.

Chiliborus rosaceus (King & Broderip, 1831) [172]

(Fig. 37D)

Bulinus rosaceus King & Broderip, 1831: 341.

Bulimus rosaceus; Hidalgo, 1870: 43; Hidalgo, 1872: 53; Hidalgo, 1893a: 89; Hidalgo, 1893b: 198.

Type locality. “ad oras Americae meridionalis (Chile)”.

Type material. Not located.

Material examined.“Valparaiso”, “(Cat. Am. mer. no. 59)”, Coll. Paz, MNCN 15.05/13269 (4); “Valparaiso”, “Pacifico 59”, Coll. Hidalgo, ex “Paz y Martinez”, MNCN 15.05/36925 (9); “Huasco ! Paz”, Coll. Azpeitia ex Paz leg., MNCN 15.05/7344 (1); “Chile”, “(Cat. Am. mer. no. 59)”, Coll. Azpeitia, MNCN 15.05/8113 (4).

Remarks. Hidalgo mentioned as locality “Valparaiso (Paz et Martínez)”.

Genus Gonyostomus Beck, 1837

Bulimus (Gonyostomus) Beck, 1837: 53.

Type species.Helix (Cochlogena) goniostoma Férussac, 1821, by tautonymy.

Gonyostomus egregius (Pfeiffer, 1845) [173]

(Fig. 37E)

Bulimus egregius Pfeiffer, 1845a: 67; Hidalgo, 1893a: 122.

Type locality.“Locality unknown”.

Type material. NHMUK 19991589 (3), syntypes.

Material examined. “Cabo Frio, Rio Jan[eiro].”, “(Cat. Am. mer. no. 97)”, Coll. Paz, MNCN 15.05/13368 (4) [as Bulimus hybridus Gould]; “Macahé, Brasil”, Coll. Azpeitia, MNCN 15.05/7208 (1).

Gonyostomus goniostomus (Férussac, 1821) [174]

(Fig. 37F)

Helix (Cochlogena) goniostoma Férussac, 1821 [ 1821–1822]: 57.

Bulimus goniostomus; Hidalgo, 1870: 49; Hidalgo, 1872: 77; Hidalgo, 1893a: 96.

Type locality. “Le Brésil, près Rio Janeiro, à l’aqueduc de Corcovado”.

Type material. Not located.

Material examined. “Cabo Frio, Brasil”, “(Cat. Am. mer. no. 96)”, Coll. Paz, MNCN 15.05/13369 (4); “(comprado)”, Coll. Hidalgo, MNCN 15.05/7205 (2); “Macahé, Brasil”, Coll. Azpeitia, MNCN 15.05/8110 (2).

Remarks. Hidalgo (1870) gave as locality “Macahé, près de Cabo Frio, Brésil (Paz)”.

GenusMegalobulimus Miller, 1878

Bulimus (Megalobulimus) Miller, 1878: 172.

Type species. Boris garciamoreni Miller, 1878, by monotypy.

Figure 38 Material collected by the CCP.

(A–D) Strophocheilidae. Megalobulimus granulosus (Rang, 1831), MNCN 15.05/13294, (A) ventral view; Megalobulimus gummatus (Hidalgo, 1870), MNCN 15.05/3199, (B) ventral view; Megalobulimus ovatus (Müller, 1774), MNCN 15.05/7336, (C) ventral view; Megalobulimus oblongus (Müller, 1774), MNCN 15.05/36948, (D) ventral view. Scale line 1 cm.

Megalobulimus granulosus (Rang, 1831) [175]

(Fig. 38A)

Helix (Cochlogena) granulosa Rang, 1831: 53, pl. 2.

Bulimus granulosus; Hidalgo, 1870: 43; Hidalgo, 1872: 51; Hidalgo, 1893a: 89; Hidalgo, 1893b:197.

Type locality.“l’intérieur du Brésil”.

Type material. Not located.

Material examined. “Sta. Catalina”, “(Cat. Am. mer. no. 58)”, Coll. Paz, MNCN 15.05/13294 (2); “Sta. Catalina (Brasil)”, Coll. Hidalgo ex “Martínez y Paz” leg., MNCN 15.05/36847 (6); “I. Sta. Catalina, Brasil”, Coll. Azpeitia, MNCN 15.05/7206 (3).

Megalobulimus gummatus (Hidalgo, 1870) [176]

(Fig. 38B)

Bulimus gummatus Hidalgo, 1870: 41; Hidalgo, 1872: 49, pl. 4 fig. 1; Hidalgo, 1875: 128; Hidalgo, 1893a: 62, 88; Hidalgo, 1893b: 195; Azpeitia, 1923: 73; Calvo, 1994: 284.

Type locality. [Brazil] “Rio Janeiro”.

Type material.“Rio Janeiro”, “Viaje al Pacifico, M[oluscos]”, Coll. Paz, MNCN 15.05/7899 (3); “Pacifico 55”, Coll. Hidalgo, MNCN 15.05/3204 (1); “Rio Janeiro, Brasil”, Coll. Azpeitia, MNCN 15.05/3199 (1), syntypes.

Remarks. The name Bulimus gummatus was introduced by Hidalgo in his catalogue (Hidalgo, 1870), based on material from Paz, with reference to Bulimus cantagallanus Pfeiffer, 1859 not Rang, 1831. Hidalgo gave a lengthy discussion about the differences in the descriptions of the two authors. The taxon was considered a subspecies of Strophocheilus terrestris Spix in Wagner, 1827 by Bequaert (1948: 115), but treated a distinct species by Simone (2006: 211); however, the latter author gave an erroneous year of publication.

Megalobulimus oblongus (Müller, 1774) [177]

(Fig. 38D)

Helix oblonga Müller, 1774: 86.

Bulimus oblongus; Hidalgo, 1870: 43; Hidalgo, 1872: 52; Hidalgo, 1893a: 89; Hidalgo, 1893b: 196.

Type locality. Not given.

Type material. Not located.

Material examined. “Uruguay”, “Pacifico 57”, Coll. Hidalgo ex Martínez leg., MNCN 15.05/36948 (5); “Brasil”, “(Cat. Am. mer. no. 57)”, Coll. Paz, MNCN 15.05/13292 (1), MNCN 15.05/13297 (2).

Remarks. Hidalgo (1870, 1872, 1893a) reported the material from “Uruguay (Martinez)”; it is unclear why the material of Paz (with the correct catalogue number) was not mentioned by Hidalgo.

Megalobulimus ovatus (Müller, 1774) [178]

(Fig. 38C)

Helix ovata Müller, 1774: 85.

Bulimus ovatus; Hidalgo, 1870: 40; Hidalgo, 1872: 47; Hidalgo, 1893a: 88; Hidalgo, 1893b: 194.

Type locality. “in India orientali [sic]”.

Type material. Not located.

Material examined. “Macahé (Brazil)”, “Viaje al Pacifico”, Coll. Paz, MNCN 15.05/36943 (2); “Macahé, Brasil”, Coll. Azpeitia, MNCN 15.05/7336 (2); “Viaje al Pacifico”, Coll. Graells, MNCN 15.05/7900 (5).

Remarks. This material was mentioned by Hidalgo (1870) from “Macahé, Brésil (Paz et Martinez)”.

Figure 39 Material collected by the CCP.

(A–E) Strophocheilidae. Megalobulimus popelairianus (Nyst, 1845), MNCN 15.05/48045, (A) ventral view; Megalobulimus terrestris (Spix in Wagner, 1827), MNCN 15.05/36940, (B) ventral view; Mirinaba planidens (Michelin, 1831), MNCN 15.05/13284, (C) ventral view; Strophocheilus pudicus (Müller, 1774), MNCN 15.05/13283, (D) ventral view; Megalobulimus valenciennesii (Pfeiffer, 1842), MNCN 15.05/7487, (E) ventral view. Scale line 1 cm.

Megalobulimus popelairianus (Nyst, 1845) [179]

(Fig. 39A)

Bulimus popelairianus Nyst, 1845: 151, pl. 3 fig. 5; Hidalgo, 1870: 40; Hidalgo, 1872: 46; Hidalgo, 1893a: 88.

Type locality. “South America”.

Type material. RBINS MT.2890, syntype.

Material examined. “Pacifico 53”, Coll. Hidalgo, MNCN 15.05/36952 (1); “Napo, Ecuador”, Coll. Azpeitia, MNCN 15.05/48045 (3).

Remarks. Hidalgo (1870) gave as localities “Quito (Isern), Bodega (Paz), Napo (Martínez)”; the second locality was not mentioned in Hidalgo (1872). The shell from lot MNCN 15.05/36952 is only tentatively referred to this species, as it is relatively slender.

Megalobulimus terrestris (Spix in Wagner, 1827) [180]

(Fig. 39B)

Bulimus terrestris Spix in Wagner, 1827: pl. 6 fig. 1.

Bulimus cantagallanus Rang; Hidalgo, 1870: 43; Hidalgo, 1872; 50; Hidalgo, 1893a: 89.

Type locality. [Brazil] “Provinciae Bahiensis”.

Type material. ZSM.

Material examined. “Brazil”, Coll. Paz “(Cat. Am. mer. no. 56)”, MNCN 15.05/13295 (1); Coll. Hidalgo “Pacifico 56”, MNCN 15.05/36940 (2).

Remarks. The name used by Hidalgo is considered a junior subjective synonym by Bequaert (1948: 108). The material was recorded by Hidalgo (1870) from “Rio Janeiro (Paz)”.

Megalobulimus valenciennesii (Pfeiffer, 1842) [181]

(Fig. 39D)

Bulimus valenciennesii Pfeiffer, 1842: 52; Hidalgo, 1893a: 122.

Type locality. “Brasil int[erior].”.

Type material. Not located.

Material examined. “Brasil”, Coll. Paz, MNCN 15.05/7487 (1).

Remarks. This material was mentioned by Hidalgo (1893a) from “República del Brasil (Paz)”.

GenusMirinaba Morretes, 1952

Strophocheilus (Mirinaba) Morretes, 1952: 111.

Type species. Strophocheilus erythrostoma Pilsbry, 1895, by original designation.

Mirinaba planidens (Michelin, 1831) [182]

(Fig. 39C)

Bulimus planidens Michelin, 1831: pl. 25; Hidalgo, 1870: 46; Hidalgo, 1893a: 93.

Type locality. “Brazil”.

Type material. MNHN ?

Material examined.“Rio Janeiro”, “(Cat. Am. mer. no. 76)”, Coll. Paz, MNCN 15.05/13284 (2); “Rio Janeiro (Brasil)”, Coll. Hidalgo ex Paz leg., MNCN 15.05/36827 (2); “Corcovado, Rio Janeiro, Brasil”, Coll. Azpeitia, MNCN 15.05/7339 (2).

Remarks. See Bequaert (1948: 40) for a discussion on the dates of publication of this species. This material was mentioned by Hidalgo (1870) from “Corcobado, à Rio Janeiro (Paz)”.

Genus Speironepion Bequaert, 1948

Strophocheilus (Speironepion) Bequaert, 1948: 26.

Type species. Bulinus milleri Sowerby, 1838, by original designation.

Speironepion milleri (Sowerby I in Sowerby I & II, 1838) [183]

(Fig. 37F)

Bulinus milleri Sowerby I in Sowerby I & II, 1838 [1832–1838]: fig. 94.

Bulimus milleri; Hidalgo, 1893a: 124.

Type locality. Not given.

Type material. Not located.

Material examined. “Brasil”, Coll. Paz, MNCN 15.05/13298 (2).

GenusStrophocheilus Spix in Wagner, 1827

Strophocheilus Spix in Wagner, 1827: pl. 11.

Type species. Strophocheilus almeida Spix in Wagner, 1827, by subsequent designation (Nevill, 1878: 122).

Strophocheilus pudicus (Müller, 1774) [184]

(Fig. 39D)

Helix pudica Müller, 1774: 97.

Bulimus almeida Spix; Hidalgo, 1893a: 124.

Type locality. Not given.

Type material. Not located.

Material examined. “Brasil”, Coll. Paz, MNCN 15.05/13283 (2); “Bahia, Brasil”, Coll. Azpeitia, MNCN 15.05/7179 (2).

Remarks. Hidalgo (1893a) published as locality “Bahia, en el Brasil (Paz)”.

Family Scolodontidae H.B. Baker, 1928

Genus Happia Bourguignat, 1889

Happia Bourguignat, 1889: 39. Nom. nov. for Ammonoceras Pfeiffer, 1856 not Lamarck, 1822.

Type species. Helix vitrina Wagner, 1827, by subsequent designation (Gude, 1902: 233).

Happia cf. cuzcana (Philippi, 1869) [185]

(Figs. 40A–40C)

Figure 40 Material collected by the CCP.

(A–I) Scolodontidae. Happia cf. cuzcana (Philippi, 1869), MNCN 15.05/3304, (A) ventral view, (B) umbilical view, (C) apical view; Happia vitrina (Wagner, 1827), MNCN 15.05/12959, (D) ventral view, (E) umbilical view, (F) apical view; Prohappia besckei (Dunker, 1847), MNCN 15.05/12578, (G) ventral view, (H) umbilical view, (I) apical view. Scale line 5 mm.

Helix cuzcana Philippi, 1869: 37.

Helix baezensis Hidalgo, 1869c: 411; Hidalgo, 1870: 38, pl. 6 fig. 2; Hidalgo, 1872: 26, 152; Hidalgo, 1875: 127; Hidalgo, 1893a: 86; Hidalgo, 1893b: 281; Azpeitia, 1923: 85; Calvo, 1994: 284.

Type locality. [Peru] “valle Setae Crucis, dept. del Cuzco”.

Type material. Not located.

Additional type material examined. “Baeza”, “(Cat. Am. mer. no. 45)”, Coll. Paz, MNCN 15.05/3304 (2); Coll. Hidalgo, MNCN 15.05/3177 (1), syntypes of Helix baezensis Hidalgo.

Remarks. Both taxa from Philippi and Hidalgo have been synonymised in literature (e.g., Cousin, 1887), but we prefer to do this only tentatively given the great geographical distance between the type localities and the lack of in-depth studies for this group.

Happia vitrina (Wagner, 1827) [186]

(Figs. 40D–40F)

Helix vitrina Wagner, 1827: 25; Hidalgo, 1870: 36; Hidalgo, 1893a: 84.

Type locality. “Provinciis australioribus Brasiliae”.

Type material. Not located.

Material examined. “Rio Janeiro”, “(Cat. Am. mer. no. 34)”, Coll. Paz, MNCN 15.05/12959 (3); Coll. Hidalgo, MNCN 15.05/39934 (1).

Genus Prohappia Thiele, 1927

Happia (Prohappia) Thiele, 1927: 313.

Type species. Helix besckei Dunker, 1847, by original designation.

Prohappia besckei (Dunker, 1847) [187]

(Figs. 40G–40I)

Helix besckei Dunker, 1847: 81; Hidalgo, 1870: 37; Hidalgo, 1893a: 85.

Type locality. “Brasilia”.

Type material. Not located.

Material examined. “Rio Janeiro”, Coll. Paz, MNCN 15.05/12578 (2); “Rio Janeiro”, Coll. Hidalgo ex Martínez y Saez leg., MNCN 15.05/39950 (1); Coll. Hidalgo ex “Martínez y Paz”, MNCN 15.05/39928 (1).

Family Charopidae Hutton, 1884

Genus Lilloiconcha Weyrauch, 1965

Lilloiconcha Weyrauch, 1965: 127.

Type species. Austrodiscus superbus tucumanus Hylton Scott, 1963, by original designation.

Lilloiconcha pazi (Philippi, 1866) [188]

(Figs. 41A–41C)

Figure 41 Material collected by the CCP.

(A–L) Charopidae. Lilloiconcha pazi (Philippi, 1866), MNCN 15.05/76220, (A) ventral view, (B) umbilical view, (C) apical view; Ptychodon amancaezensis (Hidalgo, 1869), MNCN 15.05/3173, (D) ventral view, (E) umbilical view, (F) apical view; Stephanoda binneyana (Pfeiffer, 1847), MNCN 15.05/12956, (G) ventral view, (H) umbilical view, (I) apical view; Zilchogyra costellata (d’Orbigny, 1835), MNCN 15.05/76209, (J) ventral view, (K) umbilical view, (L) apical view. Scale line 500 µm (D–F), 1 mm (A–C), 5 mm (G–L).

Helix pazi; Philippi, 1866: 39; Hidalgo, 1870: 39; Hidalgo, 1872: 44, pl. 2 figs. 10–11; Hidalgo, 1875: 127; Hidalgo, 1893a: 85; Hidalgo, 1893b: 145.

Type locality. [Chile] “Prope Valparaiso”.

Type material. Not located.

Material examined.“Valparaiso”, Coll. Hidalgo ex Paz leg., MNCN 15.05/76220 (88).

Remarks. The material which Philippi used for his description was collected by the CCP, and likely presented to him during their meeting on the 18th May 1863 in Santiago de Chile (Blanco, Rodríguez & Rodríguez, 2006: 112–113).

Etymology. Named after Patricio Paz y Membiela.

GenusPtychodon Ancey, 1888

Ptychodon Ancey, 1888: 372.

Type species. Strobila leiodus Hutton, 1883, by original designation.

Ptychodon amancaezensis (Hidalgo, 1869) [189]

(Figs. 41D–41F)

Helix amacaezensis Hidalgo, 1869: 411; Hidalgo, 1870: 38, pl. 6 fig. 3; Hidalgo, 1893a: 55, 86; Azpeitia, 1923: 80; Calvo, 1994: 284.

Type locality. “Amancaez, in vicinio urbis Lima dictae, reipublicae Peruvianae”.

Type material. “Amancaez”, Coll. Paz “(Cat. Am. mer. no. 44)”, MNCN 15.05/3173 (62), syntypes.

Stephanoda Martens in Albers, 1860

Helix (Stephanoda) Martens in Albers, 1860: 88.

Type species. Helix dissimilis d’Orbigny, 1837, by original designation.

Stephanoda binneyana (Pfeiffer, 1847) [190]

(Figs. 41G–41I)

Helix binneyana Pfeiffer, 1847b: 13; Hidalgo, 1870: 34; Hidalgo, 1872: 24; Hidalgo, 1893a: 85; Hidalgo, 1893b: 148.

Type locality. [Chile] “insula Chiloe”.

Type material. Not located.

Material examined. “Valdivia”, Coll. Paz “(Cat. Am. mer. no. 37)”, MNCN 15.05/12956 (4); [Coll. Hidalgo,] MNCN 15.05/76234 (1).

Remarks. Lot 76234 only has a species label, but is written in Hidalgo’s hand.

Genus Zilchogyra Weyrauch, 1965

Zilchogyra Weyrauch, 1965: 122.

Type species. Helix costellata d’Orbigny, 1835, by original designation.

Zilchogyra costellata (d’Orbigny, 1835) [191]

(Figs. 41J–41L)

Helix costellata d’Orbigny, 1835: 5; Hidalgo, 1870: 37; Hidalgo, 1872: 31; Hidalgo, 1893a: 85; Hidalgo, 1893b: 146.

Type locality. “Montevideo (republica Paraguayensi orientali)”.

Type material. NHMUK 1854.12.4.69 (2), syntypes.

Material examined. “Sta. Lucia (Montevideo)”, Coll. Hidalgo ex Paz leg., MNCN 15.05/76209 (13).

Family Euconulidae H.B. Baker, 1928

Euconulus martinezi (Hidalgo, 1869) [192]

(Figs. 42A–42B)

Figure 42 Material collected by the CCP.

(A–B) Euconulidae. Euconulus martinezi (Hidalgo, 1869), MNCN 15.05/3190, (A) ventral view, (B) apical view. (C–F) Pleurodontidae. Labyrinthus manueli Higgins, 1872, MNCN 15.05/13803, (C) ventral view, (D) apical view, (E) umbilical view, (F) lateral view (lip). Scale line 1 mm (A–B), 5 mm (C–F).

Helix martinezi Hidalgo, 1869: 411; Hidalgo, 1870: 38, pl. 6 fig. 4; Hidalgo, 1872: 23, pl. 2 figs. 12–13; Hidalgo, 1893a: 54, 86; Hidalgo, 1893b: 144; Azpeitia, 1923: 89; Calvo, 1994: 284.

Type locality. “Bahia, imperii Brasiliani”.

Type material. “Bahia”, “(Cat. Am. mer. no. 43)”, Coll. Hidalgo ex Paz leg., MNCN 15.05/3188 (86); “Bahia”, Coll. Hidalgo ex Paz leg., MNCN 15.05/3189 (35); “Bahia”, Coll. Hidalgo ex Martínez leg., MNCN 15.05/3190 (38), syntypes.

Additional material examined. “Bahia, Brasil”, Coll. Azpeitia, MNCN 15.05/3300 (6); “Brasil? (Ej. Paz)”, Coll. Azpeitia, MNCN 15.05/2299 (8), Coll. Hidalgo, MNCN 15.05/3202 (16).

Etymology. Named after Francisco de Paula Martinez y Sáez.

Family Pleurodontidae Ihering, 1912

Genus Labyrinthus Beck, 1837

Helix (Labyrinthus) Beck, 1837: 33.

Type species. Helix otis Lightfoot, 1786, by subsequent designation (Gray, 1847: 173).

Labyrinthus manueli Higgins, 1872 [193]

(Figs. 42C–42F)

Helix quadridentata Broderip; Hidalgo, 1870: 33; Hidalgo, 1872: 16, pl. 1 figs. 8–9; Hidalgo, 1893a: 81. Not Caracolla quadridentata Broderip, 1832.

Labyrinthus manueli Higgins, 1872: 686, pl. 56 fig. 5a.

Helix manueli; Hidalgo, 1893b: 182.

Type locality. “Macas, Ecuador”.

Type material. NMW 1955.158.01192 (1), syntype.

Material examined.“Ecuador”, “(Cat. Am. 21)”, Coll. Paz, MNCN 15.05/13803 (2); “Napo (Ecuador)”, Coll. Hidalgo ex Martínez leg., “individuo figurado”, MNCN 15.05/58498; “Napo, Ecuador”, Coll. Azpeitia, MNCN 15.05/58499 (2); “21. Pacifico”, Coll. Graells, MNCN 15.05/58500 (1).

Remarks. Hidalgo (1870) mentioned “Napo, République de l’Équateur (Martínez)”. His label appears to have been written after 1872, as he gave the correct species name (“Helix Manueli Higgins / (quadridentata Brod.)”).

Labyrinthis raimondii (Philippi, 1867) [194]

(Figs. 43A–43D)

Figure 43 Material collected by the CCP.

(A–H) Pleurodontidae. Labyrinthis raimondii (Philippi, 1867), MNCN 15.05/58495, (A) ventral view, (B) apical view, (C) umbilical view, (D) lateral view (lip); Labyrinthus otis otis (Lightfoot, 1786), MNCN 15.05/13957, (E) ventral view, (F) apical view, (G) umbilical view, (H) lateral view (lip). Scale line 5 mm.

Helix raimondii Philippi, 1867: 65; Hidalgo, 1870: 33; Hidalgo, 1872: 17, pl. 2 figs. 4–5; Hidalgo, 1893a: 81; Hidalgo, 1893b: 180.

Type locality. “provincia Loreto inter S[anta]. Catalina et Yanayaco”.

Type material. Not located.

Material examined. “Napo (Ecuador)”, Coll. Hidalgo, MNCN 15.05/58495 (5); “Napo, Ecuador”, Coll. Azpeitia, MNCN 15.05/58493 (2); “Ecuador”, “(Cat. Am. mer. no. 20)”, Coll. Paz, MNCN 15.05/14081 (2), MNCN 15.05/14116 (1); “Ecuador”, Coll. Graells, MNCN 15.05/58494 (1).

Remarks. Hidalgo (1870) mentioned “Napo, République de l’Équateur (Martínez)”, and stated he found the shells nearly the same as Helix taratoponensis Moricand. The figured specimen corresponds to Hidalgo ( 1872: pl. 2 figs. 4–5).

Labyrinthus otis otis (Lightfoot, 1786) [195]

(Figs. 43E–43H)

Helix otis Lightfoot, 1786: 38, 53.

Helix labyrinthus Chemnitz; Hidalgo, 1870: 33; Hidalgo, 1893a: 81.

Type locality. Not given.

Type material. Not located.

Material examined. “Panama”, “(Cat. Am. mer. no. 18)”, Coll. Paz, MNCN 15.05/13957 (1); “Panamá”, Coll. Hidalgo, MNCN 15.05/58515 (1).

Remarks. The material was likely collected by Martinez (Calatayud, 1994: 259).

Labyrinthus plicatus (Born, 1780) [196]

(Figs. 44A–44D)

Figure 44 Material collected by the CCP.

(A–G) Pleurodontidae. Labyrinthis plicatus (Born, 1780), MNCN 15.05/14208, (A) ventral view, (B) apical view, (C) umbilical view, (D) lateral view (lip); Isomeria aequatoriana (Hidalgo, 1867), MNCN 15.05/3170, (E) ventral view, (F) apical view, (G) umbilical view. Scale line 5 mm (A–D), 1 cm (E–G).

Helix plicata Born, 1780: 368; Helix plicatus; Hidalgo, 1870: 33; Hidalgo, 1893a: 81.

Type locality. “East Indies” [sic, see Solem, 1966: 122]

Type material. Not located.

Material examined. “Panamá”, “(Cat. Am. mer. no. 19)”, Coll. Paz, MNCN 15.05/14208 (2); “Panama”, Coll. Azpeitia, MNCN 15.05/58497 (1); “Pacifico 19”, Coll. Grealls, MNCN 15.05/58496 (3).

Remarks. The material was likely collected by Martinez (Calatayud, 1994: 259).

Genus Isomeria Albers, 1850

Helix (Isomeria) Albers, 1850: 126.

Type species. Helix oreas Koch, 1844, by monotypy.

Isomeria aequatoriana (Hidalgo, 1867) [197]

(Figs. 44E–44G)

Helix aequatoriana Hidalgo, 1867b: 307, pl. 8 fig. 2; Hidalgo, 1870: 31; Hidalgo, 1893a: 45, 79; Azpeitia, 1923: 84; Calvo, 1994: 284.

Type locality. “Republica Aequatoris”.

Type material. “(Cat. Am. mer. no. 8)”, Coll. Paz, MNCN 15.05/3170 (1), MNCN 15.05/3171 (1), syntypes.

Additional material examined. “Quito, Ecuador”, Coll. Azpeitia, MNCN 15.05/3172 (2).

Remarks. Hidalgo had the largest syntype sent to Paris for illustration and marked it “tipo”.

Isomeria bituberculata (Pfeiffer, 1853) [198]

(Figs. 45A–45C)

Figure 45 Material collected by the CCP.

(A–I) Pleurodontidae. Isomeria bituberculata (Pfeiffer, 1853), MNCN 15.05/58506, (A) ventral view, (B) umbilical view, (C) apical view; Isomeria bourcieri (Pfeiffer, 1853), MNCN 15.05/58504, (D) lateral view (lip), (E) umbilical view, (F) apical view; Isomeria cymatodes (Pfeiffer, 1852), MNCN 15.05/58506, (G) ventral view, (H) umbilical view, (I) apical view. Scale line 5 mm.

Helix bituberculata Pfeiffer, 1853: 242; Hidalgo, 1870: 32; Hidalgo, 1872: 14; Hidalgo, 1893a: 80; Hidalgo, 1893b: 176.

Type locality. [Ecuador] “prope Tunguragua reipublicae Aequatoris”.

Type material. NHMUK 20160369 (3), syntypes.

Material examined. “Quito”, Coll. Hidalgo ex Martínez leg., MNCN 15.05/58505 (2); “Quito”, “(Cat. Am. mer. no. 14)”, Coll. Paz MNCN 15.05/13582 (2); “Quito”, Coll. Paz “(Cat. Am. mer. no. 15)”, MNCN 15.05/14022 (3); “Quito, Ecuador”, Coll. Azpeitia, MNCN 15.05/58508 (2); “P-14”, Coll. Graells, MNCN 15.05/58506 (1); “Ecuador”, Coll. Hidalgo, MNCN 15.05/58507 (1).

Isomeria bourcieri (Pfeiffer, 1853) [199]

(Figs. 45D–45F)

Helix bourcieri Pfeiffer, 1853: 209; Hidalgo, 1870: 32; Hidalgo, 1872: 15; Hidalgo, 1893a: 80; Hidalgo, 1893b: 178.

Type locality. [Ecuador] “Otoralo [sic, Otovalo] reipublicae Aequatoris”.

Type material. NHMUK 20160370 (3), syntypes.

Material examined. “Pacifico” “P-15”, Coll. Graells, MNCN 15.05/58504 (3); “Nanegal, Ecuador”, Coll. Azpeitia, MNCN 15.05/58503 (3).

Remarks. Hidalgo (1870) reported the material from “Nanegal”; this locality was not mentioned in the itinerary of the CCP (Calatayud, 1994).

Isomeria cymatodes (Pfeiffer, 1852) [200]

(Figs. 45G–45I)

Helix cymatodes Pfeiffer, 1852b: 92; Hidalgo, 1870: 31; Hidalgo, 1872: 11, pl. 2 figs. 1, 3; Hidalgo, 1893a: 79; Hidalgo, 1893b: 171.

Type locality. “…?”.

Type material. Not located.

Material examined. “Ecuador”, “(Cat. Am. mer. no. 9)”, Coll. Paz, MNCN 15.05/14128 (2); “Napo (Ecuador)”, Coll. Hidalgo ex Martínez leg., MNCN 15.05/58502 (2); “Napo, Ecuador”, Coll. Azpeitia, MNCN 15.05/58501 (3).

Isomeria globosa (Broderip in Broderip & Sowerby I, 1832) [201]

(Figs. 46A–46C)

Figure 46 Material collected by the CCP.

(A–I) Pleurodontidae. Isomeria globosa (Broderip in Broderip & Sowerby I, 1832), MNCN 15.05/58516, (A) ventral view, (B) umbilical view, (C) apical view; Isomeria jacksoni Solem, 1966, MNCN 15.05/58512, (D) lateral view (lip), (E) umbilical view, (F) apical view; Isomeria juno (Pfeiffer, 1850), MNCN 15.05/13984, (G) ventral view, (H) umbilical view, (I) apical view. Scale line 5 mm.

Carocolla globosa Broderip in Broderip & Sowerby I, 1832a: 30.

Helix subcastanea Pfeiffer; Hidalgo, 1870: 32; Hidalgo, 1872: 14; Hidalgo, 1893a: 80; Hidalgo, 1893b: 176.

Type locality.“Insulae Tumaco, Columbiae Occidentalis”.

Type material. Not located.

Material examined. “Ecuador”, “(Cat. Am. mer. no. 11)”, Coll. Paz, MNCN 15.05/14100 (1), MNCN 15.05/14203 (2); “Ecuador”, Coll. Hidalgo, MNCN 15.05/58516 (1).

Remarks. As Solem (1966: 191) has shown, the replacement name introduced by Pfeiffer and used by Hidalgo, was an unnecessary proposal. Hidalgo (1870) reported the material from “Quito”.

Isomeria jacksoni Solem, 1966 [202]

(Figs. 46D–46F)

Helix atrata Pfeiffer; Hidalgo, 1870: 31; Hidalgo, 1872: 12; Hidalgo, 1893a: 79; Hidalgo, 1893b: 173.

Isomeria jacksoni Solem, 1966: 178. New name for Helix atrata Pfeiffer, 1854 not Reeve, 1852.

Type locality. “Puntophaya, reipublicae Aequatoris” (Pfeiffer, 1854: 153).

Type material. NHMUK 200160372 (3), syntypes.

Material examined. “Ecuador”, Coll. Paz, MNCN 15.05/14127 (1); “Napo (Ecuador)”, Coll. Hidalgo ex Martínez y Saez leg., MNCN 15.05/58512 (4); “Pacifico”, Coll. Graells, MNCN 15.05/58511 (1).

Remarks. Hidalgo (1870) mentioned material from “Macas et Napo, République de l’Équateur (Martínez)”; the specimens from Macas have not been located.

Isomeria juno (Pfeiffer, 1850) [203]

(Figs. 46G–46I)

Helix juno Pfeiffer, 1850: 66; Hidalgo, 1870: 32; Hidalgo, 1872: 13, pl. 1 figs. 6–7; Hidalgo, 1893a: 80; Hidalgo, 1893b: 175.

Type locality. “Andibus Columbiae”.

Type material. Not located.

Material examined. “Quito”, Coll. Hidalgo, MNCN 15.05/58483 (2); “Baeza, Ecuador”, “(Cat. Am. mer. no. 13)”, Coll. Paz, MNCN 15.05/13982 (2), MNCN 15.05/13984 (2), MNCN 15.05/13990 (2); “Baeza (Ecuador)”, “Pacifico 13”, Coll. Hidalgo ex Martínez leg., MNCN 15.05/58481 (27); “Baeza, Ecuador”, Coll. Azpeitia, MNCN 15.05/58486 (20); “Ecuador”, Coll. Graells, MNCN 15.05/58484 (3).

Isomeria morula (Hidalgo, 1870) [204]

(Figs. 47A–47C)

Helix martinii Bernardi, 1858: 93, pl. 1 fig. 3. Not Pfeiffer, 1854.

Helix morula Hidalgo, 1870: 32 (new name for Helix martinii Bernardi not Pfeiffer); Hidalgo, 1893a: 80.

Type locality. “Quito, République de l’Équateur”.

Type material examined. “Ecuador”, “(Cat. Am. mer. no. 12)”, Coll. Paz, MNCN 15.05/60012, lectotype (Borrero & Araujo, 2012: 146).

Remarks. Bernardi based himself on material from Paz; Borrero & Araujo (2012) assumed this specimen was returned to Paz, and they considered it as type material.

Figure 47 Material collected by the CCP.

(A–L) Pleurodontidae. Isomeria morula (Hidalgo, 1870), MNCN 15.05/60012, (A) ventral view, (B) umbilical view, (C) apical view; Polygyratia polygyrata (Born, 1778), MNCN 15.05/14101, (D) lateral view (lip), (E) umbilical view, (F) apical view; Polygyratia heligmoida (d’Orbigny, 1835), MNCN 15.05/76238, (G) ventral view, (H) umbilical view, (I) apical view; Polygyratia reyrei (Souverbie, 1858), MNCN 15.05/76235, (J) ventral view, (K) umbilical view, (L) apical view. Scale line 1 mm (J–L), 5 mm (A–I).

GenusPolygyratia Gray, 1847

Polygyratia Gray, 1847: 173.

Type species. Helix polygyrata Born, 1778, by monotypy.

Polygyratia polygyrata (Born, 1778) [205]

(Figs. 47D–47F)

Helix polygyrata Born, 1778: 382; Hidalgo, 1870: 32; Hidalgo, 1893a: 80.

Type locality. Not given.

Type material. Not located.

Material examined. “Brasil”, “(Cat. Am. mer. no. 16)”, Coll. Paz MNCN 15.05/14101 (1); “Pacifico 16”, Coll. Hidalgo, MNCN 15.05/39937 (4); “Bahia, Brasil”, Coll. Azpeitia, MNCN 15.05/39938 (3).

Remarks. Hidalgo (1870) gave as locality “Bahia, Brésil (Paz)”.

Polygyratia heligmoida (d’Orbigny, 1835) [206]

(Figs. 47G–47I)

Helix (Helicogena) heligmoida d’Orbigny, 1835: 2.

Helix heligmoida; Hidalgo, 1870: 33; Hidalgo, 1872: 25; Hidalgo, 1893a: 81; Hidalgo, 1893b: 151.

Type locality. “provincia Guayaquilensi (republica Colombiana)”.

Type material. NHMUK 1854.12.4.106 (3), syntypes.

Material examined.“Guayaquil”, Coll. Hidalgo ex Paz, MNCN 15.05/76238 (15).

Polygyratia reyrei (Souverbie, 1858) [207]

(Figs. 47J–47L)

Helix reyrei Souverbie, 1858: 65; Hidalgo, 1870: 36; Hidalgo, 1893a: 84.

Type locality. “Guayaquil (Columbia)”.

Type material. Not located.

Material examined. [Guayaquil], Coll. Hidalgo, MNCN 15.05/19733 (12); “Ecuador”, Coll. Azpeitia, MNCN 15.05/76235 (12).

Genus Solaropsis Beck, 1837

Helix (Solaropsis) Beck, 1837: 27.

Type species. Helix pellis serpentis Chemnitz, 1795 (=Helix undata Lightfoot, 1786), by subsequent designation (Herrmannsen, 1848 [1847–1849]: 468).

Solaropsis brasiliana (Deshayes, 1832) [208]

(Figs. 48A–48C)

Figure 48 Material collected by the CCP.

(A–I) Pleurodontidae. Solaropsis brasiliana (Deshayes, 1832), MNCN 15.05/14082, (A) ventral view, (B) umbilical view, (C) apical view; Solaropsis gibboni (Pfeiffer, 1846), MNCN 15.05/3169, (D) lateral view (lip), (E) umbilical view, (F) apical view; Psadara quadrivittata (Hidalgo, 1869), MNCN 15.05/3193, (G) ventral view, (H) umbilical view, (I) apical view. (J–L) Bradybaenidae. Bradybaena similaris (Férussac in Rang, 1831), MNCN 15.05/13124, (J) ventral view, (K) umbilical view, (L) apical view. Scale line 5 mm (A–C, G–L), 1 cm (D–F).

Helix brasiliana Deshayes in Férussac & Deshayes, 1832 [1819–1851]: 211; Hidalgo, 1870: 31; Hidalgo, 1893a: 79; Hidalgo, 1893b: 183.

Type locality. “le Brésil”.

Type material. Not located.

Material examined. “Brasil”, “(Cat. Am. mer. no. 7)”, Coll. Paz, MNCN 15.05/14082 (1), MNCN 15.05/14117 (3); “Rio Janeiro”, “Pacifico 7” “(comprado)”, Coll. Hidalgo ex Paz, MNCN 15.05/39932 (5); “Brasil”, Coll. Azpeitia, MNCN 15.05/39939 (2); Coll. Graells, MNCN 15.05/39933 (1).

Solaropsis gibboni (Pfeiffer, 1846) [209]

(Figs. 48D–48F)

Helix gibboni Pfeiffer, 1846: 37. New name for Helix magnifica Lea, 1838 not Férussac, 1821.

Helix amori Hidalgo, 1867: 71, pl. 1 fig. 3; Hidalgo, 1870: 30; Hidalgo, 1872: 7, pl. 1 figs. 1–3; Hidalgo, 1893a: 40, 79; Azpeitia, 1923: 85.

Type locality.“New Granada” (Lea, 1838: 89).

Type material. USNM 105367, holotype.

Additional type material. (Helix amori Hidalgo, 1867) “Tena (Ecuador)”, Coll. Paz ex Martínez leg., MNCN 15.05/3166 (2), MNCN 15.05/3168 (2), syntypes.

Additional material examined. “Tena, Ecuador”, Coll. Azpeitia, MNCN 15.05/3169 (1).

Remarks. Hidalgo (1867) described his taxon from “Tena Republican Aequatoris”, and mentioned it had been collected by Martinez.

Etymology. Hidalgo named this species after Fernando Amor y Mayor.

GenusPsadara Miller, 1878

Helix (Psadara) Miller, 1878: 162.

Type species. Helix selenostoma Pfeiffer, 1852, by subsequent designation (Pilsbry, 1926: 13).

Psadara quadrivittata (Hidalgo, 1869) [210]

(Figs. 48G–48I)

Helix quadrivittata Hidalgo, 1869c: 410; Hidalgo, 1870: 34, pl. 6 fig. 1; Hidalgo, 1872: 10, pl. 2, figs. 6–7; Hidalgo, 1893a: 52, 82; Hidalgo, 1893b: 185; Azpeitia, 1923: 91.

Type locality.“Baeza, reipublicae Aequatoris”.

Type material examined. “Baeza Ecuador”, Coll. Hidalgo “(Cat. Am. mer. no. 25)” ex Martínez y Saez leg., MNCN 15.05/3193 (1); “Baeza (Ecuador)”, Coll. Hidalgo ex Martínez y Saez leg., MNCN 15.05/3194 (1), syntypes.

Family Bradybaenidae Pilsbry, 1939

Genus Bradybaena Beck, 1837

Bradybaena Beck, 1837: 18.

Type species. Helix (Helicella) similaris Rang, 1831, by subsequent designation (Gray, 1847: 173).

Bradybaena similaris (Férussac in Rang, 1831) [211]

(Figs. 48J–48L)

Helix similaris Férussac, 1821 [1821–1822]: 43 (nomen nudum); Férussac in Rang, 1831: 15.

Helix similaris; Hidalgo, 1870: 36; Hidalgo, 1872: 20; Hidalgo, 1893a: 84; Hidalgo, 1893b: 154.

Type locality. “Timor” (Férussac, 1821: 43).

Type material. Not located.

Material examined. “Bahia”, Coll. Paz “(Cat. Am. mer. no. 23)”, MNCN 15.05/12970 (5), MNCN 15.05/13124 (4); “Bahia”, Coll. Hidalgo ex “Martinez y Paz”, MNCN 15.05/39936 (21); Coll. Hidalgo “P-33”, MNCN 15.05/20280 (8); Coll. Hidalgo, MNCN 15.05/39935 (2).

Remarks. This species has invariably been cited as ‘(Férussac, 1821)’, overlooking the fact that the original citation was a nomen nudum. The first description appeared in Rang (1831), who cited Férussac as author.

Material supposed to be present but not located

Despite intensive searching, we have been unable to locate material of the following 34 species listed by Hidalgo (numbers between parenthesis before the species name refer to his catalogue): (185) Clausilia crossei Hidalgo, 1869 [22], (193) Cyclophorus hidalgoi Crosse, 1866 [212], (134) Bulimus cuneus Pfeiffer, 1853 [213], (231) B. elegans Pfeiffer, 1842 [214], (210) B. fucatus Reeve, 1849 [215], (74) B. inca d’Orbigny, 1835 [216], (205) B. kuehnholtzianus Crosse, 1870 [217], (233) B. musivus Pfeiffer, 1855 [218], (130) B. ochsenii Dunker, 1856 [219], (171) B. orophilus Morelet, 1860 [220], (211) B. peliostomus Pfeiffer, 1867 [221], (232) B. petasites Miller, 1878 [222], (208) B. sylvaticus Spix in Wagner, 1827 [223], (230) B. variegatus Pfeiffer, 1842 [224], (170) B. vespertinus Pfeiffer, 1858 [225], (219) B. wagneri Pfeiffer, 1842 [226], (179) Glandina striata Müller, 1774 [227], (201) Helicina rhynchostoma Shuttleworth in Pfeiffer, 1865 [228], (23) Helix andium Philippi in Pfeiffer, 1867 [229], (27) H. angrandi Morelet, 1863 [230], (41) H. bryophyla Philippi, 1855 [231], (28) H. claromphalos Hupé & Deville, 1850 [232], (24) H. flora Pfeiffer, 1850 [233], (216) H. hidalgonis Doering, 1878 [234], (38) H. insignis d’Orbigny, 1835 [235], (32) H. lactea Müller, 1774 [236], (214) H. mauritii Jousseaume, 1887 [237], (215) H. patasensis Pfeiffer, 1859 [238], (30) H. reentsi Philippi, 1855 [239], (26) H. trenquelleonis Grateloup in Pfeiffer, 1850 [240], (42) H. trochilioneides d’Orbigny, 1835 [241], (29) H. tschudiana Philippi, 1867 [242], (240) Orthalicus phlogerus d’Orbigny, 1835 [243], (178) Tornatellina funcki (Pfeiffer, 1848) [244], (177) T. lamellosa (Reeve, 1849) [245].

Table 1 New taxa described on the basis of CCP material.

Taxa arranged alphabetically on species name, with country of origin. Junior subjective synonyms indicated by asterisk.

Genus	(Subgenus)	Species	Authority	Country	
Isomeria		aequatorianus	(Hidalgo, 1867)	Ecuador	
Ptychodon		amancaezensis	(Hidalgo, 1869)	Peru	
Solaropsis		amori	(Hidalgo, 1867)	Ecuador	
Plekocheilus	(Eurytus)	aristaceus	(Crosse, 1869)	Ecuador	
Happia		baezensis*	(Hidalgo, 1869)	Ecuador	
Drymaeus	(Drymaeus)	baezensis	(Hidalgo, 1869)	Ecuador	
Plekocheilus	(Plekocheilus)	cecepeus	Breure & Araujo, 2015	Ecuador	
Drymaeus	(Drymaeus)	chanchamayensis	(Hidalgo, 1870)	Peru	
Stenostylus		colmeiroi	(Hidalgo, 1872)	Ecuador	
Clathrorthalicus		corydon	(Crosse, 1869)	Ecuador	
Neocyclotus		crosseanus	(Hidalgo, 1866)	Ecuador	
Streptaxis		crossei	(Pfeiffer, 1867)	Brazil	
Incania		crossei	(Hidalgo, 1869)	Ecuador	
Neocyclotus		fischeri*	(Hidalgo, 1867)	Ecuador	
Sultana	(Metorthalicus)	fungarinoi	(Hidalgo, 1867)	Ecuador	
Megalobulimus		gummatus	(Hidalgo, 1872)	Brazil	
Neocyclotus		hidalgoi	(Crosse, 1866)	Ecuador	
Scholvienia		iserni	(Philippi, 1867)	Peru	
Plekocheilus	(Eurytus)	jimenezi	(Hidalgo, 1872)	Ecuador	
Euconulus		martinezi	(Hidalgo, 1869)	Brazil	
Buckleyia		martinezi	(Hidalgo, 1866)	Ecuador	
Drymaeus	(Drymaeus)	membielinus	(Crosse, 1867)	Ecuador	
Isomeria		morula	(Hidalgo, 1870)	Ecuador	
Hypselartemon		paivanus	(Pfeiffer, 1867)	Brazil	
Lilloiconcha		pazi	(Philippi, 1866)	Chile	
Gastrocopta		pazi	(Hidalgo, 1869)	Peru, Ecuador	
Neocyclotus		pazi	(Crosse, 1866)	Ecuador	
Neocyclotus		perezi	(Hidalgo, 1866)	Ecuador	
Corona		pfeifferi	(Hidalgo, 1869)	Ecuador	
Psadara		quadrivittata	(Hidalgo, 1869)	Ecuador	
Plekocheilus	(Eurytus)	semipictus*	(Hidalgo, 1869)	Ecuador	
Synapterpes		visendus	(Hidalgo, 1869)	Ecuador	

Discussion

When the Spanish expedition set out in 1862, many areas they would visit had already been explored and many species described. Brazil had been visited by Spix (Wagner, 1827) and by Blanchet in Bahia (species described by Moricand; see Breure & Tardy, 2016; Breure, 2016). d’Orbigny and some French expeditions had explored Brazil, Argentina, Bolivia, Chile and Peru (d’Orbigny [1834–1847]; Hupé, 1857); Peru also had been explored by Angrand (species described by Morelet; see Breure, 2016). The countries at the western coast of South America had previously been visited by Cuming (many species described by Broderip, Sowerby, Reeve and Pfeiffer). Thus of the regions visited, only Ecuador was relatively poorly explored; hence the majority of the new species from the CCP material originated from this country (Table 1). In total 31 new species were described, of which 22 by Hidalgo, five by Crosse, two by Pfeiffer and two by Philippi; compare Calvo (1994) who listed only 19 species, all described by Hidalgo. The CCP may have collected a substantial larger number of land molluscs than hitherto known, not so much in the number of species but in the number of specimens. Almagro (1866: 162–164) listed a total number of ‘Univalvos terrestres’ of 2,117 specimens, including those collected from Tenerife and those received from the Pacific. However, it cannot be excluded this was an underestimation, as the title of his book suggest it may have been restricted to those that were on public display. After the return of the CCP in Madrid, and during the following decades, specimens have been in a ‘state of flux’, being partially transferred from the collection of Paz, to Hidalgo’s collection, and from there to collections abroad and to Azpeitia. Moreover, duplicates from the CCP material have probably been distributed within Spain, but we do not know how much shell material was involved.

The number of lots recognised as CCP material has been augmented through this study from 230 to 560 lots, totalling 3,470 specimens. Actually, this number is somewhat larger as we know from the correspondence of Hidalgo with Crosse that during the years of study of the CCP material, he gifted Crosse and some others material originating from this expedition (Breure & Backhuys, 2017). When Hidalgo started to study the CCP material, original labels seem to have been removed or lost. The example of the label with very precise locality data in the handwriting of Paz (MNCN 15.05/7344) makes one wonder if originally similar labels were present with other specimens (at least Paz could have had the opportunity of doing so). Other indications are the meticulous way in which part of the CCP members kept their diaries (Almagro, Isern, Jiménez de la Espada, and Martinez), and the detailed locality data with the botanical material of Isern (Blanco, Rodríguez & Rodríguez, 2006). It is remarkable that Hidalgo (1870: 56) in only one case wrote “L’étiquette qui portrait la localité exacte de cette coquille a été égarée”, which could be an indication that these original labels with more precise information were provided by the collectors. Hidalgo published in several cases more precise localities than the current labels show, and the original labels may have disappeared through the flux of the collection over time.

When Hidalgo, describing a new species, had more than one specimen at hand, he appeared to have kept in Madrid the shell of which he mentioned the dimensions in the text of his publications, and sent another one to Crosse for illustration in the Journal de Conchyliologie (see e.g., Stenostylus colmeiroi). As in such cases this shell was often kept by Crosse and ended up in “Coll. JdeC”, and consequently is now in the MNHN, these specimens were often not the ones which the author had used as ‘the type’. Later authors, being unaware of this mechanism, may thus have considered this material as the ‘holotype’ (Fischer-Piette, 1950) or ‘lectotype’ (Breure, 1975), noticing at the same time that the dimensions did not match those given in the original publication. This shows once more that contextual information from early science networks can help to give a more precise interpretation when studying historical collections.

Conclusion

The CCP expedition yielded 245 species of land molluscs, of which 32 were new to science and described by six authors between 1866 and 2015. In total 3,470 specimens have been located in the MNCN collection that (presumably) originate from this expedition; these specimens belong to 211 species. Nearly all of the original labels have been lost, either at the initial stage of determination by Hidalgo or subsequently during the ‘flux’ of the collection. The publication of collection localities by Hidalgo often reveals more precise localities than the current labels suggest; the published diaries of some CCP members allowed for a check of these localities and also gives a collection date in the majoriy of cases. Research in archives has revealed that the study of this material and the publication of its results have been hampered by several obstacles. This contextual research has thus shed light on the historical collection by this Spanish expedition.

The following colleagues supplied information on material in collections under their charge or provided otherwise help with identifications: Jonathan Ablett (London), Juan Francisco Araya (Copiapó), Virginie Héros (Paris), Eugenia Salas Oroño (Tucumán), Enrico Schwabe (München). Many thanks are due to Javier de Andrés, Lola Bragado, and the Servício de Fotografía (Jesús Muñoz), all MNCN, for extensive practical assistance during this study. We thank José Leal (The Nautilus) for allowing the reproduction of the Pfeiffer portrait, and Wim Backhuys (Crosse Foundation) for permission to use a portrait and a letter present in the Crosse archive. Jonathan Ablett deserves our gratefulness for a grammatical check of a draft of the manuscript. The reviews of Carl Christensen and an anonymous reviewer helped to improve the manuscript and is here thankfully acknowledged.

Additional Information and Declarations

Competing Interests

Author Contributions

Data Availability

The authors declare there are no competing interests.

Abraham S.H. Breure conceived and designed the experiments, performed the experiments, analyzed the data, contributed reagents/materials/analysis tools, wrote the paper, prepared figures and/or tables, reviewed drafts of the paper.

Rafael Araujo analyzed the data, contributed reagents/materials/analysis tools, wrote the paper, reviewed drafts of the paper.

The following information was supplied regarding data availability:

A supplementary file with a list of species of land Mollusca collected by the Comisión Científica del Pacífico (1862–1866) in South America, and stored in the collection of the Museo Nacional de Historia Natural in Madrid, Spain, is on Figshare: Breure, Bram (2016): Summary of CCP material studied. figshare. https://doi.org/10.6084/m9.figshare.4231904.v1.

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
