# Peer review of "The Neotropical land snails (Mollusca, Gastropoda) collected by the ‘Comisión Científica del Pacífico’"

_PeerJ, doi:10.7717/peerj.3065_

## Round 0.1 · original submission · Minor Revisions

This manuscript has now received two reviews by experts with backgrounds in molluscs and taxonomics. Both Referees agreed that this study and reanalysis of a 19th century collection would make an excellent contribution to PeerJ. The paper provides good historical information providing context on the importance of expedition and collection. I agree the report was well written. I also agree that it is difficult to evaluate the impact of the work from the perspective of history converging with biology. Nonetheless, I see value in careful descriptions of historical collections that contributes to the taxonomic validity of the taxa involved. Along these lines, the Referees make several useful suggestions about certain caveats that should be relatively easy to accommodate or justify, including a reduction in the text in association with the history of the CCP. More importantly, given the authors state they are proposing new lectotype designations, they will need to confirm they have satisfied any ICZN requirements for doing so and address the reviewers’ concerns regarding some of the characters examined. Please address each and every question raised by the reviewers in your revision. Thank you for submitting your work to PeerJ.

Reviewer 1 ·

Basic reporting

The present manuscript is a checklist of the material collected by the ‘Comisión
Científica del Pacifíco’ (CCP) with historical information on the collections, publications and biographic details of the persons directly involved in the study of the land molluscs. Author stated that: “Material from some collections was included when a similar lot had been found with an undisputed CCP origin, while references were given to publications citing the CCP material”.

This manuscript provide good historical information and clarify dates of publications on different species original descriptions, information that will solve nomenclatural problems. It also clarify collection localities by Hidalgo.
The article has a good Introduction and background information giving an appropriate historical context of the expedition.

The structure of the article has an acceptable format of PeerJ sections.
Figures are relevant and with good quality, appropriately described and labeld.
English used by the Authors is clear. Relevant literature is provided.
The submission is self-contained and represents an appropriate unit of publication and includes all the results. There is no hypothesis provided.
As far as I can tell, all appropriate raw data have been made available.

Experimental design

The present manuscript makes special reference to the history of the collection and biographical aspects of researchers, these aspects are relevant in areas of Biology such as taxonomy and nomenclature. Methods are sufficiently detailed.

Validity of the findings

The impact of this type of work, where history converge with biology is difficult to predict. The greatest value is to clarify dates of original descriptions, as well as the sequence of appearance of publications, which are data that contribute to the nomenclature and ultimately to the taxonomy of the taxa involved.
Figures provided are excellent to illustrate species found in the different countries where no taxonomist is currently working, such as Ecuador. However, information should be carefully considered in future research, since this is a checklist not based on a sound taxonomic revision of the taxa involved.
Therefore, designating species lectotypes without having reviewed syntype material of the species in other museums or collections is not appropriate and can lead to errors.
Lectotype designations as well as new combination names should be better justified.

Descriptions and biographical data of participants in the expedition, described under the title: History of the CCP could be substantially shortened.

Areas where the article fails:
File2501- page number of Azpeitia, 1923 is lacking
File 3305: Bulimulus visendus Hidalgo, 1869: authors provide a new combination name (Synapterpes visendus). However, they do not clearly justify why they classified B. visendus in Synapterpes, only mentioning its smooth protoconch, a character not exclusive of Synapterpes, since it is also present in other genera of the family. Besides, protoconch sculpture is an intra specific variable character. Errors of considering a smooth protoconch when it is really a worn shell has previously occurred in other lectotype designations and new name combinations.

File 2498: The new combination of Bulimus chanchamayensis Hidalgo, 1870 as Drymaeus (Drymaeus) chanchamayensis is not properly justified. There are no shell characters described supporting this new combination.
Authors assumed “that Hidalgo had material collected by the CCP at hand when introducing his taxon” (line 2514) changing the type material ZMB 11833 for MNCN 15.05/3157. I see no adequate basis other than an author supposition to make this change.

File 4188: The material identified as Solaropsis brasiliana is probably Solaropsis pilsbry, since these species are often confused. Distribution of S. brasiliana is restricted to Brazil.

Additional comments

See validity of the findings

·

Basic reporting

This article is a reanalysis of an important 19th century collection of terrestrial mollusks from Central and (mostly) South America that was the principal resource available to J. G. Hidalgo, author of several major works on the systematics of these animals published during the period 1870-1893. The report provides useful background information on the history of the collection and the individuals associated with its assembly and study. It then reviews the current views as to the identities of the various species and provides excellent color photographs of specimens examined. The report is readable and well-written.

Experimental design

The scope of the project is well-defined and appropriate to the subject matter. The authors are knowledgeable of the subject matter (in particular, Breure has previously written extensively on the land snail fauna of the region, especially the Bulimulidae and related taxa which are a major portion of the material under study). The land snail fauna of that region was not well-known at that time of Hidalgo’s work, and thus resolution of uncertainties regarding the identities and taxonomy of the species upon which he reported is essential to a sound understanding of their taxonomy.

Validity of the findings

Although this reviewer is not personally familiar with the molluscan fauna of South America, the analysis presented appears to be meticulous and thorough, and the conclusions drawn are logically presented and well-stated. Remaining uncertainties, especially as to the availability or non-availability of type material, are identified.

---

## Round 0.2 · accepted · Accept

The authors have done a thorough job of addressing the questions and concerns that were raised in the previous revision. Although the MS could benefit from further edits for brevity, I can understand their request to retain the full text. The paper has also been registered with ZooBank. Overall, this study and revision of this impressive land snail collection is interesting. Thank you for sending your work to PeerJ.